# A human neural crest model reveals the developmental impact of neuroblastoma-associated chromosomal aberrations

Ingrid M. Saldana-Guerrero [1,2,3,13], Luis F. Montano-Gutierrez [4,13], Katy Boswell [1,2], Christoph Hafemeister [4], Evon Poon [5], Lisa E. Shaw [6], Dylan Stavish [1,2], Rebecca A. Lea [1,2], Sara Wernig-Zorc[4], Eva Bozsaky [4], Irfete S. Fetahu [4,12], Peter Zoescher [4], Ulrike Pötschger[4], Marie Bernkopf [4,7], Andrea Wenninger-Weinzierl[4], Caterina Sturtzel[4], Celine Souilhol[1,2,8], Sophia Tarelli[1,2], Mohamed R. Shoeb [4], Polyxeni Bozatzi[4], Magdalena Rados[4], Maria Guarini [9], Michelle C. Buri [4], Wolfgang Weninger[6], Eva M. Putz [4], Miller Huang [10,11], Ruth Ladenstein [4], Peter W. Andrews [1], Ivana Barbaric [1,2], George D. Cresswell [4], Helen E. Bryant [3], Martin Distel [4], Louis Chesler [5], Sabine Taschner-Mandl [4], Matthias Farlik [6], Anestis Tsakiridis [1,2,14] ✉ & Florian Halbritter [4,14] ✉

Early childhood tumours arise from transformed embryonic cells, which often carry large copy number alterations (CNA). However, it remains unclear how CNAs contribute to embryonic tumourigenesis due to a lack of suitable models. Here we employ female human embryonic stem cell (hESC) differentiation and single-cell transcriptome and epigenome analysis to assess the effects of chromosome 17q/1q gains, which are prevalent in the embryonal tumour neuroblastoma (NB). We show that CNAs impair the specification of trunk neural crest (NC) cells and their sympathoadrenal derivatives, the putative cells-of-origin of NB. This effect is exacerbated upon overexpression of *MYCN*, whose amplification co-occurs with CNAs in NB. Moreover, CNAs potentiate the pro-tumourigenic effects of *MYCN* and mutant NC cells resemble NB cells in tumours. These changes correlate with a stepwise aberration of developmental transcription factor networks. Together, our results sketch a mechanistic framework for the CNA-driven initiation of embryonal tumours.

Cancers in early childhood are driven by sparse genetic aberrations arising in utero, which are thought to lead to defective differentiation and uncontrolled proliferation[1–4]. Most tumours harbour large genomic rearrangements and chromosomal copy number alterations (CNA), which co-occur with mutations in tumour suppressors or tumourigenic transcription factors (TF)[5,6]. The mechanistic interactions between different mutations and early developmental processes are likely foundational drivers of tumour heterogeneity. However,

since visible tumours are only detected long after their initiation, early mutation-driven interactions leading to the healthy-to-tumour transition have remained largely intractable.

Neuroblastoma (NB) is the most common extra-cranial solid tumour in infants and an archetypal "developmental cancer"[7–9]. NB tumours are usually found in the adrenal gland or sympathetic ganglia, tissues derived from the trunk neural crest (NC) lineage during embryonic development[10,11], and studies using transgenic animal

models and transcriptome analysis have anchored NB tumourigenesis in impaired sympathoadrenal differentiation of trunk NC cells[12–23]. CNAs, such as gains of the long arms of chromosomes 17 (chr17q) and 1 (chr1q) have been identified in the majority (up to 65%) of NB tumours[24–28] and their emergence is considered an early tumourigenesis "priming" event[29]. Chr17q/1q gains often co-occur with amplification of the *MYCN* oncogene[24,28,30–33] (at least one CNA in >95% of *MYCN*-amplified tumours[34]), suggesting they may jointly contribute to tumourigenesis. However, despite our advanced understanding of the genetic and developmental origin of NB, it remains unclear to date how CNAs disrupt embryonic cell differentiation and lead to NB initiation.

Here, we used a human embryonic stem cell (hESC)-based model to experimentally dissect the links between NB-associated CNAs, *MYCN* amplification, and tumour initiation. We interrogated the stepwise specification of trunk NC and sympathoadrenal lineages using directed differentiation of isogenic hESC lines with chr17q/1q gains and inducible *MYCN* overexpression. We found that (i) CNAs derail differentiation by potentiating immature NC progenitor phenotypes. Combining CNAs with *MYCN* overexpression completely disrupted normal NC differentiation; (ii) Mutant NC cells acquired tumourigenic hallmarks in vitro, the capacity to form tumours in xenografts, and resemble distinct subpopulations of heterogeneous NB tumours; (iii) An extensive re-wiring of chromatin connects the observed transcriptional and functional aberrations with a dysregulated network of developmental TFs. Collectively, our data put forward a CNA-driven distortion of trunk NC and sympathoadrenal differentiation as a priming mechanism for subsequent MYCN-induced tumour initiation.

## Results

### Differentiation of human embryonic stem cells recapitulates key stages of trunk NC and sympathoadrenal development

To model the initiation stage and cell types relevant to NB tumourigenesis, we turned to an in-vitro modelling approach. We have previously described an efficient strategy to produce human trunk NC, sympathoadrenal progenitors, and sympathetic neurons from hESCs[35,36]. Our protocol involves treatment with defined cocktails of signalling pathway agonists/antagonists that induce neuromesodermal-potent axial progenitors (NMPs) at day 3 of differentiation (D3)[37], and subsequently steer NMPs toward trunk NC cells (D9) and their sympathoadrenal derivatives (>D14). At D19, the protocol yields catecholamine-producing sympathetic neurons marked by peripherin-expressing axons[35] (Fig. 1a, Supplementary Fig. 1).

As a prerequisite for studying the effects of CNAs on trunk NC differentiation, we first needed to define a molecular roadmap of normal hESC differentiation as a control. Therefore, we employed our protocol for the differentiation of karyotypically normal hESCs (H7[38]; 46XX) and performed droplet-based single-cell RNA sequencing (scRNA-seq) at key differentiation stages (D0 ≈ hESCs, D3 ≈ NMPs, D9 ≈ trunk NC, D14 ≈ sympathoadrenal progenitors, D19 ≈ early sympathetic neurons) and intermediate/late time points (D6, D10, D12, D28) to examine the resulting cell populations (up to five replicates each; Supplementary Data 1). We obtained 29,857 cells that passed quality control, which we allocated to 14 distinct clusters (C1-C14) (Fig. 1b; Supplementary Fig. 2a–g). We bioinformatically annotated these cell clusters using two complementary approaches: (i) by identifying characteristic marker genes (Fig. 1c; Supplementary Fig. 2h; Supplementary Data 2), and (ii) by mapping our data to single-cell transcriptomes of trunk NC derivatives in human embryos[15,16] (Fig. 1d–f, Supplementary Fig. 2i, j). This strategy identified cells at different stages of trunk NC development, including NMP-like cells (marked by *CDX1/2*, *NKX1-2*, and FGF signalling-associated transcripts[37]; cluster C2 in Fig. 1c, Supplementary Data 2) and later cell populations of a predominantly trunk axial identity (Supplementary Fig. 2h) exhibiting characteristics of Schwann cell precursors (SCP), sympathoblasts (SYM), as well as mesenchymal features (MES). For example, D9 cells

split into subpopulations expressing markers of trunk NC/early SCPs (C3; e.g., *SOX10*[16], Fig. 1c; weak SCP-like signature, Fig. 1e) and sensory neurons (C5; *ONECUT1*[39], Fig. 1c; weak SYM-like signature, Fig. 1f). At D14, cells started to assume a sympathoadrenal/autonomic progenitor (C8; *ASCL1*) or mesenchymal (C11; *FN1*) identity, and by day D19, we observed three distinct fractions: mature SCP-like cells (C9; *POSTN*[40]; strong SCP signature), autonomic sympathoblasts (C12-C14; *PHOX2A/B*, *ELAVL4*[16,41]; strong SYM signature), and MES-like cells (C11; *COL1A1, FN1*). This is in line with findings showing that trunk NC and SCPs are competent to generate mesenchyme[40,42,43]. Interestingly, we also found cells at the intersection of MES and SYM identity, as observed in mice[39] and NB cell lines[44–48] (Supplementary Fig. 3; Supplementary Data 3). After 4 weeks (D28), we also observed some cells with a partial chromaffin-like cell identity (part of C14; *CHGA*[+], PMNT[-]) (Fig. 1d).

Together, these data confirm that our hESC-based model successfully captures trunk NC and sympathoadrenal cells as found in embryos during the onset of NB tumourigenesis. Moreover, they reveal two major developmental branching events: (i) an early commitment of trunk NC toward a sensory neuron fate; (ii) the late generation of multipotent SCP/sympathoadrenal progenitors, which subsequently give rise to three distinct cell types: mature SCPs, MES, and SYM.

### CNAs and *MYCN* cumulatively disrupt human trunk NC differentiation

Having established a reliable model of trunk NC lineages relevant for NB pathogenesis, we next asked how chr17q and chr1q gains and their interplay with *MYCN* overexpression, which often co-occur in NB[24,28,30–34], influence NC development. To this end, we employed two clonal isogenic hESC lines with NB-associated CNAs that were acquired by H7 hESCs ("WT"; used in Fig. 1) as a result of culture adaptation[49] (Fig. 2a; Supplementary Fig. 4a): (i) a gain of chromosome arm 17q11-qter ("17q")[50], and (ii) an additional gain of chr1q in the 17q background ("17q1q"). Whole-exome sequencing of 17q and 17q1q cells compared to the parental H7 hESCs revealed a small number of additional mutations and a loss of a small region in chromosome 2 (Supplementary Fig. 4b; Supplementary Data 4, 5). For brevity, we labelled the cell lines by their major CNAs, which overlap regions commonly gained in NB tumours[51] (Supplementary Fig. 4c). 17q1q hESCs were engineered to include a Doxycycline (DOX)-inducible *MYCN* expression cassette to mimic *MYCN* amplification in a temporally controlled manner ("17q1qMYCN"). DOX treatment of 17q1qMYCN resulted in robust induction of MYCN, similar to expression levels in *MYCN*-amplified tumours (Supplementary Fig. 4d–f). In our experiments, we induced *MYCN* overexpression at D5 (when cells adopt a definitive NC identity[35]) to avoid bias toward central nervous system differentiation, as seen upon *MYCN* overexpression in earlier pre-NC progenitors[52].

Equipped with these three isogenic mutant hESC lines, we performed differentiation toward trunk NC and carried out scRNA-seq as described above, yielding a combined dataset comprising 95,766 cells (Supplementary Data 1). To assess how differentiation was affected in each mutant cell line, we first focused on stages D9, D14, and D19 for which we had data from all four experimental conditions, and bioinformatically mapped the transcriptomes of mutant cells to our reference of normal trunk NC differentiation (cp. Fig. 1b). While many 17q cells intertwined with all WT cell types (~98% matching the cognate WT stage), fewer 17q1q and 17q1qMYCN cells advanced beyond WT D14 (only ~48% and ~22% matched with WT, respectively; Fig. 2b). Only ~4% of 17q1qMYCN cells mapped to mature cell types (Fig. 2b). Altogether, at this level of resolution, we found no evidence that 17q affected differentiation. In contrast, 17q1q and 17q1qMYCN cells matched WT cells of earlier developmental stages, suggesting impaired differentiation (Fig. 2c).

Next, we tested whether the cell types induced from mutant hESCs still truthfully recapitulated in-vivo cell types as seen for WT. Mapping mutant cells onto the same human embryonic adrenal gland

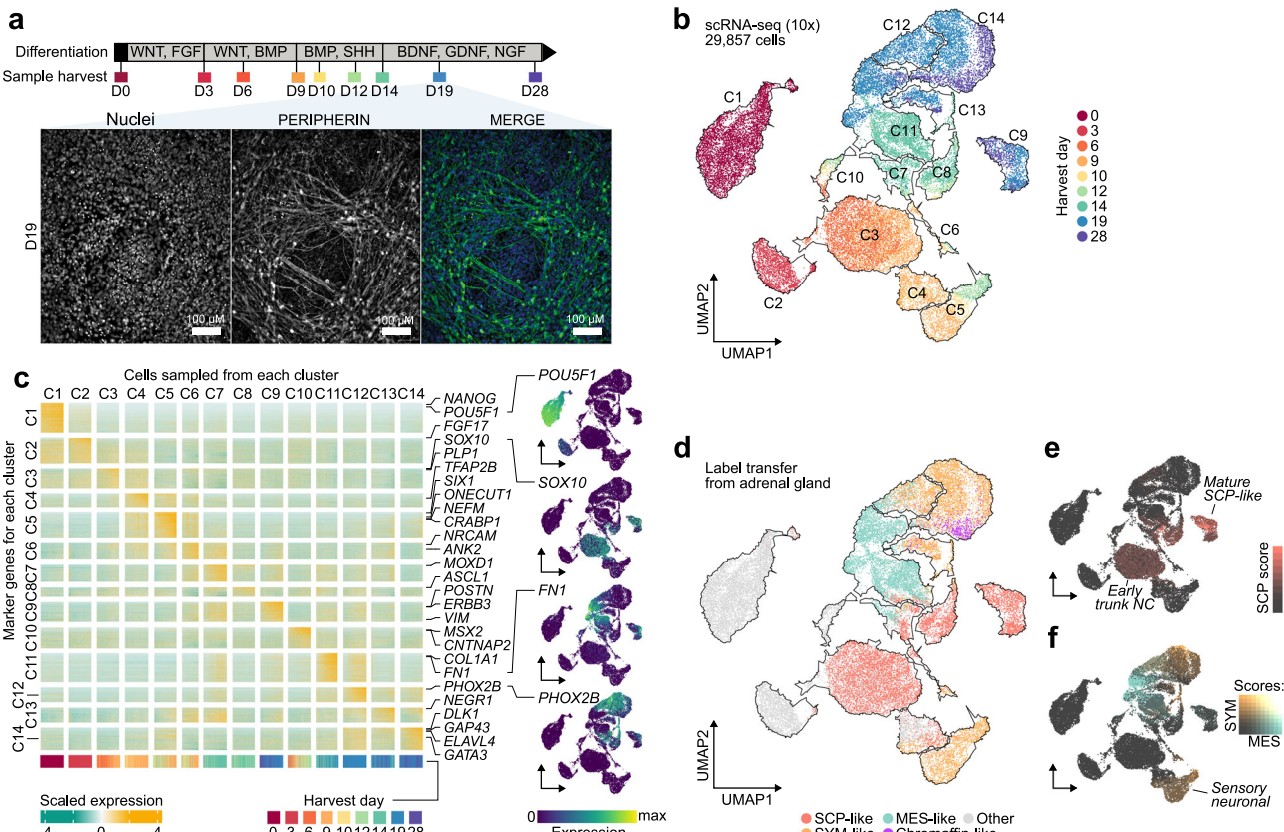

**Fig. 1 | In-vitro culture efficiently generates human trunk NC cells and their sympathoadrenal derivatives from hESCs. a** Diagram depicting (top) the extrinsically supplemented signals employed to direct hESCs toward trunk NC cells and their downstream derivatives, sample harvesting time points for scRNA-seq analysis (the colours indicated in this schematic are used throughout the manuscript to refer to the respective days), and immunofluorescence analysis (bottom) of PERIPHERIN protein expression illustrating the generation of sympathetic neurons at D19. Cell nuclei were counterstained using Hoechst 33342. All experiments were repeated at least three times with similar results. **b** UMAP of scRNA-seq data from wild-type hESCs collected at 9 stages (indicated by different colours) of differentiation to trunk neural crest and sympathoadrenal derivatives. Cells were divided into 14 distinct clusters as indicated by the contours. **c** Heatmap of gene markers for each cluster in (**b**). Selected genes have been highlighted and UMAPs indicate the expression level of canonical markers for stem (*POU5F1*), neural crest (*SOX10*), mesenchymal (*FN1*), and sympathetic (*PHOX2B*) cells. All marker genes are

reported in Supplementary Data 2. **d** UMAP as in (**b**), labelled by their closest matching cell type from the human embryonic adrenal gland reference[16] via label transfer. Cells in grey could not be verified with markers (Supplementary Fig. 2i) or could not be assigned to a single type. **e** Cells from panel d coloured by the strength of their SCP marker signature (Seurat module score) in red. The score distinguishes a cluster of early SCP-like/trunk NC (low score) and a late cluster with more mature SCP-like cells (high score). **f** Same as above but visualising simultaneously SYM (orange) and MES (teal) marker signature. Cells with overlapping marker signatures appear in grey tones, with the highest mixture in C12. An early diverging cluster of sensory neuron-like cells has a weak match to the SYM signature. A pseudotime trajectory for the MES-SYM transition in clusters C11-C14 can be found in Supplementary Fig. 3. Source data are provided as a Source Data file. hESC, human embryonic stem cells; D0/3/6/9/10/12/14/19/28, day 0/3/6/9/10/12/14/19/28; UMAP Uniform Manifold Approximation and Projection, C1-C14 cell clusters, SCP Schwann cell precursor, SYM sympathoblast, MES mesenchymal.

reference[16] identified proportionally fewer SYM- and MES-like cells in 17q1q and 17q1qMYCN (Fig. 2d, e). For cells mapped to the respective cell types, we observed a slightly stronger SCP signature in 17q and 17q1q, while the expression of both MES and SYM genes was weaker relative to the WT (Fig. 2f). In 17q1qMYCN, the expression of all signatures was weak, suggesting a failure to fully specify the expected cell types (Fig. 2d–g). Consistently, antibody staining for SOX10 and HOXC9 and flow cytometry revealed depletion of SOX10+ trunk NC cells in 17q1qMYCN cultures (Fig. 2h; Supplementary Fig. 5). The reduced ability of 17q1qMYCN hESCs to differentiate toward trunk NC derivatives was also reflected by their failure to generate PERIPHERIN-positive neuronal axons (Supplementary Fig. 4g). A similar, albeit milder effect was observed upon DOX-induced *MYCN* overexpression at later timepoints (Supplementary Fig. 4h).

Differential analysis identified 941 (17q vs. WT), 2039 (17q1q vs. WT), and 5915 (17q1qMYCN vs. WT) differentially expressed genes (DEG) at D9 (Supplementary Data 6). As expected, many upregulated genes were located within the known CNAs (41.4% within chr17q for 17q cells; 18.7% within chr17q and 25.6% within chr1q for 17q1q cells;

Supplementary Fig. 6a). Pathway analysis indicated an enrichment of genes related to E2F and MTORC1 signalling components for DEGs on chr17q (e.g., *BRCA1*, *NME1*), and of apoptosis-related and members of the p53 pathway on chr1q (e.g., the anti-apoptotic regulator *MCL1*; Fig. 3a–c; Supplementary Data 7). Notably, genes upregulated in 17q1q also include the p53 inhibitor *MDM4*[53] (Supplementary Data 6). These perturbed pathways may contribute to deregulation of expression of genes outside CNAs (e.g., upregulation of MYC targets and oxidative phosphorylation, and downregulation of G2-M checkpoint-related genes in 17q1qMYCN; Fig. 3a). To better resolve the molecular impact of each mutation, we integrated all datasets into a joint projection of WT and aberrant trunk NC differentiation (Fig. 4a; Supplementary Figs. 6b–h; Supplementary Data 8). The strongest changes were observed in 17q1qMYCN, which formed disconnected cell clusters not normally produced in our protocol. To delineate the stepwise alteration of transcriptional programmes, we placed cells from D9 on a spectrum from WT to 17q1qMYCN by scoring each cell between 0 and 1 based on the fraction of mutant cells among its gene expression neighbours ("mutation score"; Fig. 4b). This allowed us to identify four

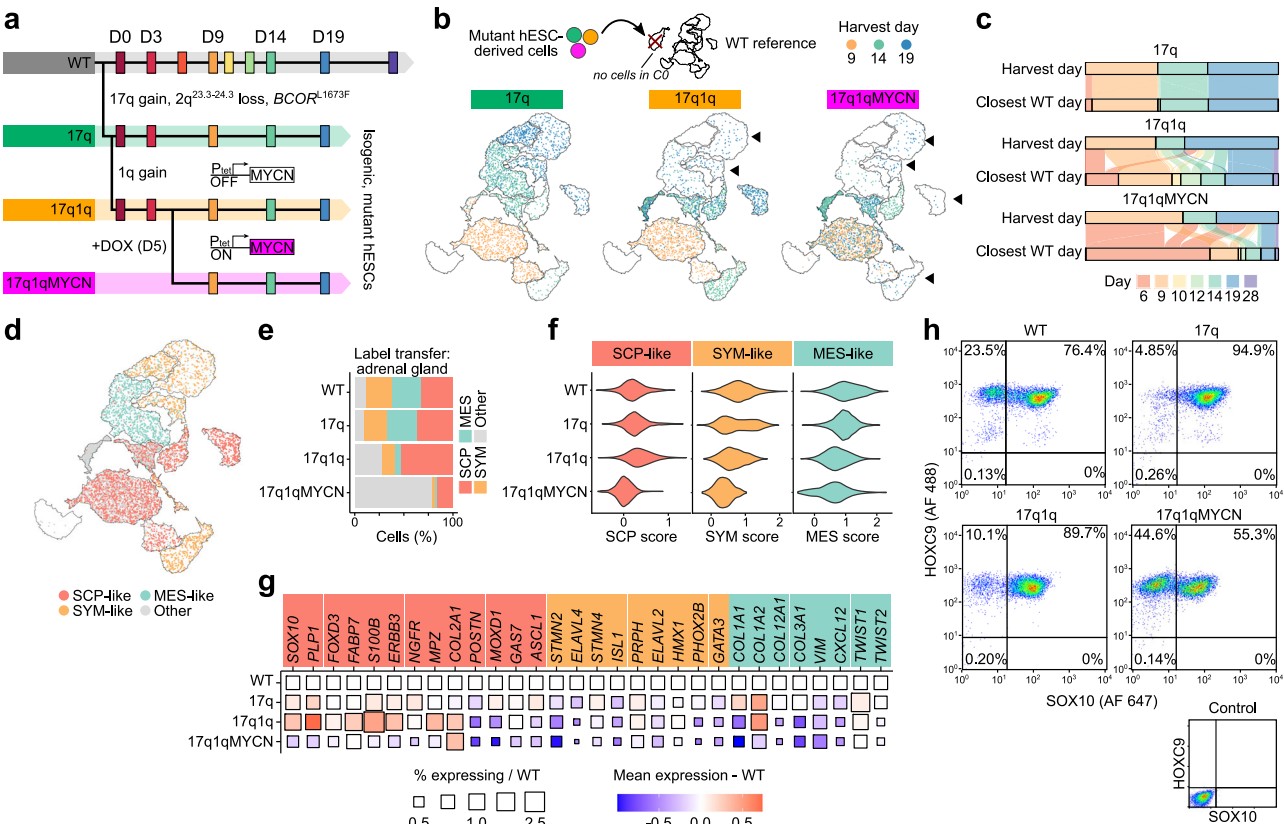

**Fig. 2 | Copy number alterations and overexpression of *MYCN* impair the specification of trunk NC derivatives. a** Scheme depicting the different hESC genetic backgrounds employed, the time points of sample collection, and the timing of DOX-induced *MYCN* overexpression during trunk NC differentiation. **b** scRNA-seq data from mutant cells (17q, 17q1q, 17q1qMYCN at D9, D14, and D19; coloured by stage) were mapped to the WT reference (cp. Fig. 1; undifferentiated hESC cluster C0 was excluded for simplicity). Glasswork UMAP plots depicting the destination clusters of mapped cells. Fewer 17q1q and 17q1qMYCN cells mapped to late stages (highlighted by arrows). **c** Alluvial plots comparing the stage at which each mutant cell was harvested versus its transcriptionally closest stage in the WT reference (based on label transfer as in **b**). In each subplot, the top bar indicates the proportion of cells collected at each stage (D9, D14, D19). The bottom bar indicates the distribution of matches for the same cells in WT, and streams indicate which subpopulations flow into cognate or non-cognate WT stages. The plots suggest that cells from 17q1q/17q1qMYCN mapped to earlier stages compared to WT. **d** Glasswork UMAPs of mapped 17q, 17q1q, and 17q1qMYCN cells (as in **b**) coloured by closest-matching cell type in an adrenal gland reference[16]. The category "other" comprises other cell types in the reference dataset and mappings that could not be validated by cell type markers (Supplementary Fig. 2i). **e** Percentage of cells mapped to each cell type in (**d**), split by cell line. **f** Violin plots indicating the strength of the SCP/SYM/MES (left to right) signature (Seurat module score) for cells mapped to the respective cell type, split by cell line. **g** Plot indicating the change in mean expression (colour) and the percentage of cells expressing the gene (size) for each gene in the signatures from panel e relative to WT. WT squares (size = 1, white) are shown for reference. **h** Flow cytometric analysis of trunk NC markers HOXC9 and SOX10 of WT, 17q, 17q1q, and 17q1qMYCN at D9. All experiments were repeated at least three times with similar results. Source data are provided as a Source Data file. WT wild-type H7 hESCs; D0/6/3/9/10/12/14/19/28, day 0/6/3/9/10/12/14/19/28; UMAP Uniform Manifold Approximation and Projection, SCP Schwann cell precursor, SYM sympathoblast, MES mesenchymal.

sets of genes (D9_1–D9_4) that were correlated with mutations (Fig. 4c, Supplementary Fig. 7a, b; Supplementary Data 9): Gain of CNAs led to a decrease in expression of genes (gene set D9_3, Fig. 4c) involved in NC development (e.g., *TFAP2B*[54,55]) and gradual induction of genes (D9_4, Fig. 4c) associated with NC/NB cell migration (e.g., *ZIC2*, *HOXD3*, *GPC3*[56–58]). *MYCN* overexpression in 17q1qMYCN further repressed genes related to NC development (D9_2; e.g., WNT-antagonist *SFRP1*[59] and nuclear receptors *NR2F1/2*[60]) and led to upregulation of MYCN targets implicated in NB (D9_1; e.g., *NME1* on 17q[61]; Supplementary Data 9). Interestingly, we had also found *SFRP1* and *NR2F1* to mark the SYM-MES transition state in WT differentiating sympathoadrenal cells (cp. Supplementary Data 3). Moreover, we found that many of genes that are upregulated in 17q1qMYCN (D9_1) were also highly expressed in NB tumours (Supplementary Fig. 7c).

We further sought to disentangle the relative contributions of *MYCN* overexpression and CNAs to the observed differentiation block phenotype in 17q1q cells. To this end, we generated additional cell lines derived from WT and 17q H7 hESCs by equipping each with a DOX-inducible *MYCN* construct (WTMYCN, 17qMYCN; Supplementary Fig. 8a). Moreover, we introduced the same inducible *MYCN*

expression cassette into a second female hESC cell line[38] (H9) which had independently acquired chr17q and 1q gains (H9-WT, H9-17q1q, H9-17q1qMYCN). The differentiation trajectories of these cell lines in the presence and absence of *MYCN* overexpression were interrogated using split-pool single-cell RNA-seq. To ensure consistency, we also included the previously analysed H7 cell lines (WT, 17q, 17q1q, 17q1qMYCN) and performed 2–4 replicate experiments. We obtained a total of 45,546 cells (all D9) post-QC and mapped each dataset onto our WT reference, as we had done before (Supplementary Fig. 8b). Starting with gain of chr17q, we found a reduction in cells allocated to sensory neuronal differentiation (cluster C4 in Fig. 1b, c) and instead a slight increase in a transitional progeny (C7). With chr1q, we also saw an increase of cells in C10. On top of these changes, *MYCN* overexpression led to most cells allocating to earlier developmental stages including clusters C2 and C3 (Supplementary Fig. 8b, c) – reflecting the differentiation block we had observed before (cp. Fig. 2b, c). The observed changes were consistent for derivatives of both parental hESC lines (H7 and H9). Analysis of marker gene expression associated with the altered cell clusters (C2, C4, C7, C10) in the different mutant cell lines identified an upregulation of genes like *AZIN1* in all

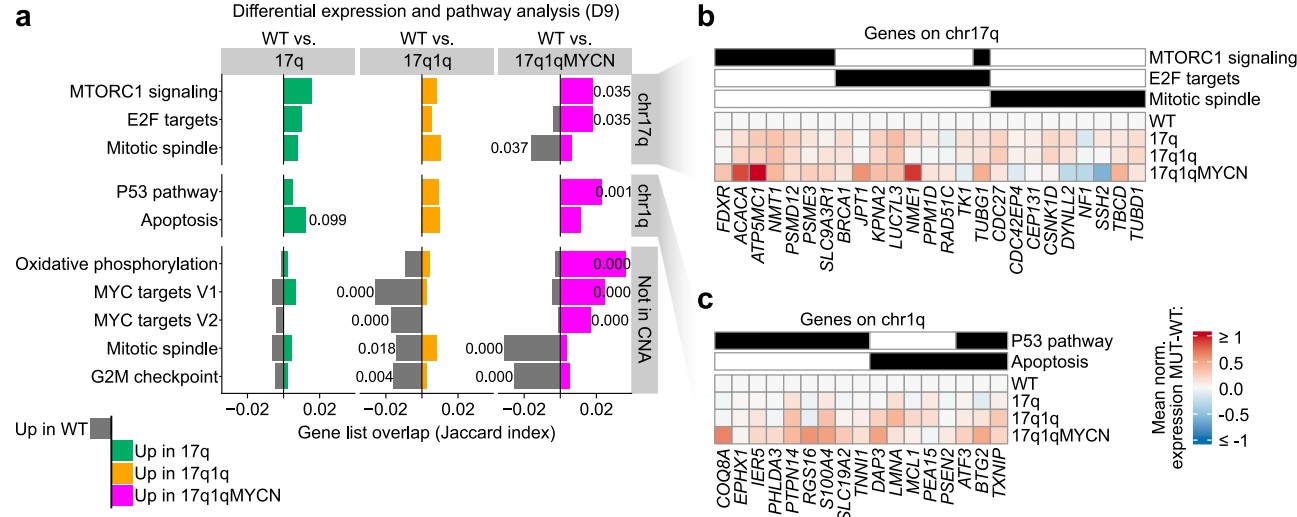

**Fig. 3 | Copy number alterations and *MYCN* overexpression influence cancer-associated pathways. a** We performed differential expression analysis between WT and mutant cells at D9 (two-sided tests using DESeq2[147] and aggregated pseudobulk counts per replicate[133]; *P* values were corrected for multiple hypothesis testing using the Benjamini-Hochberg method; $P_{adj} \leq 0.05$, $|log_2FoldChange| > 0.25$) and summarised differentially expressed genes (DEG) using pathway enrichment analysis. Enrichment was determined by hypergeometric tests (hypeR[135], background: all detected genes in our scRNA-seq dataset) using pathways from MSigDB[134]. The overlap between up- and down-regulated DEGs with the pathway genes is indicated as a positive (green/orange/magenta colour bars) or negative (grey colour) number, respectively. We additionally distinguished between DEGs located on chromosome arms chr17q, chr1q, or anywhere else in the genome to analyse potential direct and indirect effects of CNAs (split from top to bottom). All adjusted *P* values for enriched terms are shown ($P_{adj} \leq 0.1$; *P* values were adjusted for multiple hypothesis testing using the Benjamini-Hochberg method). Selected pathways are shown in the figure and all DEGs and pathway enrichments are available in Supplementary Data 6 and 7. DEGs located on chr17q (**b**) and chr1q (**c**) from the enriched pathways in (**a**). The heatmap indicates the mean normalised expression difference between each indicated mutant cell line and WT (at D9). The annotation bars on top of the heatmaps indicate membership (black colour) of genes in the selected pathways (MSigDB hallmark database). Source data are provided as a Source Data file. WT, wild-type H7 hESCs; MUT, a mutant hESC line (17q, 17q1q, or 17q1qMYCN); D9, day 9.

*MYCN*-overexpressing cells that was not active in their wild-type or CNA-only counterparts (Supplementary Fig. 8d). Conversely, these cells downregulated developmental regulators. For instance, even though MYCN-overexpressing cells still expressed remnant gene signatures leading them to map to differentiating WT cell clusters they downregulated genes in developmental pathways like *HHIP* in cluster C7 or *WNT1* in C10. Moreover, the neurogenic capacity of the mutant trunk NC cells (as reflected by the presence of PERIPHERIN-positive neuronal axons) was found to be disrupted by *MYCN* overexpression primarily in the presence of CNAs, with the strongest phenotype being observed in the presence of both chr17q and chr1q gains (Supplementary Fig. 8e), in line with our earlier findings. Collectively, these data indicate that CNAs potentiate the *MYCN*-driven block to the induction of a trunk NC/sympathoadrenal identity.

We conclude that NB-associated CNAs alter the differentiation landscape of hESC-derived trunk NC lineages by promoting transitional progenitor states at the expense of mature sympathoadrenal cell types. In conjunction with *MYCN* elevation, they block differentiation and trigger atypical transcriptional programmes incompatible with normal trunk NC development.

**Impaired trunk NC differentiation correlates with acquisition of tumourigenic hallmarks**

We observed that ectopic *MYCN* induction altered the morphology of cultures by D14 only in the presence of CNAs as cells lost their ability to spread out and form neurites, and 17q1qMYCN cells even formed tight, dome-like colonies (Fig. 5a). As this phenomenon is reminiscent of loss of contact inhibition, a cancer hallmark, we next examined whether CNAs/*MYCN* overexpression led to further cellular changes that are typical of tumourigenesis. We first carried out cell cycle analysis of trunk NC cells (D9) generated from each *MYCN*-overexpressing hESC line (from WT/17q/17q1q backgrounds) by monitoring EdU (5-ethynyl-2´-deoxyuridine) incorporation via flow cytometry.

We found a significant increase in the proportion of cells in S-phase only when *MYCN* overexpression was combined with CNAs ($p = 0.0233$ and $p = 0.0073$ respectively; two-way ANOVA; Fig. 5b), indicating altered cell cycle and increased replication similar to NB tumours and cell lines[62–64]. Immunofluorescence analysis of Ki-67 expression further showed that 17q1qMYCN and 17qMYCN cultures exhibited a higher proliferation rate by D14 compared to their CNA-only counterparts ($p < 0.0001$ and $p = 0.0078$, respectively; two-way ANOVA; Fig. 5c). We next tested how CNAs/*MYCN* influenced colony formation, another hallmark of tumourigenesis. Low-density plating of trunk NC cells (D9) and image analysis showed significantly increased clonogenicity ($p = 0.0109$; two-way ANOVA) exclusively in 17q1qMYCN cells (Fig. 5d). DOX treatment of the unedited parental wild type and chr17q gain cell lines had no effect (Supplementary Fig. 9a).

Previous work has indicated that *MYCN* overexpression alone is associated with increased apoptosis in early sympathoadrenal cells[65,66] and can trigger tumourigenesis only in combination with additional mutations[13,14,67]. Therefore, we also examined apoptosis levels during the transition of D9 trunk NC cells toward the SCP/sympathoblast stage (D14) by assessing cleaved Caspase-3 levels using flow cytometry. We found that *MYCN* overexpression indeed resulted in a higher rate of apoptosis in the WT background, while this increase was reversed in 17q cells (Supplementary Fig. 9b). However, this was not the case in cultures derived from *MYCN*-overexpressing cells with chr17q1q gains, which exhibited apoptosis levels similar to their *MYCN*-overexpressing WT counterparts (Supplementary Fig. 9b, c). This phenomenon may be linked to the presence of both pro- and anti-apoptotic genes in chromosome arm chr1q (cp. Fig. 3c) as well as increased DNA damage (assessed by the presence of γH2AX foci) following *MYCN* overexpression specifically in the 17q1q background (Supplementary Fig. 9d, e). Interestingly, we detected lower levels of DNA damage in the absence of MYCN overexpression in 17q and 17q1q trunk NC cells at D9 compared to WT controls suggesting a potential protective effect of

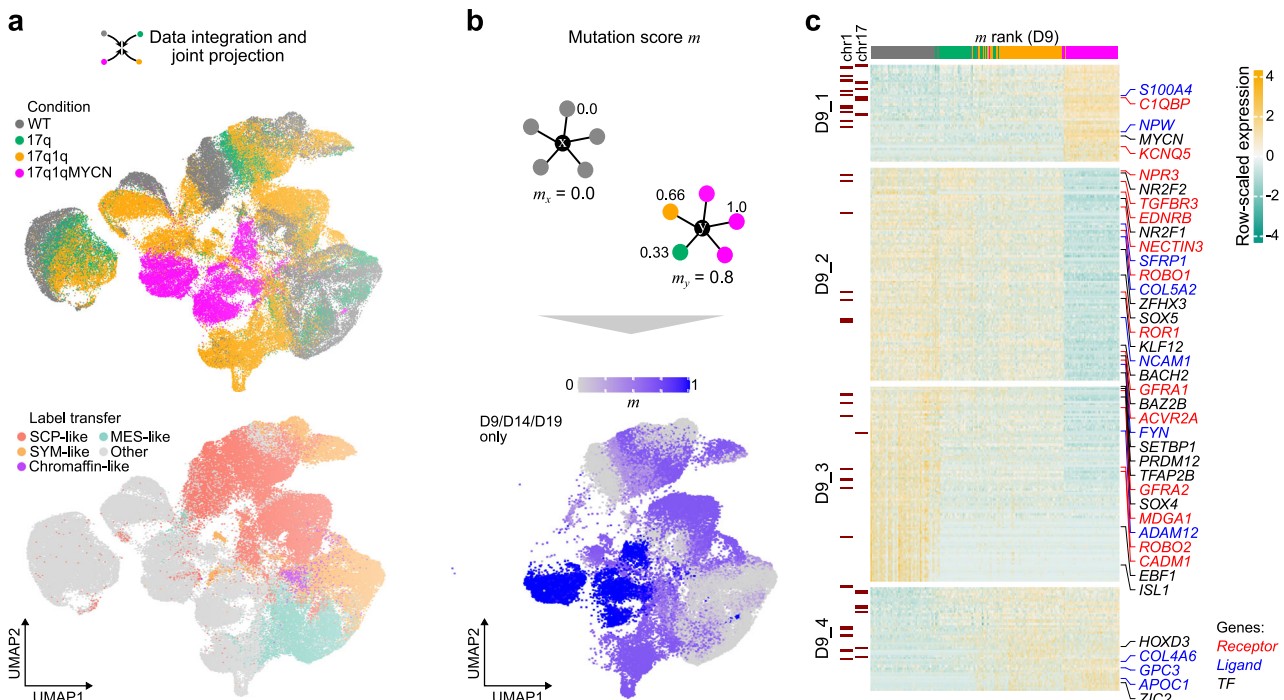

**Fig. 4 | Neuroblastoma-like genetic aberrations cumulatively distort trunk NC differentiation. a** Top: UMAP of scRNA-seq data from WT and mutant hESCs (indicated by colour) throughout differentiation. Bottom: the same dataset coloured by closest-matching cell type in an adrenal gland reference[16]. The category "other" comprises other cell types in the reference dataset and mappings that could not be validated by cell type markers (Supplementary Fig. 2i). **b** Illustration (top) of the calculation of mutation scores $m$ as average score of each cell's neighbours (k-nearest neighbour average). In this calculation, each neighbour weighs in by its cell line (0 = WT, 1/3 = 17q, 2/3 = 17q1q, 1 = 17q1qMYCN) such that the mutation score allows ordering cells from WT to MYCN mutation. Only cells from D9, D14, and D19 were used, for which data from all conditions were available. The actual scores are shown overlaid on the UMAP from panel a (bottom). **c** Heatmap showing the expression of the top genes correlated with the mutation score (from b) across all cells from D9. Genes have been divided into four groups by hierarchical clustering, and selected TFs, receptors, and ligands are highlighted in different colours. All correlated genes are reported in Supplementary Data 9. Genes located on chr17q or chr1q are indicated. Source data are provided as a Source Data file. WT, wild-type H7 hESCs; D9/14/19, day 9/14/19; UMAP Uniform Manifold Approximation and Projection, m mutation score, TF transcription factor, SCP Schwann cell precursor, SYM sympathoblast, MES mesenchymal.

17q/1q gains (Supplementary Fig. 9d). Moreover, we investigated whether *MYCN*-overexpressing cells from different backgrounds had acquired additional mutations during differentiation. Whole-exome sequencing analysis at D19 of differentiation did not reveal any new large CNAs and detected only few mutations (<10 mutations with variant allele frequency ≥0.2 between D19 and its common ancestor with D0; Fig. 5e, Supplementary Fig. 9f; Supplementary Data 4). None of the discovered mutations have previously been reported in NB, leading us to conclude that the observed phenotypic changes in 17q1qMYCN were likely a product of the CNAs and *MYCN* overexpression rather than an expansion of new clonal cell populations with additional pathognomonic mutations. Despite an increase of proliferation (cp. Fig. 5b, c), *MYCN* overexpression did not yield more high-frequency mutations during the short timeframe of our differentiation experiments, consistent with earlier work in human neuroepithelial stem cells in vitro and after xenotransplantation[68] ($p = 0.3458$, two-sided, paired Wilcoxon test, $n = 3$ per group; Supplementary Fig. 9g).

Finally, we sought to examine the tumourigenic potential of 17q1q-, 17q1qMYCN- and WT-derived trunk NC (D9 of differentiation) cells in vivo by xenografting them into immunodeficient NSG mice. We first injected aliquots of about 1 million cells subcutaneously into the recipient animals ($n = 6$ per cell line) and monitored tumour volume over time. After 3–5 weeks with continuous DOX administration, all 17q1qMYCN-injected mice developed visible tumours at the injection site (Fig. 6a). In contrast, neither WT- nor 17q1q-injected control animals displayed any signs of tumours for up to 16 weeks (Fig. 6a). Likewise, orthotopic injection into the adrenal gland ($n = 3$ mice per condition) yielded tumour growths visible by magnetic resonance imaging (MRI) after 5 weeks only when *MYCN* overexpression was

induced by DOX in 17q1qMYCN-grafted animals (Fig. 6b, c). We found that both subcutaneous and adrenal xenograft-derived tumours consisted of undifferentiated, small round cells similar to tumours from transgenic Th-*MYCN* mice[12] (Supplementary Fig. 10a). Complementary to our analysis in mice, we also performed exploratory xenografts of the same cell lines in zebrafish larvae. To this end, we labelled our 17q1qMYCN and WT cells at D9 with a fluorescent dye (CellTrace Violet) and injected them into the perivitelline space of zebrafish larvae on day 2 post fertilisation. Consistent with our results in mice, we found that 17q1qMYCN cells survived longer in zebrafish than WT, which had diminished in number at day 1 post injection (dpi) and were completely absent at 3 dpi (Supplementary Fig. 10b, c). In contrast, 17q1qMYCN cells survived until 3 dpi, with 16% of larvae even showing an increase in xenotransplant size. For comparison, injection of cells from a *MYCN*-amplified NB cell line (SK-N-BE2C-H2B-GFP[69]) resulted in engraftment with subsequent tumour cell growth in 84% of larvae (Supplementary Fig. 10d).

Together, our results demonstrate that CNA-carrying trunk NC cells transit into an undifferentiated pre-tumourigenic state and acquire altered cellular properties reminiscent of cancer hallmarks, such as increased proliferation, clonogenic and tumour formation capacity under the influence of *MYCN* overexpression. Our data also suggest that CNAs enhance the pro-tumourigenic effects of *MYCN*.

**In-vitro differentiation of mutant hESCs captures NB tumour cell heterogeneity**

Given that cells in our in-vitro model exhibit similarities to NB cells, we asked whether our data could provide insights into cellular heterogeneity in NB tumours. To this end, we collected scRNA-seq data from

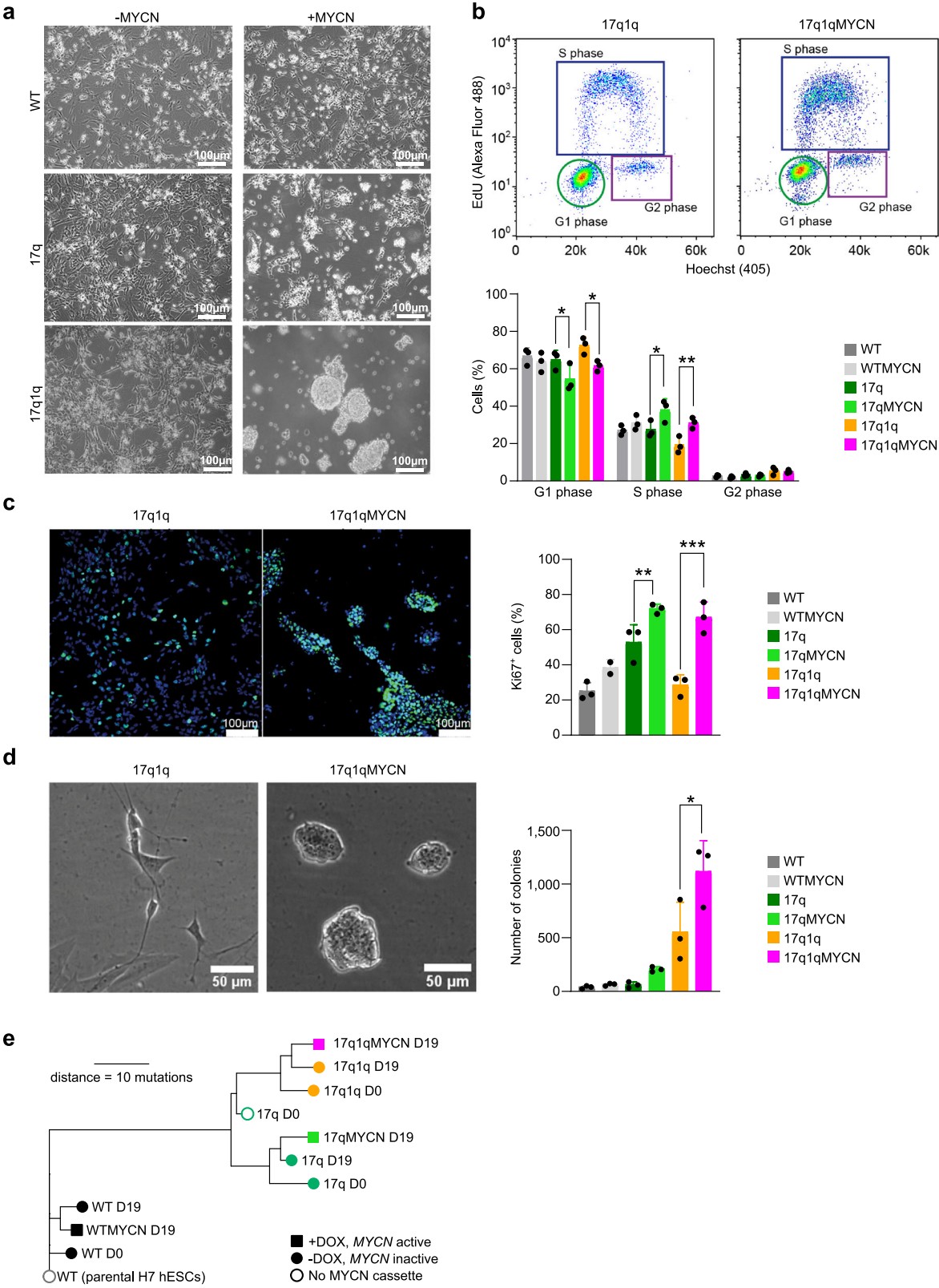

ten *MYCN*-amplified NB tumours from three independent sources[15,17,23]. For each dataset, we curated *MYCN*⁺ tumour cells and bioinformatically mapped these to our reference (Fig. 7a). For example, this approach matched most cells from tumour dataset *Jansky_NB14*[15] to clusters C13 and C14, which correspond to late SYM-like cell states (Fig. 7b; cp. Fig. 1). A few cells also mapped to clusters C11 and C12, i.e., cells with MES-like characteristics. The observed transcriptional heterogeneity

was surprising, given that most tumour cells appeared karyotypically homogeneous (including a chr17q gain) and expressed *MYCN* (Fig. 7b).

Extending the in-vitro reference mapping to all ten tumours portrayed a spectrum of *MYCN*-amplified cells with a majority C13- or C14-like expression profile, and a subset of cells mapping to other differentiating trunk NC cell states (Supplementary Fig. 11a, b). We observed apparent differences between studies and tumour types, but

**Fig. 5 | Impaired trunk NC specification correlates with acquisition of tumourigenic hallmarks. a** Representative brightfield images of D14 cultures following differentiation of hESCs with the indicated genotypes. All experiments were repeated at least three times with similar results. **b** Flow cytometric analysis of cell cycle in D9 cultures of each cell line. Top: Representative FACS plots. Bottom: Percentage of cells in each cell cycle stage. G1 (17q vs. 17qMYCN, $p = 0.0266 = *$; 17q1q vs. 17q1qMYCN, $p = 0.0153 = *$), S (17q vs. 17qMYCN, $p = 0.0233 = *$; 17q1q vs. 17q1qMYCN, $p = 0.0073 = **$). Only comparisons examining the effect of *MYCN* overexpression in different backgrounds are shown. **c** Immunofluorescence analysis of the cell proliferation marker KI-67 (green) in D14 cultures of each cell line. Cell nuclei were counterstained using Hoechst 33342 (blue). Left: Representative images. Right: Percentage of KI-67-positive cells. 17q vs. 17qMYCN, $p = 0.0078 = **$; 17q1q vs. 17q1qMYCN, $p = 0.0001 = ***$. **d** Low-density plating of D9 cultures of each cell line ($n = 3$ biological replicates). Left: Representative brightfield images after 84 h. Right: Number of colonies counted after 5 days. 17q1q vs. 17q1qMYCN, $p = 0.0109 = *$. **b**–**d** Bar plots showing the mean of $n = 3$ biological replicates (error

bars = SD). Statistical analysis was performed using ordinary two-way (**b**, **c**) or one-way (**d**) ANOVA with Tukey correction. Only comparisons examining the effect of *MYCN* overexpression in different backgrounds are shown. **e** Phylogenetic tree indicating the genetic relationship and distance (number of SNVs detected by whole-exome sequencing) between different hESC lines before (D0) and after differentiation (D19). Node shape indicates samples without a *MYCN* overexpression cassette (unfilled circles), with an inactive cassette (filled circles), and with an activated (by addition of DOX from D5 onwards) cassette (filled squares). The colours match those used elsewhere in the paper, without specific meaning. Short distances (<10 mutations) between differentiated cells and the shared ancestor with the matching D0 samples suggest that few additional mutations occurred during differentiation. Supplementary Data 4 and 5 report SNVs/CNAs identified in our analyses. Source data are provided as a Source Data file. D0/5/14/19, day 0/5/14/19; WT, wild-type H7 hESCs, SD standard deviation, SNV single-nucleotide variant, CNA copy number alteration.

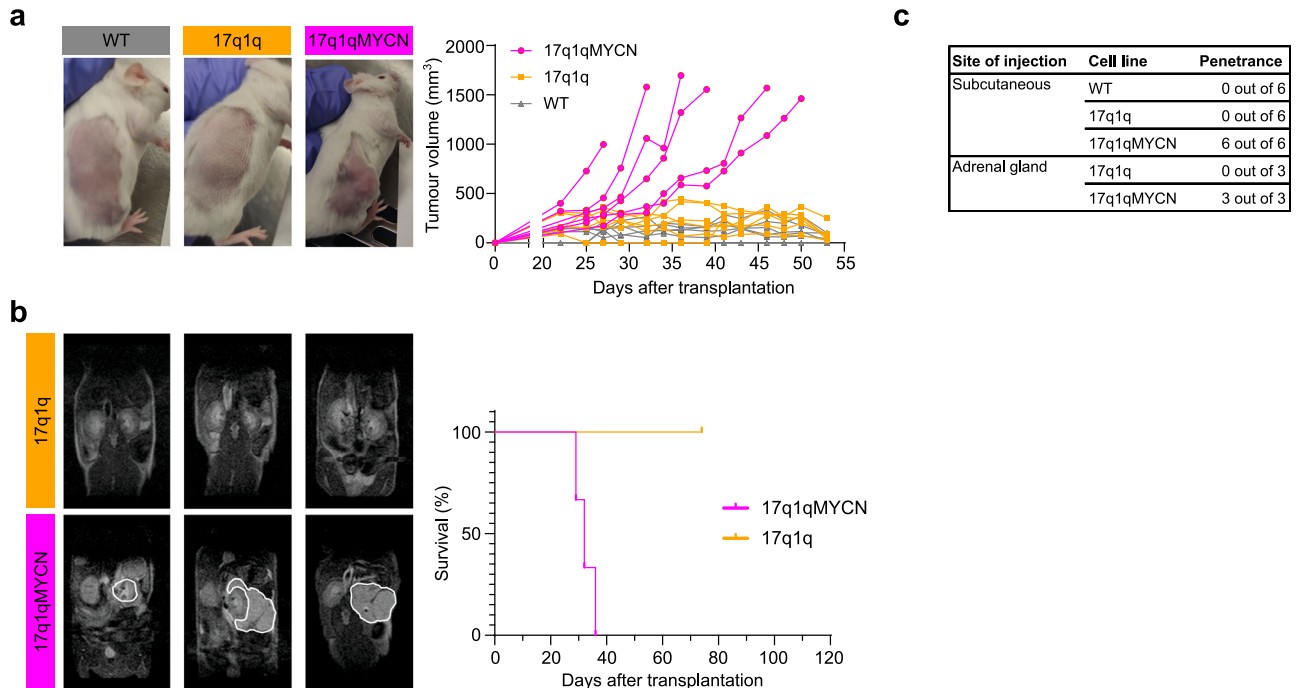

**Fig. 6 | hESC-derived trunk NC cells with CNAs form tumours in mice upon *MYCN* overexpression. a** Left: Representative images of subcutaneous xenografts of trunk NC cells derived from the indicated cell lines in the presence (17q1qMYCN) and absence (WT, 17q1q) of DOX treatment. Right: Graph showing tumour growth per mouse corresponding to xenografts of indicated cell lines ($n = 6$ animals per cell line). **b** Left: Representative MRI sections of mice at week 5 following xenografting

of indicated cell lines in the adrenal gland and DOX treatment regimens. The white lines indicate the tumour perimeter. Right: Graph showing survival of the recipient animals after xenografting ($n = 3$ animals per cell line). **c** Summary of mouse xenograft experiments. Source data are provided as a Source Data file. DOX doxycycline, MRI magnetic resonance imaging.

to date there is only a limited amount of single-cell data from NB tumours to robustly interpret such heterogeneity. We therefore sought to interrogate a large collection of bulk RNA-seq data from NB tumours (SEQC[70,71]). We first intersected the development-related gene signatures (C1-C14 from Fig. 1) with marker genes identified for the tumour cells that had been mapped to those respective clusters (from all 10 investigated samples; Fig. 7c; Supplementary Data 10) and labelled each refined signature with an asterisk to distinguish it from the original gene signature (e.g., signature C13* contained genes such as *DLC1* and *RORA*; Fig. 7c). Applying these gene signatures to the NB tumour data, we found that expression signatures C5* (sensory neuron-like cells) and C13* (differentiating SYM-like cells), jointly separated *MYCN*-amplified and non-amplified tumours, as well as tumours at different clinical stages (Fig. 7d). The C13* signature effectively stratified tumours with a good and poor survival across the entire cohort even when corrected for INSS stage, *MYCN* amplification

status, and age (Cox regression analysis with covariates; Fig. 7e; Supplementary Data 11).

Jointly, these observations demonstrate that our in-vitro model generates cell types that transcriptionally resemble different NB cell subpopulations and that it facilitates the systematic dissection of intratumour heterogeneity in NB tumours.

### CNAs and *MYCN* disrupt the configuration of NC regulatory circuits during differentiation

NB tumours and cell lines are marked by a re-wiring of non-coding regulatory elements (e.g., enhancers) giving rise to tumour-specific regulatory circuitries[44,45,72–76]. We therefore hypothesised that disruption of developmental TFs also underpins the aberrant differentiation observed in our mutant hESCs (cp. Figs. 2, 3) and employed the assay for transposase-accessible chromatin followed by sequencing (ATAC-seq[77]) to profile chromatin accessibility in the same samples used for

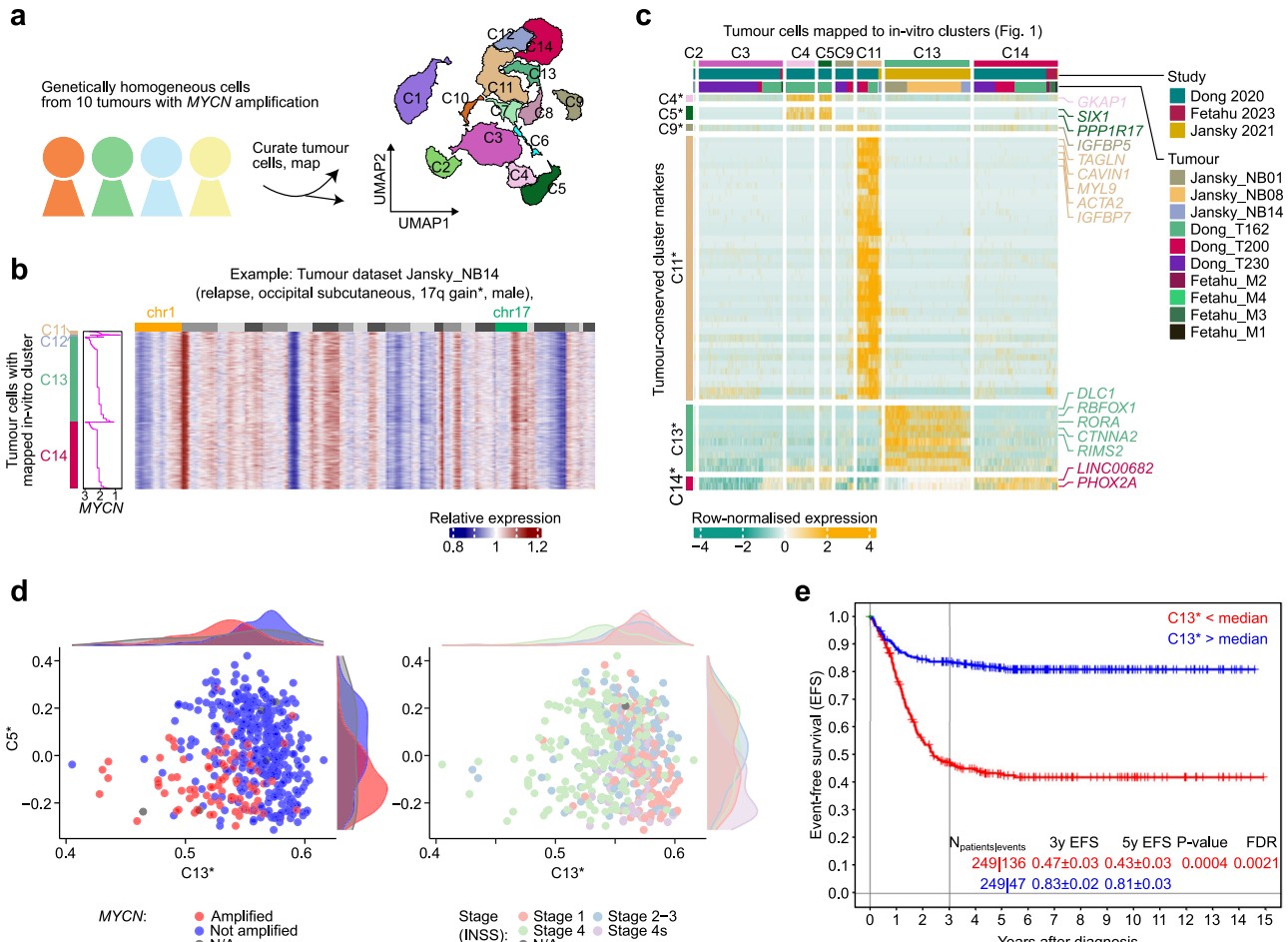

**Fig. 7 | Comparison to hESC-based trunk NC differentiation resolves structured heterogeneity across neuroblastoma tumours. a** We curated tumour cells from 10 *MYCN*-amplified NB samples[15,17,23] from three studies and mapped them onto our WT in-vitro reference (cp. Fig. 1)[126]. Mapping is represented as cells in each cluster of the reference (depicted as contours). **b** Heatmap depicting relative gene expression in dataset Jansky_NB14[15]. Values are inferCNV[139] copy number estimations per gene, relative to haematopoietic cells ordered by genomic position and chromosome (1–22). Cells (one per row) are ordered by the respective cluster that they were mapped to (C11 to C14) and therein by *MYCN* expression levels (depth-normalised sliding window average, width = 20 cells). Mappings of other datasets are shown in Supplementary Fig. 11. **c** Heatmap showing markers from gene expression signatures C4*, C5*, C9*, C13*, and C14* (rows, top to bottom) in cells from 10 tumour datasets that were mapped to our in-vitro reference (cp. **a**). Each signature is the intersection of the cluster markers in our reference (as in Fig. 1) and differentially expressed genes between the respective tumour cells. Markers for tumour cells

mapped to C2 and C3 showed no overlap with in-vitro cluster markers; hence, only mapped cells are shown. All genes identified in this analysis are reported in Supplementary Data 10. **d** Scatterplots evaluating the strength of signatures C5* and C13* (from **c**; calculated using GSVA[148]) in bulk RNA-seq data from SEQC[70,71]. Each dot indicates one tumour dataset coloured by *MYCN* amplification status (left) or clinical stage (right). The density of points (kernel density estimate) in each group is indicated in the margins. **e** Survival analysis for data from the SEQC cohort stratified by strength of the C13* signature. Groups were split by the median. Cox regression adjusted for age-group (<18, 18–60, or >60 months), INSS stage 4 (yes/no), and *MYCN* amplification (yes/no). n = 249 patients per group, or 136 [C13* low] and 47 [C13* high] events. All results are reported in Supplementary Data 11. Source data are provided as a Source Data file. UMAP Uniform Manifold Approximation and Projection, EFS event-free survival, INSS International Neuroblastoma Staging System, FDR false discovery rate.

scRNA-seq analysis (*n* = 51; Supplementary Data 1). Chromatin accessibility serves as a proxy for the dynamic regulatory DNA element activity during differentiation. For instance, the promoters of the hESC regulator *POU5F1* and trunk NC regulator *SOX10* were accessible only at D0 and D9, respectively (Fig. 8a), while the *PHOX2B* promoter exhibited reduced accessibility in 17q1q and 17q1qMYCN cells at D19 consistent with impaired differentiation (Fig. 8b).

Unsupervised analysis of chromatin patterns on a global level showed that WT and 17q hESCs changed consistently throughout differentiation (Fig. 8c). In contrast, 17q1q and 17q1qMYCN appeared not to follow the same path as WT in this low-dimensional projection, in line the differentiation defects observed in our previous analyses (cp. Fig. 2b, c). To delineate chromatin changes in detail, we performed differential accessibility analysis between all differentiation stages per cell line and between all cell lines at matched stages (Supplementary

Data 12, 13). As in our DEG analysis, we found an increasing number of regions with altered accessibility in 17q (*n* = 477 regions), 17q1q (*n* = 2826), and 17q1qMYCN (*n* = 6663; Fig. 8d). In total, there were 45,583 regions with differential accessibility in at least one comparison, which we divided into nine chromatin modules R1-R9 (Fig. 9a). Modules R1-R7 reflect differentiation order, e.g., regions in module R1 were mostly accessible at D0, and R6 comprises regions accessible at D14 and D19. Most changes observed in mutant hESC-derivatives fell within these modules (Supplementary Fig. 12a, b). 17q1q and 17q1qMYCN cells failed to close chromatin that is usually specific to D9 (R4, R5) and conversely to open chromatin regions indicative of late sympathoadrenal differentiation (R6, R7; Supplementary Fig. 12c). Additionally, modules R8 and R9 comprised regions with reduced and increased accessibility in mutant hESC derivatives, respectively, independently of differentiation stage.

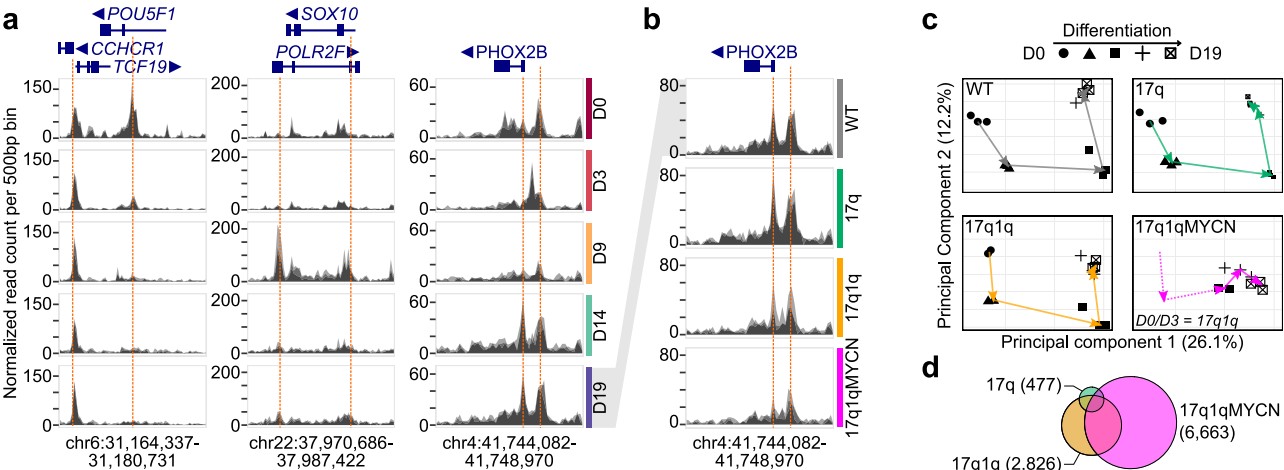

**Fig. 8 | Copy number alterations and *MYCN* overexpression disrupt chromatin reconfiguration during trunk NC differentiation. a** ATAC-seq read coverage for WT cells at three example loci. The area reports the normalised read count aggregated per genomic bin (width = 500 bp). Multiple semi-transparent plots are overlaid for each replicate. Genes within each locus are shown on top with thin/ thick lines indicating introns/exons. The arrows next to gene names indicate the direction of transcription. Selected peaks have been highlighted manually (dashed orange lines). **b** ATAC-seq read coverage of WT and mutant cells at D19 near the *PHOX2B* locus. Plots as in (**a**). **c** Principal component analysis of all ATAC-seq

datasets, split by condition. Each data point represents one ATAC-seq sample with the shape indicating the stage at which the sample was collected. The geometric means of all data of the same stages are connected by arrows to visualise the stepwise chromatin changes. **d** Euler diagram visualising the overlap of differentially accessible regions in mutant hESC-derived trunk NC derivatives compared to WT (*DEseq2*[147]; $P_{adj} \leq 0.005$, $|log_2FoldChange| \geq log_2(1.5)$). Numbers indicate the total number of regions per cell line aggregated over all stages. Source data are provided as a Source Data file. D0/3/9/14/19, day 0/3/9/14/19; WT, wild-type H7 hESCs.

We sought to annotate our chromatin modules by looking for overlaps with genomic regions accessible in human tissues[78–80] (Supplementary Data 14). In line with our transcriptome data, we found a stepwise change toward chromatin resembling differentiated tissues, e.g., neural tissues in R3-R5 and mesenchyme/stroma in R6/R7 (Supplementary Fig. 12d–f). Next, we examined the identity of genes near the chromatin modules (Fig. 9b). For each module, we found enrichments of specific marker genes identified in our scRNA-seq analysis of WT trunk NC differentiation (i.e., clusters C1-C14 from Fig. 1b, c). For example, chromatin module R7 (accessible in late differentiation stages, lost in mutants) was linked to clusters C11/C12 (MES-like cells). Next, we examined TF binding motifs in each module to identify potential upstream regulators (Fig. 9c). We found an enrichment of known regulators of each developmental stage, e.g., TFs associated with trunk NC in R3-R5 (e.g., SOX10) and with sympathetic neuron development in R6/R7 (e.g., PHOX2A/B)[39]. Moreover, we found enriched overlaps of modules R2/R4/R5/R8/R9 with super-enhancers identified in mesenchymal NB cell lines or adrenergic super-enhancers (in the case of R8), depending on the source annotation used[44,45]. Furthermore, R7 and R9 overlapped with super-enhancers associated with subsets of NB tumours[73] with mesenchymal characteristics and with non-*MYCN*-amplified high-risk tumours, respectively (Fig. 9d). No significant overlap was found with super-enhancers specific for *MYCN*-amplified NB. Finally, we examined the accessibility of each module across a range of NB cell lines (Supplementary Fig. 12g). As expected, we found that modules R1 and R2 (undifferentiated, early embryonic developmental stages) and modules R4 and R5 (early trunk NC to sympathoadrenal differentiation) were not accessible in NB cell lines, while modules R6-R8 (late sympathetic neurons and consistently open in mutants) were often highly accessible in cell lines. Interestingly, R3 (accessible at NMP and NC stage) was accessible in most NB cell lines examined except in those with mesenchymal characteristics (SK-N-AS and SHEP; Supplementary Fig. 12g). Using data from other studies, we found that R6-R8 were also accessible in non-NB cell lines and tissues, while R3 was only found accessible in brain tissue (Supplementary Fig. 12g).

Together, our results suggest a systematic reprogramming of chromatin throughout trunk NC differentiation. In cells with CNAs and

*MYCN* overexpression, this orderly reconfiguration of chromatin was severely disrupted in a manner similar to NB cells, providing a plausible mechanism for the link between the observed developmental defects and tumour initiation.

## CNA/*MYCN*-driven cell identity loss is mediated by sets of developmental transcription factors

Finally, we investigated the links between CNA/*MYCN*-induced changes in chromatin dynamics, gene-regulatory networks, and the distorted differentiation trajectories observed at the transcriptional level. In our scRNA-seq analyses, we had recorded a stepwise alteration of expression from WT to 17q1qMYCN at D9 comprising four gene sets: D9_1 – D9_4 (cp. Fig. 4c), which revealed *MYCN*-driven disruptions of early NC and the sensory neuron lineage specification. We hypothesised that these mutation-linked gene sets were also regulated by distinct TFs and therefore we employed an algorithm to identify TF targets based on correlated expression patterns[81] (Fig. 10a). This analysis identified *NR1D1* and *TFAP4* as putative TF targets of MYCN (Fig. 10b, c; Supplementary Fig. 13a, b; Supplementary Data 15). The nuclear receptor *NR1D1* has been shown to correlate with *MYCN* amplification in NB patients[82,83] and *TFAP4* inhibition leads to differentiation of *MYCN*-amplified neuroblastoma cells[84,85], supporting the validity of the inferred target genes.

We intersected the inferred lists of TF targets with the mutation-linked gene sets (D9_1 – D9_4) and found an enrichment (Fig. 10d; Supplementary Data 16) of MYCN, NR1D1, TFAP4, and ZIC2 targets in D9_1 (highly expressed in 17q1qMYCN). Conversely, the gene set D9_2 (expressed in WT/17q/17q1q) was enriched for targets of TFs expected at this stage of differentiation, e.g., SOX4/5/10, TFAP2A/B, and nuclear receptors NR2F1/2. The expression of targets of these TFs increased or decreased along the mutational spectrum, corroborating their association with the mutations (Fig. 10e). While many TF targets switched expression rapidly with *MYCN* overexpression, others showed a continuous pattern with up-/down-regulation already detectable in 17q and 17q1q, e.g., targets of vagal and early NC regulators HOXB3 and CDX2[86] (up), or of sensory neurogenesis regulator NEUROD1[39] (down). To aid interpretation, we visualised cell-line-specific interactions between TFs and targets as edges in connected network diagrams

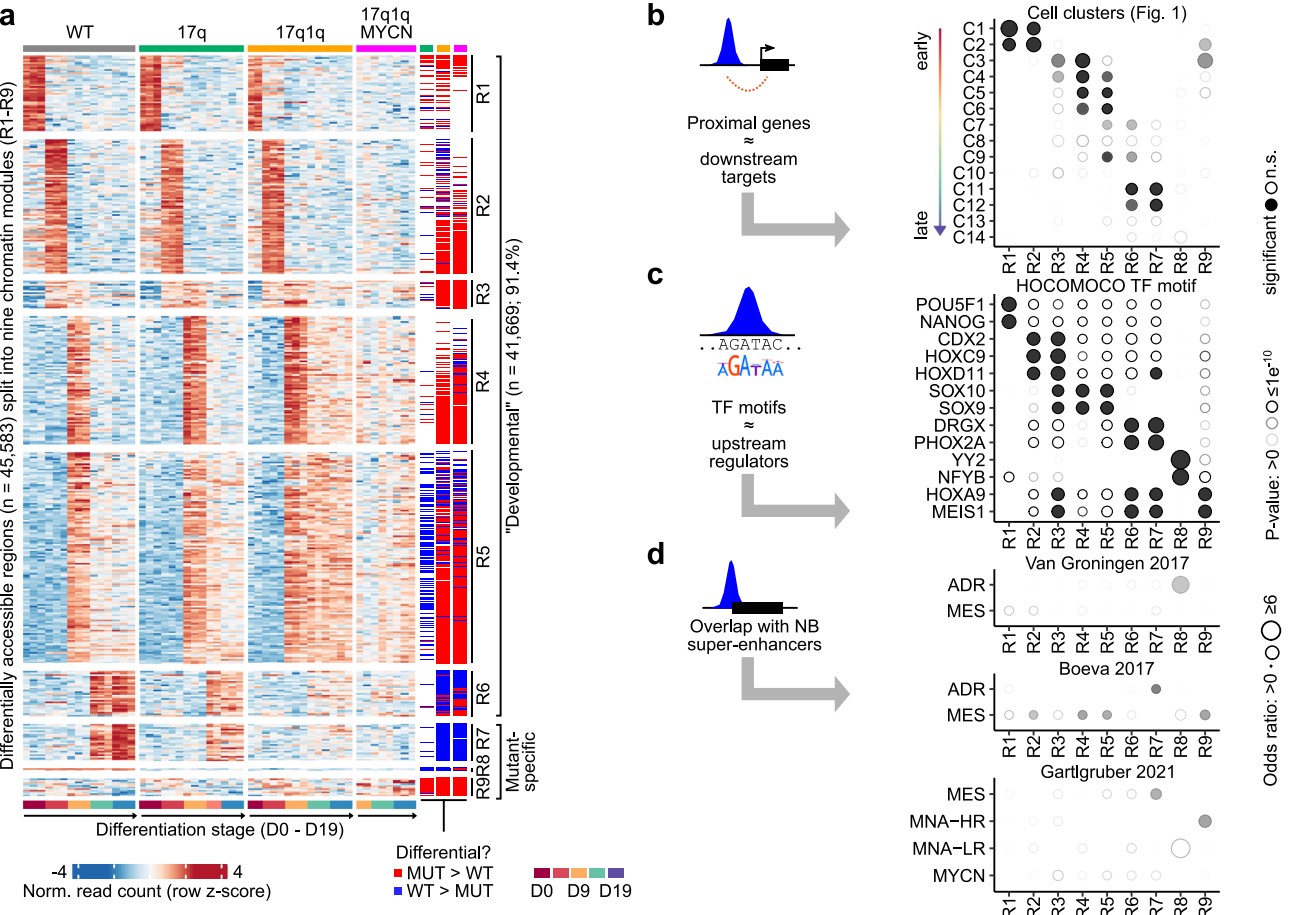

**Fig. 9 | Differentiation of wild-type and mutant hESCs is associated with changes in distinct chromatin modules. a** Heatmaps showing normalised read counts for all DARs (columns) in any pairwise comparison of two stages or conditions (*DEseq2*[147]; two-sided tests, *P* values adjusted for multiple hypothesis testing using the Benjamini-Hochberg method; $P_{adj} \leq 0.005$, |$\log_2$FoldChange| ≥ $\log_2$(1.5); $n_{total}$ = 45,583). Regions have been divided into nine non-overlapping modules (R1–R9) by hierarchical clustering. Three annotation columns are shown to the right indicating regions called down- (blue) and up-regulated (red) in each condition. All DARs are reported in Supplementary Data 12 and 13. **b** Comparisons of regions belonging to the nine chromatin modules (from **a**) and proximal cluster markers defined in our scRNA-seq analysis (cp. Fig. 1). An enrichment analysis was performed using hypergeometric tests (*hypeR*[135]; background: all genes associated with at least one ATAC-seq peak) and the size and transparency of circles indicate the odds ratio and *P* value, respectively. Significant results are indicated with filled circles (*P* values adjusted for multiple hypothesis testing using the Benjamini-Hochberg method;

$P_{adj} \leq 0.005$). All results are shown in the figure, and reported in Supplementary Data 14. **c** Enrichment analysis for overlaps between chromatin modules and known TF motifs (*HOCOMOCO* database[154], v11). Statistical tests and plots are as in (**b**), with the exception that only overlaps with $P_{adj} \leq 0.0000001$ and |$\log_2$FoldChange| ≥ $\log_2$(2) were marked as significant (background: all peaks with at least one motif match). The top results per module are shown and all results are reported in Supplementary Data 14. **d** Enrichment analysis of overlaps between regions belonging to the nine chromatin modules and super-enhancers specific to certain NB epigenetic subtypes[44,45,73] (background: all peaks with at least one overlapping region annotated in the super-enhancer analyses). Statistical tests and display as in (**b**). Source data are provided as a Source Data file. D0/9/19, day 0/9/19; WT, wild-type H7 hESCs; MUT a mutant hESC line (17q, 17q1q, or 17q1qMYCN); DAR, differentially accessible region; R1-R9, chromatin region modules; n.s., not significant; TF, transcription factor; NB, neuroblastoma; ADR, adrenergic; MES, mesenchymal; MNA-HR, not *MYCN*-amplified high-risk; MNA-LR, not *MYCN*-amplified low-risk.

(Fig. 10f; Supplementary Fig. 13c). These diagrams showcased the emergence of a new subnetwork of TFs in 17q1qMYCN that centred on MYCN and incorporated TFs like NR1D1 and TFAP4. In contrast, a subnetwork involving NC-related TFs such as SOX10 and TFAP2A/B was lost in these cells. Intriguingly, downregulation of TFs linked to sensory neuronal development (NEUROD1, ONECUT1) was visible already in 17q cells (Fig. 10f), perhaps explaining why sensory neuron-like derivatives were less abundant in 17q cultures (Fig. 2b). In 17q1q, we additionally observed upregulation of TFs related to early posterior NC specification including HOXB3, LEF1, and CDX2, which was partially reversed (HOXB3) upon *MYCN* overexpression (Fig. 10f). While many of the TFs implicated in these developmental gene-regulatory networks are weakly or not at all expressed in NB tumours (Supplementary Fig. 14a), we found that the targets of MYCN-related TFs (based on our analysis) are highly expressed in *MYCN*-amplified tumours (Supplementary Fig. 14b). Our analysis also revealed that the targets of 17q/

1q-related TFs are strongly expressed in groups of tumours, but we could not determine whether these contained the corresponding CNAs due incomplete annotations.

In summary, our data suggest a subtle rewiring of gene-regulatory networks in CNA-carrying hESCs, which may be linked to the depletion of mature sensory NC derivatives and increased early SCP signature found in our single-cell analyses (cp. Fig. 2). Overexpression of *MYCN* resulted in a switch in favour of known NC-linked TFs downstream of MYCN.

## Discussion

Although CNAs are a principal genetic hallmark of paediatric cancers, it has remained difficult to determine their exact role in tumour initiation due to the lack of suitable human models. In this study, we used hESCs carrying CNAs that are prevalent in NB (chr17q and chr1q gains). The NC is a transient embryonic tissue that is inaccessible after

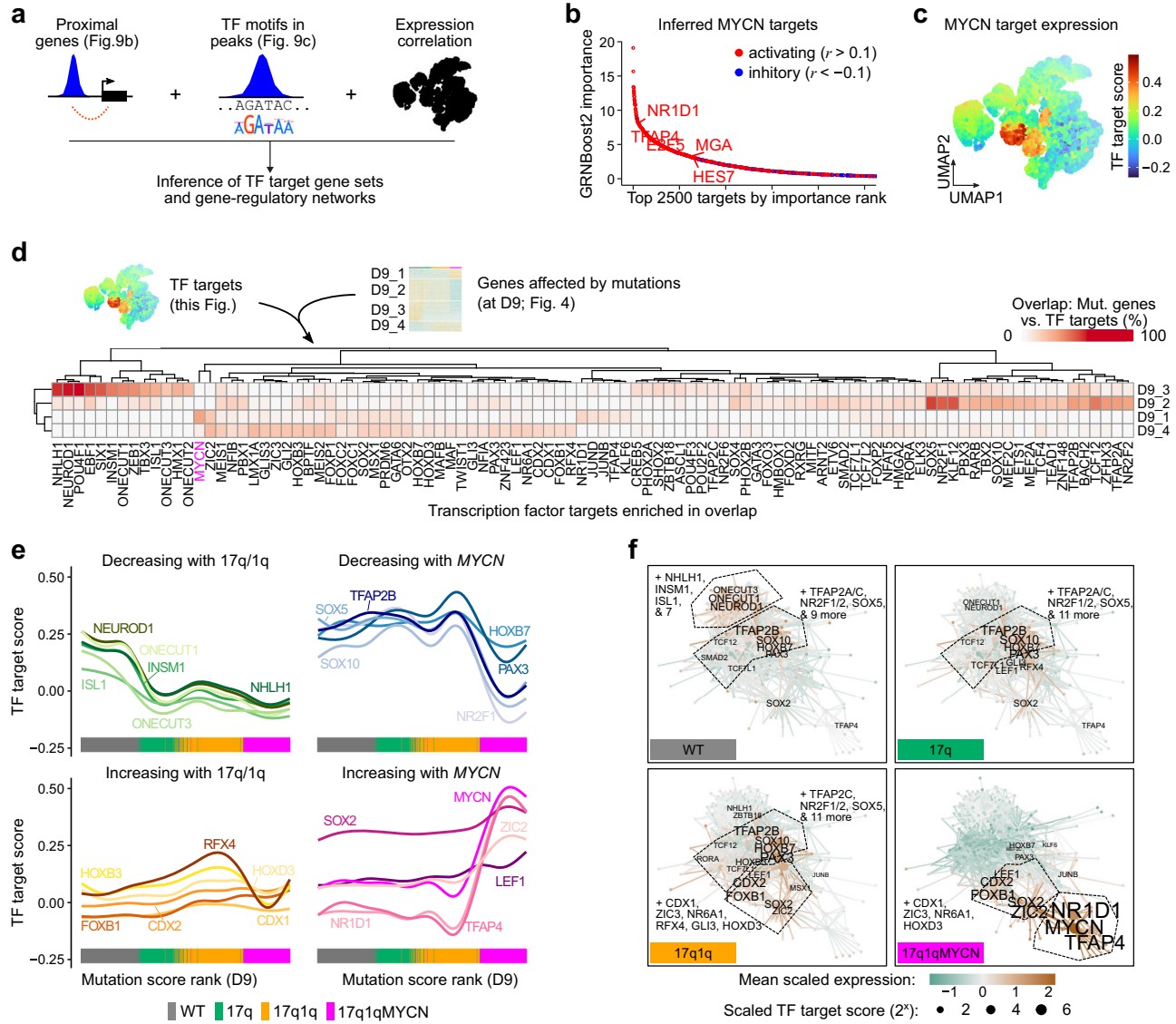

**Fig. 10 | Copy number changes facilitate *MYCN*-mediated blockage of differentiation via developmental transcription factor networks. a** To define putative target genes of TFs, we linked TF motif matches in ATAC-seq peaks with proximal genes and used *GRNboost2*[81] to identify highly correlated TF-to-target gene candidates based on our scRNA-seq data. **b** Top-2500 predicted targets of MYCN. Putative targets without support in our ATAC-seq data (motif in ≥1 peak near the gene) have been removed. The Pearson correlation coefficient (*r*) between each TF and target gene determined the direction of the putative interaction (*r* > 0.1 = activating, *r* < −0.1 = inhibitory, others = marginal). The top-5 TFs have been highlighted. All results are reported in Supplementary Data 15. **c** Average expression (Seurat module score) of the MYCN target genes (activated targets from **b**) in our integrated scRNA-seq dataset (cp. Fig. 4a). **d** Heatmap displaying the percentage of gene sets D9_1 to D9_4 (correlated with mutation score, cp. Fig. 4b, c) overlapping targets of the indicated TFs. TFs with significant overlaps in at least one comparison are shown (hypergeometric test, *hypeR*[135]; *P* values adjusted for

multiple hypothesis testing using the Benjamini-Hochberg method; $P_{adj} \leq 0.05$, | log2FoldChange| ≥ log2(4), frequency ≥5%). All results are also reported in Supplementary Data 16. **e** Smoothed line plots evaluating target gene expression (Seurat module score) with increasing mutation scores (from Fig. 4b, c). We curated selected TFs from (**d**) into groups of TFs losing or gaining activity. The cell line of each data point is indicated at the bottom. **f** Network diagrams visualising putative TF-to-target relations for enriched TF targets (cp. **c**–**e**). Each node represents a TF or target gene, and each edge is a TF-to-target link. We made these networks specific to each condition by using colour to indicate the mean scaled expression at D9 (edges coloured by source TF) and node size to indicate the mean scaled $T_{TF}$ target score of each TF. Only labels of TFs with positive scaled expression are shown and selected groups of TFs have been merged for visualisation. A diagram with all node labels is shown in Supplementary Fig. 13c. Source data are provided as a Source Data file. D9, day 9; TF, transcription factor; WT, wild-type H7 hESCs; r, Pearson correlation coefficient.

birth; therefore, hESC differentiation allowed us to experimentally study the effects of these mutations on human sympathoadrenal progenitors, the putative cells-of-origin of NB.

We provide a comprehensive knowledge base of transcriptomic and epigenetic changes in this model on a temporal (i.e., during differentiation) and a genetic (i.e., with different mutations) axis. Our data show that chr17q/1q gains impair trunk NC differentiation and potentiate an SCP-like gene signature. In this aberrant cell state, overexpression of *MYCN* (mimicking *MYCN* amplification commonly

found along with chr17q/chr1q gains in NB tumours) leads to a complete derailment of sympathoadrenal lineage specification, and a proliferative, tumour-like cellular phenotype that correlates with the emergence of NB-like tumours in vivo. Moreover, chr17q/1q gains were found to enhance the *MYCN*-driven differentiation block and acquisition of tumourigenic hallmarks such as proliferation, clonogenicity, and resistance to apoptosis. In line with recent studies[29,87], we speculate that CNAs provide an early selective advantage manifested by the expansion of undifferentiated cells, which act

subsequently as a NB-initiating entity upon a second oncogenic hit such as *MYCN* overexpression.

The accumulation of NB-associated lesions correlated with a failure to reprogram chromatin during trunk NC differentiation. Following gain of chr17q/1q, cells lost the activity of TFs associated with sensory differentiation (e.g., NEUROD1) and instead upregulated vagal NC TFs like HOXB3 and WNT-related effector LEF1[86,88]. *MYCN* overexpression on top of these CNAs abolished chromatin states indicative of sympathoadrenal differentiation, and instead led to the induction of targets of NR1D1, TFAP4, and other TFs of the reported NB regulatory circuitry[44,45,72–76]. TFAP4 is a well-established downstream effector of MYCN[84,85], and NR1D1 (Rev-erbα) is a circadian rhythm and metabolic regulator, and a downstream effector of MYCN hyperactivity in NB[82,83]. Together with the appearance of sensory-related signatures in NB tumours (C4* and C5*, Fig. 7) our "early *MYCN* onset" scenario reveals previously uncharted disruptions of the early sensory NC lineage, which might complement the currently prevailing model of dichotomic mesenchymal/adrenergic heterogeneity in NB[11,44–46,76,89–92]. Thus, our model will facilitate the functional dissection of these TFs via loss-/gain-of-function approaches to decipher their crosstalk with *MYCN*/CNA-driven tumourigenesis.

Complementing earlier studies using cell lines and animal models[12–14,18,19,22], recent single-cell transcriptomic analyses of NB tumours and metastases[15–17,23] corroborated an origin of NB from neuroblastic, SCP-like progenitors, and highlighted intra-tumour heterogeneity comprising subtypes of tumour cells with adrenergic and mesenchymal properties. In our in-vitro experiments, we also observed cells expressing signatures of both cell types, suggesting that our model could be useful to experimentally investigate the transition between these and other NB-relevant cell types, providing a new scope into their role in therapy resistance[89]. Furthermore, *MYCN* overexpression (in conjunction with chr17q/1q gains) in nascent trunk NC cells was sufficient to drive tumourigenic traits, suggesting that in some cases NB initiation might occur before SCP/neuroblast emergence and that acquisition of an SCP-like identity may also be a consequence of mutations in earlier stages rather than the origin. We also observed that *MYCN* induction resulted in an apparent block of differentiation when activated at other stages. In future, our cell models will provide the means to dissect the specific effects of *MYCN* at different time points and in specific cell types.

In this study we exploited the phenomenon of culture adaptation of hESCs[49], to obtain sets of cell lines with CNAs that are commonly observed in NB in an otherwise largely isogenic background. Our detailed genetic analyses of the used cell lines revealed other mutations that had naturally arisen in these cell lines (Supplementary Data 4), including a point mutation in the BCL6-interacting corepressor *BCOR* (*BCOR^L1673f*). *BCOR* mutations have been previously observed in human induced pluripotent stem cell cultures[93,94] and NB patients with *BCOR* mutations exhibit a high frequency of CNAs[87]. *BCOR* mutations have also been reported together with CNAs in other cancers, e.g., retinoblastoma[95]. It would be tempting to speculate that *BCOR* dysfunction might facilitate the ability of cells to tolerate the emergence of certain CNAs; however, to date a causal relationship remains to be established.

Our hESC-based model provides a tractable system for analysing tumour initiation events within disease-relevant human embryonic cell-like populations. In this study, we focused on cell-intrinsic transcriptional regulation since our cultures lack tumour-relevant, non-NC cell types (e.g., immune cells or Schwann cells) and do not recapitulate the structural and physical properties of the human tumour micro- and macroenvironment[96–99]. In the future, it will be possible to combine our system with 3D co-culture approaches with defined cell types or to use biomimetic scaffolds to emulate cell-cell interactions and extrinsic environmental influences.

In conclusion, this study unravels the developmental effects of NB-associated mutations and proposes the progressive corruption of gene-regulatory networks by CNAs as an early step toward tumour initiation by selection of undifferentiated progenitor phenotypes. Transformation is then triggered by a second hit with *MYCN* overexpression, which tilts cells toward increased proliferation and formation of aberrant cell types. Our data provide a direct link between CNAs that commonly emerge in hESC cultures with impaired differentiation and the acquisition of tumourigenic hallmarks, thus highlighting the importance of rigorous monitoring of such cultures prior to their use in disease modelling or cell therapy application in line with recent recommendations from the International Society for Stem Cell Research[49,100].

## Methods
### Human embryonic stem cell (hESC) cell culture and differentiation
**Cell lines and cell culture.** We employed H7 (https://hpscreg.eu/cell-line/WAe007-A) and H9 (https://hpscreg.eu/cell-line/WAe009-A) hESCs as karyotypically normal, female WT controls[38]. Use of human embryonic cells has been approved by the Human Embryonic Stem Cell UK Steering Committee (SCSC23-29). Their isogenic chr17q counterparts carry a gain in chromosome 17q (region q27q11) via an unbalanced translocation with chromosome 6 (H7) or a gain of 17q via an unbalanced translocation with chromosome 21 with breakpoints at 17q21 and 21p11.2 (H9)[50,101]. The chr17q1q hESC lines were clonally derived, after their spontaneous emergence following the genetic modification of chr17q hESCs. The H7 chr17q1q-MYCN hESC line was generated by introducing a TetOn-PiggyBac plasmid (PB-TRE3G-MYCN, plasmid#104542, Addgene) carrying the wild-type version of the *MYCN* gene[102] while the H9 chr17q1q-MYCN and H7 WT-MYCN and 17q-MYCN hESC lines were produced using a Tet-On "all-in-one" inducible expression cassette containing the TRE3G promoter driving the expression of *MYCN* with a 2A peptide-linked fluorescent reporter (mScarlet) and a pCAG promoter-driven rtTA3G transactivator[103,104]. Plasmids were introduced via nucleofection using either the Neon NxT Electroporation System (Thermo Fisher Scientific) or the Lonza 4D-Nucleofector System. In the case of the latter, the Amaxa 4D-Nucleofector Basic Protocol for Human Stem Cells was employed with the following modification: $2 \times 10^6$ cells were transfected with 2 µg plasmid in 100 µl Nucleocuvettes. All cell lines were tested regularly for mycoplasma and expression of pluripotency markers. Karyotypic analysis was carried out using G-banding (number of cells examined = 20–30). A rapid quantitative PCR (qPCR) assay was also regularly employed to detect the emergence of common CNAs such as chr17q and 1q gains in our hESC lines[105]. hESCs were cultured routinely in feeder-free conditions at 37 °C and 5% $CO_2$ in E8 media[106] complemented with GlutaMax (Cat# 35050061, Thermo Fisher Scientific) on Vitronectin (Cat# A14700, Thermo Fisher Scientific) or on Geltrex LDEV-Free Reduced Growth Factor Basement Membrane Matrix (Cat# A1413202, Thermo Fisher Scientific) as an attachment substrate. All hESC lines described in this manuscript are available upon request and completion of a Material Transfer Agreement.

**Differentiation toward trunk neural crest.** hESC differentiation toward trunk NC and its derivatives was performed using our established protocols[35,36]. Briefly, hESCs were harvested using StemPro Accutase Cell Dissociation Reagent (Cat# A1110501, Thermo Fisher Scientific) and plated at 60,000 cells/cm² in N2B27 medium supplemented with 20 ng/ml of FGF2 (Cat# 233-FB/CF, R&D) and 4 µM of CHIR 99021 (Cat# 4423, Tocris) and 10 µM of Rock Inhibitor (Y-27632) (Cat# A11001, Generon) in a volume of 300 µl/cm². The N2B27 medium consisted of 50:50 DMEM F12 Merck Life Science/Neurobasal medium (Gibco) and 1x N2 supplement (Cat# 17502048, Invitrogen), 1x B27 (Cat#17504044, Invitrogen), 1x GlutaMAX (Cat# 35050061, Thermo Fisher Scientific), 1x MEM Non-essential amino acids (NEAA; Cat#11140050, Thermo Fisher Scientific), 50 µM 2-Mercaptoethanol

(Cat# 31350010, Thermo Fisher Scientific). After 24 h, media was refreshed removing the Rock Inhibitor and cells were cultured for a further 2 days in FGF2/CHIR to generate NMPs (300 µl/cm²). NMPs at D3 were then re-plated at 50,000 cells/cm² (H7) or 40,000 cells/cm² (H9) in neural crest inducing medium consisting of DMEM/F12, 1x N2 supplement, 1x GlutaMAX, 1x MEM NEAA, the TGF-beta/Activin/Nodal inhibitor SB-431542 (2 µM, Cat# 1614, Tocris), CHIR99021 (1 µM, Cat# 4423, Tocris), BMP4 (15 ng/ml, Cat# PHC9534, Thermo Fisher Scientific), the BMP type-I receptor inhibitor DMH-1 (1 µM, Cat# 4126, Tocris), 10 µM of Rock Inhibitor (Y-27632) on Geltrex LDEV-Free Reduced Growth Factor Basement Membrane Matrix (Cat# A1413202, Thermo Fisher Scientific) in a volume of 300 µl/cm². 48 h later (D5), media was replaced removing the Rock Inhibitor. Media was refreshed at D7 and D8 increasing volume to 500 µl/cm². On D5, the expression of MYCN was induced by supplementing the neural crest media with 100 ng/ml (H7-17q1q-MYCN), 200 ng/ml (H7 WT-MYCN, 17q-MYCN), or 1000 ng/ml (H9-derived lines) of Doxycycline (Cat# D3447, Merck). On D9, cells were re-plated at 150,000–250,000 cells/cm² in plates coated with Geltrex (Thermo Fisher Scientific) in the presence of medium containing BrainPhys (Cat# 05790, Stem Cell Technologies), 1x B27 supplement (Cat# 17504044, Invitrogen), 1x N2 supplement (Cat# 17502048, Invitrogen), 1x MEM NEAA (Cat# 11140050, Thermo Fisher Scientific) and 1x Glutamax (Cat# 35050061, Thermo Fisher Scientific), BMP4 (50 ng/ml, Cat# PHC9534, Thermo Fisher Scientific), recombinant SHH (C24II) (50 ng/ml, Cat# 1845-SH-025, R and D) and purmorphamine (1.5 µM, Cat# SML0868, Sigma) and cultured for 5 days (=D14 of differentiation) in a volume of 250 µl/cm². Media was refreshed daily. For further sympathetic neuron differentiation, D14 cells were switched into a medium containing BrainPhys neuronal medium (Stem Cell Technologies), 1x B27 supplement (Invitrogen), 1x N2 supplement (Invitrogen), 1x NEAA (Thermo Fisher Scientific) and 1x Glutamax (Thermo Fisher Scientific), NGF (10 ng/ml, Cat#450-01 Peprotech), BDNF (10 ng/ml, Cat# 450-02, Peprotech) and GDNF (10 ng/ml, Cat# 450-10, Peprotech) for a further 5–14 days (volume of 300 µl/cm² changing media every other day). Volume was increased up to 500 µl/cm², depending on cell density, after day 17 of differentiation.

**Immunostaining.** Cells were fixed using 4% PFA (P6148, Sigma-Aldrich) at room temperature for 10 min, then washed twice with PBS (without Ca²⁺, Mg²⁺) to remove any traces of PFA and permeabilised using a PBS supplemented with 10% FCS, 0.1% BSA and 0.5% Triton X-100 for 10 min. Cells were then incubated in blocking buffer (PBS supplemented with 10% FCS and 0.1% BSA) for 1 h at RT or overnight at 4 °C. Primary and secondary antibodies were diluted in the blocking buffer; the former were left overnight at 4 °C and the latter for 2 h at 4 °C on an orbital shaker. Samples were washed twice with blocking buffer between the primary and secondary antibodies. Hoechst 33342 (H3570, Invitrogen) was added at a ratio of 1:1000 to the secondary antibodies' mixture to label nuclei in the cells. We used the following primary antibodies SOX10 (D5V9L) (Cell Signalling, 89356S, 1:500); HOXC9 (Abcam, Ab50839,1:50); MYCN (Santa Cruz, SC-53993, 1:100); PHOX2B (Santa Cruz, SC-376997, 1:100); MASH1 (ASCL1) (Abcam, Ab211327, 1:100); Ki67 (Abcam, Ab238020, 1:100); PERIPHERIN (Sigma-Aldrich, AB1530, 1:400); Cleaved Caspase 3 (Asp175) (Cell Signalling, 9661S, 1:400), yH2AX (Cell Signalling, S139/9718S, 1:400). Secondary antibodies: Goat anti-Mouse Affinipure IgG+IgM (H + L) AlexaFluor 647 (Stratech (Jackson ImmunoResearch) 115-605-044-JIR, Polyclonal 1:500); Donkey anti-Rabbit IgG (H + L) Alexa Fluor 488 (Invitrogen, A-21206, 1:1000).

**Intracellular flow cytometry staining.** Cells were detached and resuspended as single cells using StemPro Accutase Cell Dissociation Reagent (Cat# A1110501, Thermo Fisher Scientific) and then counted. Next, 10 million cells/ml were resuspended in 4% PFA at room temperature for 10 min. Then cells were washed once with PBS (without Ca²⁺, Mg²⁺) and pelleted at 200 g. Cells were resuspended in PBS at 10

million/ml and used for antibody staining. Permeabilisation buffer (0.5% Triton X-100 in PBS with 10% FCS and 0.1% BSA) was added to each sample, followed by incubation at room temperature for 10 min. Samples were then washed once with staining buffer (PBS with 10% FCS and 0.1% BSA) and pelleted at 200 g. Then samples were resuspended in staining buffer containing pre-diluted primary antibodies: SOX10 (D5V9L) (1:500; 89356S, Cell Signalling); HOXC9 (1:50; Ab50839, Abcam); cleaved Caspase 3 (Asp175) (Cell Signalling, 9661S, 1:400). The samples were left at 4 °C on an orbital shaker overnight. Then, the primary antibodies were removed, and samples were washed two times with staining buffer. After washing, staining buffer with pre-diluted secondary antibody was added to the samples and incubated at 4 °C for 2 h. The secondary antibodies used were Goat anti-Mouse Affinipure IgG+IgM (H + L) AlexaFluor 647 (Stratech (Jackson ImmunoResearch) 115-605-044-JIR, Polyclonal 1:500); Donkey anti-Rabbit IgG (H + L) Alexa Fluor 488 (Invitrogen, A-21206, 1:1000). Finally, samples were washed once with staining buffer, resuspended in staining buffer and analysed using a BD FACSJazz or a CytoFLEX (Beckman Coulter) flow cytometer. A secondary antibody-only sample was used as a control to set the gating.

**Cell cycle analysis.** The 5-ethynyl-2′-deoxyuridine (EdU) assay was performed following the manufacturer's instructions (Thermo Fisher Scientific, C10633 Alexa Fluor 488). We used 10 µM of Edu for a 2-h incubation. Cells were analysed in the flow cytometer (BD FACSJazz) using the 405 nm laser to detect the Hoechst/DAPI staining and 488 nm to detect the EdU staining.

**Low-density plating.** Day 9 trunk NC cells derived from hESCs as described above were harvested and plated at a density of 500 cells/cm² in plates pre-coated with Geltrex LDEV-Free Reduced Growth Factor Basement Membrane Matrix (Cat# A1413202, Thermo Fisher Scientific) in the presence of DMEM/F12 (Sigma-Aldrich), 1x N2 supplement, 1x GlutaMAX, 1x MEM NEAA, the TGF-beta/Activin/Nodal inhibitor SB-431542 (2 µM, Tocris), CHIR99021 (1 µM, Tocris), BMP4 (15 ng/ml, Thermo Fisher Scientific), the BMP type-I receptor inhibitor DMH-1 (1 µM, Tocris) and ROCK inhibitor Y-27632 2HCl (10 µM) (300 µl/cm²). The culture medium was replaced the following day with medium containing BrainPhys (Stem Cell Technologies), 1x B27 supplement (Invitrogen), 1x N2 supplement (Invitrogen), 1x NEAA (Thermo Fisher Scientific) and 1x Glutamax (Thermo Fisher Scientific), BMP4 (50 ng/ml, Thermo Fisher Scientific), recombinant SHH (C24II) (50 ng/ml, R and D) and Purmorphamine (1.5 µM, Sigma) (250 µl/cm²). Plates were then incubated at 37 °C at 5% CO₂. The media was refreshed every 48 h. After 5 days of culture, cells were fixed (PFA 4%/10 min) and stained with Hoechst 33342 (Cat# H3570, Invitrogen) for 5 min. Colonies were detected using an InCell Analyser 2200 (GE Healthcare) at a 4X magnification. Images were processed using Cell Profiler.

**DNA damage analysis.** DNA damage was measured by assessing the phosphorylation state of the histone H2AX on the SerCells were fixed and immunostained using the anti-yH2AX as described above at different time points. Stained cells were imaged using the InCell Analyser 2200 (GE Healthcare) at 40X magnification. Image analysis was performed using a pipeline in CellProfiler that allowed us to detect the number of foci of yH2AX antibody per nuclei.

**Quantitative real-time PCR.** RNA extractions were performed using the total RNA purification kit (Norgen Biotek, 17200) according to the manufacturer's instructions. cDNA synthesis was performed using the High-Capacity cDNA Reverse Transcription kit (ThermoFisher, 4368814). Quantitative real-time PCR was performed using PowerUp SYBR master mix (ThermoFisher, A25780) and run on a QuantStudio 12 K Flex (Applied Biosystems). Primers used for *MYCN*: cca-caaggccctcagtacc (forward), tcttcctcttcatcatcttcatca (reverse).

## Mouse experiments

**Cell preparation for xenotransplantation.** H7 wild type, 17q1q, and 17q1qMYCN hESCs were differentiated up to day 9 following the protocol described above. Cells were harvested using Accutase to create a single cell suspension, counted, and resuspended with media containing Matrigel before injection.

**Mice and in-vivo experiments.** All animal experiments were approved by The Institute of Cancer Research Animal Welfare and Ethical Review Body and performed in accordance with the UK Home Office Animals (Scientific Procedures) Act 1986, the UK National Cancer Research Institute guidelines for the welfare of animals in cancer research and the ARRIVE (animal research: reporting in-vivo experiments) guidelines. Female NSG mice were obtained from Charles River and enrolled into trial at 6–8 weeks of age. Mice were maintained on a regular diet in a pathogen-free facility on a 12 h light/dark cycle with unlimited access to food and water.

**Subcutaneous xenograft.** One million cells with 50% Matrigel were injected subcutaneously into the right flank of NSG mice (female; 6–8 weeks old) and allowed to establish a murine xenograft model. Studies were terminated when the mean diameter of the tumour reached 15 mm. Tumour volumes were measured by Vernier caliper across two perpendicular diameters, and volumes were calculated according to the formula $V = 4/3\pi \, [(d1 + d2)/4]3$; where d1 and d2 were the two perpendicular diameters. The weight of the mice was measured every 2 days. Mice were fed with either regular diet or DOX diet (chow containing 20 g of DOX per kg of diet) to induce the expression of *MYCN*.

**Orthotopic (adrenal)xenograft.** 100,000 cells with 50% Matrigel were injected into the right adrenal gland of NSG mice (female; 6–8 weeks old) and allowed to establish a murine xenograft model. Detection of xenografted tumours was performed by magnetic resonance imaging (MRI). The weight of the mice was measured every 2 days. Mice were fed with either standard diet or DOX diet (chow containing 20 g of DOX per kg of diet) to induce the expression of *MYCN*. Magnetic resonance images were acquired on a 1 Tesla M3 small animal MRI scanner (Aspect Imaging). Mice were anesthetised using isoflurane delivered via oxygen gas and their core temperature was maintained at 37 °C. Anatomical T2-weighted coronal images were acquired through the mouse abdomen, from which tumour volumes were determined using segmentation from regions of interest (ROI) drawn on each tumour-containing slice using the Horos medical image viewer.

**Pathology.** Tissue sections were stained with haematoxylin and eosin (H&E) or specific antibodies (MYCN, Merck; Ki67, BD Pharmingen). Immunohistochemistry was performed using standard methods. Briefly, 5 μm sections were stained with antibodies, including heat-induced epitope retrieval of specimens using citrate buffer (pH 6) or EDTA buffer.

## Zebrafish experiments

**Cell preparation for xenotransplantation.** Pre-differentiated neural crest cells were frozen on D7 during their in-vitro differentiation as described above, shipped, and subsequently thawed in DMEM at room temperature. All cells were retrieved in complete neural crest media as described above and plated onto Geltrex-coated wells in the presence of Rock inhibitor (50 μM) for 24 h. 17q1q cells were additionally treated with DOX (100 ng/ml) to induce *MYCN* expression. On D8, media were refreshed, and respective DOX treatment was continued but Rock inhibitor was discontinued. On D9, cells were collected for xenografting experiments and labeled with CellTrace™ Violet (Invitrogen, Thermo Fisher Scientific) for imaging. For this, cells were harvested with Accutase (PAN-Biotech) and resuspended at a concentration of

$1*10^6$ cells/ml in PBS. CellTrace™ Violet was added to a final concentration of 5 μM for an incubation time of 10 min at 37 °C in the dark. The cell-staining mixture was filled up with 5 volumes of DMEM supplemented with 10% FBS and the suspension was incubated for 5 min. After gentle centrifugation (5 min, 500 g, 4 °C) the collected cells were resuspended in fresh DMEM medium supplemented with 10% FBS and incubated at 37 °C for 10 min. Adhering/ clumping cells were separated via a 35 μm cell strainer. The cell number was adjusted to a concentration of 100 cells/nl in PBS. The freshly stained cells were kept on ice until transplantation. SK-N-BE2C-H2B-GFP cells[69] (a kind gift of F. Westermann) were cultured in RPMI 1640 medium with GlutaMAX™ (Cat# 61870044, Thermo Fisher Scientific) supplemented with 10% (v/v) fetal bovine serum (Cat# F7524500ML, Sigma), 80 units/ml penicillin, 80 μg/ml streptomycin (Cat# 15140122, Thermo Fisher Scientific), 1 nM sodium pyruvate (Cat# P0443100, PAN-Biotech), 25 mM Hepes buffer (PAN-Biotech) and 8 μl/ml G418. For zebrafish xenotransplantations, the GFP-labelled cells were harvested and resuspended in PBS at a density of $10^5$/μl as described above.

**Zebrafish strains, husbandry, and xenotransplantation.** Zebrafish (*Danio rerio*) were reared under standard conditions in a 14 h/10 h light cycle according to the guidelines of the local authorities (Magistratsabteilung MA58 of the municipal administration of Vienna, Austria) under licenses GZ:565304-2014-6 and GZ:534619-2014-4. For xenotransplantation experiments, the pigment mutant strain mitfa^b692/b692^; ednrba^b140/b140^ was used. mitfa^b692/b692^; ednrba^b140/b140^ embryos raised at 28 °C were anaesthetised with Tricaine (0.16 g/l Tricaine (Cat# E1052110G, Sigma-Aldrich), adjusted to pH 7 with 1 M Tris pH 9.5, in E3) and xenotransplanted at 2 days post fertilisation (dpf)[107]. For xenotransplantation, a micromanipulator (Cat# M3301R, World Precision Instruments) holding a borosilicate glass capillary (Cat# GB100T-8P, without filament, Science Products) connected to a microinjector (FemtoJet 4i, Eppendorf) was used. Transplantation capillaries were pulled with a needle puller (P-97, Sutter Instruments) and loaded with approximately 5 μl of tumour cell suspension. Cells were injected into the perivitelline space (PVS) of larvae. Visual inspection was carried out at 2 h post-injection on an Axio Zoom.V16 fluorescence microscope (Zeiss, Jena) and only correctly injected larvae were used in subsequent experiments and further maintained at 34 °C.

**Automated imaging and quantification.** One day post injection (1dpi) and 3dpi xenografted larvae were anaesthetised in 1x Tricaine and embedded in a 96-well ZF plate (Hashimoto Electronic Industry) with 0.5% ultra-low gelling agarose (Cat# A2576-25G, Sigma-Aldrich) for automated imaging on a high-content imager (Operetta CLS, PerkinElmer). Images were acquired with a 5x air objective. Exposure times for brightfield images was 40 ms at 10% LED power. CellTrace Violet was recorded with an excitation of 390–420 nm at 100% LED power and detection at 430–500 nm using an exposure time of 600 ms. GFP was excited with 460–490 nm and detected at 500–550 nm with an exposure time of 400 ms. 23 planes with a distance of 25 μm were imaged per field of view of the laterally orientated larvae to cover the whole tumour. Tumour size was quantified with Harmony Software 4.9 (PerkinElmer).

## Ethical use of data

This study did not generate any new genomics data from patients. However, we performed re-analyses of previously published (sc)RNA-seq and SNP-array data that was previously collected at our institutions. The collection and research use of human tumour specimen was performed according to the guidelines of the Council for International Organisations of Medical Sciences (CIOMS) and World Health Organisation (WHO) and has been approved by the ethics board of the Medical University of Vienna (Ethikkommission Medizinische Universität Wien;

EK2281/2016, 1216/2018, 1754/2022). Informed consent has been obtained from all patients or parents/guardians/legally authorised representatives. The age-adapted informed consent for the CCRI Biobank covers the use of left over materials from medically necessary surgery or biopsy, which after completion of routine diagnostic procedures is biobanked (EK1853/2016) and available for research purposes, including genetic analysis, that are further specified in EK1216/2018 and EK1754/2022: to conduct genetic and transcriptomic analysis and link to clinical data for survival analysis. All data obtained from external sources derived from studies where informed consent was given for broad research use.

## Whole-exome sequencing

**Library generation and sequencing.** Genomic DNA (gDNA) from cell lines was isolated using a desalting method and library preparation was performed with 100 ng gDNA and the Enzymatic Fragmentation (undifferentiated hESC lines; Supplementary Fig. 4b, c) or Enzymatic Fragmentation 2.0 (cells after differentiation; Fig. 5e, Supplementary Fig. 9f, g) kit, and Twist Universal Adapter System (Twist). For whole-exome sequencing, the libraries were pooled and enriched with the Exome v1.3 and RefSeq (Human Comprehensive Exome) spike-in capture probes (Twist) according to the manufacturer's protocols. Libraries were quantified (Qubit 4 Fluorometer) and quality-checked on 4200 TapeStation and 2100 Bioanalyzer automated electrophoresis instruments (Agilent) and diluted before sequencing by the Biomedical Sequencing Facility at CeMM on an Illumina NovaSeq SP flowcell in 2x100bp paired-end mode (median coverage 87.2; Supplementary Data 1).

**Variant identification and annotation.** Raw reads were processed using the nf-core *sarek*[108,109] WES pipeline version 2.7.2. Variant calling was performed in a tumour-normal matched mode, with the parental H7 line serving as the matched normal sample. Three variant callers, *Mutect2*, *Strelka*, and *Manta*[110–112], were employed for comprehensive variant identification. Resulting VCF files from *Mutect2* and *Strelka* were normalised using *bcftools* norm (v1.9)[113] and subsequently annotated using the Ensembl *Variant Effect Predictor* (VEP; v99.2)[114]. The identified variants were filtered based on the default quality control measures implemented in each tool (*FILTER* column in the VCF contains "PASS"). To identify biologically relevant variants a filtering strategy was applied that was partly inspired by MAPPYACTS[115]: (i) exclude variants for which GERMQ or STRQ Phred-scaled values are <30; (ii) exclude variants with a population allele frequency of over 0.1% (in 1000 Genomes or gnomAD); (iii) only include variants that have any of "coding_sequence_variant", "frameshift_variant", "incomplete_terminal_codon", "inframe_deletion", "inframe_insertion", "missense_variant", "protein_altering_variant", "start_lost", "stop_gained", "stop_lost" as Consequence; (iv) only include variants that have any of IMPACT == HIGH, SIFT == "deleterious", PolyPhen == probably_damaging or damaging[116,117]; (v) exclude variants that have a variant allele frequency <= 5%.

**Copy number calling.** CNAs were called by Sequenza (version 3.0.0)[118]. GC content was calculated for hg38 using *sequenza-utils gc_wiggle*. Depth ratio and B-allele frequency information was calculated using *bam2seqz* for each non-parental cell line using the parental cell line as a normal reference, single nucleotide polymorphisms (SNPs) were considered heterozygous if the allele frequency was in the range 0.4 to 0.6 (--*het* 0.4). Data was then binned using the *seqz_binning* command. Autosomes and the X chromosome were then extracted using Sequenza (*sequenza.extract*) and, as the cell lines are not contaminated with normal cells as is common place in tumour tissue samples, cellularity was tested in a range of 1 to 1.0002 to ensure a pure solution was produced by Sequenza. Copy number profiles were then plotted using ComplexHeatmaps[119]. Breakpoints were

considered telomeric if they were within 1Mbp of the beginning or end of the chromosome. Aberrant segments with 100 or more B-allele frequency observations (N.BAF) were considered to be confidently supported and are reported in Supplementary Data 5.

**Phylogenetic analysis.** Mutations called by Mutect2[120] with the PASS filter and of VARIANT_CLASS SNV as annotated by VEP[114] that overlapped with the exome target panel without padding were used for phylogenetic analysis. Mutations were required to have a minimum variant allele frequency (VAF) of 0.2 to ensure only high frequency clonal mutations were included in the phylogeny. Phylogenetic trees were constructed using parsimony and the *phangorn* R package[121]. The parsimony ratchet method (*pratchet*) was used to search for the best tree and the tree was rooted on the parental cell line. Branch lengths were calculated using the *acctran* function. Distance between tree tips was calculated using the *distTips* function in the *adephylo* R package[122]. Phylogenetic trees were plotted using *ggtree*[123].

**Pre-processing and analysis of NB SNP-array data.** SNP-array data from tumour or bone marrow obtained at diagnosis from Austrian cases with INSS stage 4 high-risk NB[51] were re-analysed for chr17 and chr1 CNAs using VARAN-GIE (v0.2.9), yielding 88 samples with CNAs (>10 kb) on at least one of these chromosomes. Genomic segments were manually curated and plotted using *ggplot2* (v3.3.5). The available CNA annotations based on the human genome reference hg19. Because of this, the breakpoint annotations for our own cell lines have been brought from hg38 to hg19 using *liftOver* from the R package *rtracklayer* (v1.54.0).

## Single-cell RNA sequencing (10x Genomics)

**Library generation and sequencing.** Single-cell suspensions were barcoded using oligo-conjugated lipids following the MULTI-seq workflow and frozen live[124] for G1-G13 (note, G2 was removed due to a technical failure), or frozen live and barcoded after thawing using the CELLPLEX (10x Genomics) workflow for G14-G27. After thawing cells were stained with DAPI. A maximum of 10,000 live cells per sample were sorted with a FACS-Aria v3 and pooled in sets of 3 or 4 samples by differentiation stage (from 3 to 5 independent replicate differentiation experiments). Each pooled group was processed using the 10X Genomics Single Cell 3' v3.1 workflow following the manufacturer's instructions. Enriched barcode libraries were indexed following the MULTI-seq workflow[124]. After quality control, libraries were sequenced on the Illumina NovaSeq S4 (G1-13) or S2 (G14-27) platform in 2 × 150 bp paired-end mode. Supplementary Data 1 includes an overview of sequencing data and performance metrics.

**Raw data processing and alignment.** Raw sequencing data were processed with the *CellRanger multi* v7.1.0 software (10x Genomics) for cell-level demultiplexing and alignment to the human reference transcriptome (*refdata-gex-GRCh38-2020-A* assembly provided by 10x Genomics). Following initial data processing, all subsequent analyses were performed in R (v4.1.3) using Bioconductor packages and the *Seurat*[125–127] (v4.1.0) package.

**Default basic processing.** We applied processing of scRNA-seq data in many instances across this manuscript. Unless parameters are otherwise specified, the default processing of scRNA-seq counts involved the following steps. Counts were normalised for read depth using Seurat's *SCTransform*[128] v0.3.3 (parameters: *method = "glmGamPoi"*; *variable.features.n = 5000*), followed by *RunPCA* (keeping the top 50 components), and inference of cell neighbourhoods by *FindNeighbors* on the PCA reduction. Finally, Uniform Manifold Approximation and Projection (UMAP) was performed using Seurat's *RunUMAP* function with default parameters. Clustering was performed using FindClusters.

**Quality control.** For each dataset, we first assessed technical covariates and characteristic expression profiles separately. We kept cells with less than 15% mitochondrial UMI counts, and at least 1000 detected genes and applied basic scRNA-seq processing and clustering of the cells (*SCTransform*[128] v0.3.3, parameters: *method = "glmGamPoi"*). Cell cycle scoring was calculated as recommended by Seurat and added as a variable to regress in SCTransform (*vars.to.regress = "ccvar"*). We used clusters devoid of markers and/or characterised by abnormally high mitochondrial expression, to derive a library-specific UMI count and mitochondrial percentage threshold for high-quality cells (thresholds for counts/mitochondrial percentage: G1: 3162/10%; G3: 10000/7.5%; G4: 10000/8%; G5: 3162/10%; G6: 10000/8%; G7: 12589/8%; G8: 7943/10%; G9: 7079/10%; G10: 3981/7.5%; G11: 3981/10%; G12: 5012/10%; G13: 10000/10%; G14: 5500/13%; G15: 3500/5%; G16: 3000/8%; G17: 2000/8%; G18: 3500/10%; G19: 1800/6%; G20: 3000/15%; G21: 6000/8%; G22: 5000/6%; G23: 3000/6%; G24: 1500/6%; G25: 3500/5%; G26: 2000/10%; G27: 3000/15%). In addition, empty and doublet droplets were flagged with *Emptydrops*[129] (v1.14.2; default parameters) and *scDblFinder*[130] (v1.8.0; parameters: *dbr = 0.01*), respectively. We retained only cells with *Emptydrops* FDR > 0.05 and individual scDblscore cutoffs for the datasets were: G1: 0.01; G3: 0.016; G4: 0.005; G5: 0.005; G6: 0.003; G7: 0.005; G8: 0.005; G9: 0.005; G10: 0.005; G11: 0.005; G12: 0.005; G13: 0.005; G14: 0.005; G15: 0.005; G16: 0.0075; G17: 0.002; G18: 0.007; G19: 0.00375; G20: 0.01; G21: 0.007; G22: 0.007; G23: 0.0125; G24: 0.003; G25: 0.007; G26: 0.005; G27: 0.007.

**Sample demultiplexing.** To demultiplex cells belonging to different pooled samples, we used deMULTIplex2[131] (v1.0.1) with default parameters on each dataset using the tag counts from *CellRanger multi*. All non-singlet cells were filtered out from the dataset.

**Normalisation, clustering, and marker gene analysis for the main dataset.** Raw UMI counts were normalised using Seurat's *SCTransform*[128] (parameters: *variable.features.n = 5000, method = "glmGamPoi", vars.to.regress = "ccvar"*) to account for differences in sequencing depth and cell cycle phase (the variable "ccvar" variable was calculated as the difference of S and G2/M scores using Seurat's *CellCycleScoring* method with default parameters). To integrate data from 3 to 5 independent differentiation experiments (replicates; Supplementary Data 1), we used *scVI*[132] (v0.20.3; parameters: *n_epochs = 50*) using 5000 highly variable features of the input data with Python 3.11 via reticulate (v1.24). Nearest neighbours were identified using Seurat's *FindNeighbors* function (parameters: *k = 30*) on the ten scVI components. The same scVI reduction was used to find a low dimensionality UMAP projection using Seurat's RunUMAP for both the WT-only (*n.neighbours = 50, min.dist = 0.5, dims = 1:8*) and full dataset (*n.neighbours: 30, min.dist = 0.4, dims = 1:8 method = "umap-learn", metric = "correlation"*). Clusters on the UMAP projection were defined using Seurat's *FindClusters* (parameters [full dataset]: *resolution = 0.6*, parameters [WT-only]: *resolution = 0.4, algorithm = 4*). Neighbouring clusters that shared functional markers were merged manually and relabelled to roughly reflect differentiation order. Finally, markers for each cluster were identified using the *FindAllMarkers2* function (*DElegate*[133] v1.1.0; parameters: *method = "deseq", min_fc = 1, min_rate = 0.5, replicate_column = "replicate"*), with each cluster compared to all the other cells in the dataset. Genes with an adjusted *P* value less than 0.05 were selected as markers. (Supplementary Data 2, 8). To compare mutant and wild-type cells, we filtered the integrated dataset to cells from D9 and identified pairwise DEGs ($P_{adj} \leq 0.05$, $|\log_2 FoldChange| > 0.25$) between each mutant condition and WT using the *findDE* function (Delegate v1.1.0; parameters: *group_column = "condition", method = "deseq", replicate_column = "day_rep"*). We discarded DEGs that were not expressed in at least 20% of cells on one side of the comparison. Up- and down-regulated DEGs on chr1q, on chr17q, and outside either CNA were then

tested separately to identify significant overlaps with MSigDB HALLMARK[134] gene sets using the hypergeometric test implemented in the *hypeR*[135] package (v1.10.0). DEGs and enriched pathways are listed in Supplementary Data 6 and Supplementary Data 7.

**Pseudotime trajectory analysis.** Pseudotime trajectories were inferred using *Slingshot*[136] (v2.2.0; default parameters) using a filtered dataset comprising only MES-SYM clusters C11, C12, C13, and C14 (cp. Fig. 1d; Supplementary Fig. 3). The filtered dataset was reprocessed using the basic scRNA-seq processing workflow as described above and the first two principal components were used to find trajectories between two extreme clusters. Only one trajectory was found. Genes whose expression was associated with the trajectories were identified with the generalised additive model and association test as implemented in *tradeSeq*[137] (v1.8.0; parameters: *knots = 5*). The top genes with the highest Wald statistic were selected for reporting (Supplementary Data 3). Transcription factors were identified based on the human transcription factors database[138] in Supplementary Fig. 3b.

**Cross-dataset annotation, label transfer, and signature scores.** To map data between scRNA-seq datasets, we employed Seurat's label transfer workflow. Both query and reference datasets were processed using the default basic scRNA-seq processing workflow as described above and mapped (*FindTransferAnchors*, *TransferData*, *IntegrateEmbeddings*, *NNTransform*, and *MappingScore* functions; default parameters) using the 50 first principal components of the PCA reduction from both datasets. To visualise cell mappings, we used "glasswork plots" (see below). In this study, the following mappings were performed with the same processing and parameters:

1. Human foetal adrenal reference datasets[15,16] mapped onto WT-only (Fig. 1d–f; Supplementary Figs. 2i, j, 3c) and full in-vitro (Figs. 2d–f, 4a; Supplementary Fig. 6g) scRNA-seq references. Upon obtaining consistent results for both (Supplementary Fig. 2j), the reference provided by Kameneva et al. was used throughout the analysis because of the curated cell type markers they provided (Supplementary Fig. 2i). These gene signatures were also quantified with Seurat's *AddModuleScore* function (default parameters) in Figs. 1e, f, 2f.
2. Our mutant scRNA-seq data mapped onto the wild-type reference (Fig. 2b, c).
3. NB tumour scRNA-seq data mapped onto our WT-only reference (Fig. 7b, c; Supplementary Fig. 11). See additional details about these datasets and processing in the section *"Pre-processing and mapping of NB tumour data"* below.
4. Extended data from a split-pool scRNA-seq (Parse Biosciences) mapped to the WT-only dataset (10x Genomics) (Supplementary Fig. 8b, c).

**Validation of label transfers.** WT mappings to adrenal gland references were validated by the presence of relevant markers (Supplementary Fig. 2i). Mutant and tumour cell mappings were not strictly curated via markers (i.e., they were allowed to deviate). When analysing markers of mapped mutant and tumour cells, cells with a prediction score of 0.4 or higher were used to minimise ambiguous mappings and maximise marker discovery. Shared markers were consistently found between the query and the cognate cells in the reference, even though their number varied (Fig. 7c, Supplementary Fig. 8d).

**Visualising label transfers with glasswork plots.** To visualise cell mappings, we used "glasswork plots", in which the UMAP of the reference was used to define the coordinates of concave hulls for each cluster (calculated with R package *concaveman* v1.1.0). Query cells mapping to each cluster were plotted at random positions within their mapped target cluster hull to mitigate overplotting. Input cell

populations for the plot were downsampled evenly by condition and stage ($n = 1000$ cells) to avoid sampling effects.

**Mutation score analysis.** To calculate the mutation score, we focused on days 9, 14, and 19 as they contained samples from all conditions. We encoded each cell's genotype as a number $G$ based on the genetic lineage of hESC lines: *G(WT) = 0, G(17q)=1, G(17q1q)=2,* and *G(17q1qMYCN)=3.* We then calculated the mutation score $m$ as the mean $G$ of the cell's $K$ nearest neighbouring cells ($K = 30$) in the scVI reduction's neighbourhood graph (see *"Normalisation, clustering, and marker gene analysis"*). Division by 3 yielded a score between 0 and 1. Intuitively, the mutation score of a cell indicates whether a cell phenotypically resembles wild-type cells or cells with a given number of relevant alterations independent of its own genotype. To find genes correlated with the mutation score, we calculated Pearson correlation coefficients with gene expression in three settings: (i) correlation for each gene with $m$ in all cells; (ii) correlation for each gene with $m$ leaving out the 17q1qMYCN cells, to emphasise subtle correlations with CNAs; and (iii) correlation for each gene and the neighbourhood entropy (Shannon entropy of all genotype scores $G$ of the $K$ nearest neighbours), to find genes appearing in mixed regions. All non-duplicate absolute correlations (calculated using R's *cor.test*, parameters: *method = "pearson", exact = TRUE*) were subject to Bonferroni correction and ranked. The top correlated genes ($p \leq 0.05$) per differentiation stage (D9, D14, D19) are reported in Supplementary Data 9.

**Pre-processing and mapping of NB tumour data.** We collected scRNA-seq data for tumours with reported *MYCN* amplification from three sources from the stated database or the corresponding authors:
- Three samples (all primary adrenal, 2 male [Dong_T162, Dong_T230], 1 female [Dong_T200]; accession GSE137804 [Gene Expression Omnibus])[17],
- three samples (1 primary adrenal, 1 primary intraspinal, 1 relapse/occipital subcutaneous bone metastasis [Jansky_NB14]; 1 female [Jansky_NB08], 2 male [Jansky_NB01, Jansky_NB14]; accession EGAS00001004388 [European Genome-Phenome Archive])[15],
- and four samples (all metastatic bone marrow; 3 female [Fetahu_M1, Fetahu_M3, Fetahu_M4], 1 male [Fetahu_M2]; accession GSE216176 [Gene Expression Omnibus])[23].

Additional details about each dataset are available from the original research articles. In each dataset, cells with more than 500 reads per barcode and mitochondrial DNA less than 20% were kept for further analysis. We then performed an adrenal gland mapping[16] (same workflow as described above) and discarded cells mapping to the category "HSC_and_immune". This process left us with strong CNA profiles (see below) at key genomic positions such as chr2p (*MYCN* locus). Cells were then subjected to default basic scRNA-seq processing (see above) and mapped onto our WT-only reference (see above).

**Inference of CNA profiles from scRNA-seq data.** To infer tumour cell CNA profiles from scRNA-seq expression data, we used the *infercnv*[139] R package (v1.10.1). We first removed cells with less than 500 UMI counts. Then, we created a pan-patient healthy reference cell population by sampling from each patient 500 cells that we determined to be HSC/immune cells based on a mapping to a human embryonic adrenal gland reference[16]. For every patient, we then ran *infercnv* with the non-HSC/immune cells as the main input and the pan-patient HSC/immune cells as a reference. The *cutoff* parameter was set to 0.1, all other parameters were left at their default values.

**Pre-processing and analysis of NB bulk RNA-seq data.** We obtained bulk RNA-seq counts and associated metadata from patient-derived NB samples from three sources: TARGET[24] (phs000467 [Genomic Data Commons]), SEQC[70,71] (GSE49711 [Gene Expression Omnibus]) and

from our institution[96,99,140–146] (labelled "CCRI" in the figures; GSE94035, GSE147635 and GSE172184 [Gene Expression Omnibus]). Open access unstranded counts from TARGET patients were obtained directly from the GDC data portal (subsection TARGET:NBL, phs000467). Counts from the CCRI patients were obtained in-house. Both CCRI and TARGET datasets were normalised using DESeq2[147] (v1.34.0) and transformed using the variance stabilising transformation. A prenormalised $\log_2$ SEQC matrix was exponentiated, rounded to the nearest integer, and subjected to variance stabilising transformation. In all datasets, the names of relevant marker genes were harmonised manually in case the gene was found with a different name. Each dataset was analysed separately due to differences in count quantification and normalisation. PCA projections of the normalised variables revealed mainly biological/clinical variables (and not technical variables) having major weight in the variance of the datasets. Only NB data collected at diagnosis were used for our analyses (discarding, e.g., ganglioneuroblastoma and relapse data). To quantify the in-vitro cluster signature strength, we used the intersection of markers found both in our in-vitro WT-only dataset (Supplementary Data 2) and in the tumour scRNA-seq datasets (Supplementary Data 10). We then used the function *gsva* (from *GSVA*[148] v1.42.0; parameters: *method = "ssgsea"*) to calculate signature scores for each of the shared cluster signatures.

**Survival analysis.** We obtained survival data for the SEQC cohort from the original publication[70]. Event-free survival (EFS) was defined as time from diagnosis to any of the following events: Relapse/progression of disease, or death due to any cause and secondary malignancies. Patients without events were censored at last follow-up evaluation. Statistical analyses were performed using SAS (v9.4). Cluster signatures (see previous section) were dichotomised using the median value, and the impact of these signatures on EFS was evaluated in a Cox-proportional hazards model adjusting for stage4 (yes/no), age-group (<18, 18–60, >60) and *MYCN* amplification status (yes/no). Each cluster signature was evaluated separately. Two-sided $P$ values were adjusted for multiple hypothesis testing using the Benjamini-Hochberg method and are reported together with hazard ratios and two-sided 95% confidence intervals in Supplementary Data 11.

**Split-pool single-cell RNA sequencing (Parse Biosciences)**
**Library generation and sequencing.** Cells were harvested with Accutase to create a single-cell suspension, then were counted using Bio-rad Tc10 Automated cell counter in the presence of Trypan Blue Stain (Bio-rad). For cell fixation we used the Evercode Fixation v2 Kit (SKU: ECF 2001, Parse Biosciences, Seattle, USA) as per manufacturer instructions. A maximum of 5000 cells per sample were multiplexed using the Evercode WT Mega v2 kit (Parse Biosciences). Three rounds of combinatorial barcoding were performed, and cells were then pooled and split into 16 sub-libraries (one small 5000-cell sub-library and 15 large sub-libraries of 32,000 cells each). After DNA amplification and library prep, the small library was sequenced as part of a larger Illumina NovaSeq S4 flowcell and the 15 large sub-libraries on one dedicated NovaSeq S4 platform in 2 × 150 bp paired-end mode.

**Raw data processing and alignment.** Raw sequencing data were processed with the *split-pipe* v1.0.6p software (Parse Biosciences) for cell-level demultiplexing and alignment to the human reference transcriptome (*refdata-gex-GRCh38-2020-A* assembly provided by 10x Genomics; parameters: -m all -c v2). Following initial data processing, all subsequent analyses were performed in R (v4.1.3) using Bioconductor packages and the *Seurat*[125–127] (v4.1.0) package.

**Basic processing, quality control, and marker analysis.** We applied the *cb_filter_count_matrix* (default parameters) function from *canceRbits* (v0.1.6; default parameters) to remove cells with high mitochondrial

counts (>15%), unusually high/low number of genes (<300 genes or z-score of log(genes) not in range (−3, 3)), abnormally high/low number of reads (z-score of log(transcripts) not in range (−3, 3)), or an abnormal transcript-to-gene ratio (z-score of residuals of loess fit of "log(genes) - log(transcripts)" not in range (−5, 5)), and the *cb_seurat_pipeline* function (parameters: *seurat_max_pc* = 15, *metric* = "manhattan", *k_param* = 20, *n_neighbors* = 40, *cluster_res* = 0.3) to perform a standard Seurat analysis workflow including data normalisation, dimensionality reduction, and clustering. Subsequently, the data were mapped to our 10x-based WT-only reference as described above. To identify marker genes for cells mapped to different clusters of the WT reference (for cells with prediction score >=0.4) we again used DElegate::FindAllMarkers2. (DElegate v1.1.0; *parameters: method* = "deseq", *min_fc* = 1, *min_rate* = 0.5, *replicate_column* = "replicate"), and kept all genes with adjusted pvalue of 0.05 of less.

### Chromatin accessibility mapping (ATAC-seq)

**Library generation and sequencing.** ATAC-seq was performed using established protocols[77]. Briefly, 20,000 to 50,000 cells were lysed in the transposase reaction mix (12.5 µl 2xTD buffer, 2 µl TDE1 [Illumina], 10.25 µl nuclease-free water, 0.25 µl 1% digitonin [Promega], and 0.5 µl of 50x cOmplete Mini EDTA-free Protease Inhibitor Cocktail [Roche]) for 30 min at 37 °C. Following DNA purification with the MinElute kit (Qiagen) eluting in 12 µl, 1 µl of eluted DNA was used in a qPCR reaction to estimate the optimum number of amplification cycles. The remaining 11 µl of each library were amplified for the number of cycles corresponding to the Cq value (i.e., the cycle number at which fluorescence has increased above background levels) from the qPCR using custom Nextera primers[149]. Library amplification was followed by SPRI (Beckman Coulter) size selection to exclude fragments larger than 1200 bp. Library concentration was measured with a Qubit fluorometer (Life Technologies), and libraries were quality checked using a 2100 Bioanalyzer (Agilent Technologies). Libraries were sequenced by the Biomedical Sequencing Facility at CeMM using the Illumina HiSeq 4000 platform in 1 × 50 bp single-end mode. Supplementary Data 1 includes an overview of the sequencing data and performance metrics.

**Raw data processing, alignment, and quality control.** Raw sequencing data were processed using *PEPATAC*[150] (v0.9.5; default parameters) including alignment to the human genome (*refdata-cell ranger-atac-GRCh38-1.2.0* assembly provided by 10x Genomics for maximum compatibility with scRNA-seq analyses). Following initial data processing, all subsequent analyses were performed in R (v4.1.3) using Bioconductor packages and *ggplot2*[151] (v3.3.5) and *ComplexHeatmap*[119] (v2.10.0) for plotting. After discarding low-quality data (NRF < 0.65 or PBC1 < 0.7 or PBC2 < 1 or FRiP < 0.025), we removed peaks overlapping blacklisted regions from ENCODE (http://mitra.stanford.edu/kundaje/akundaje/release/blacklists/hg38-human/hg38.blacklist.bed.gz) and merged overlapping peaks across all ATAC-seq datasets to create a common set of consensus genomic regions for subsequent analysis (Supplementary Data 12). Next, we quantified for each input dataset the number of reads overlapping these consensus peaks using *featureCounts*[152] (*Rsubread* v2.8.2).

**Differential accessibility analysis and chromatin modules.** Raw read counts were loaded into DESeq2[147] (v1.34.0; default parameters, design: *-lane+batch+sample_group*) for normalisation (variance-stabilising transformation) and differential analysis. In doing so, we estimated count size factors for normalisation excluding regions on chromosomes with known chromosomal aberrations (i.e., chr1, chr17) to avoid overcompensation due to differences in global signal strength. We queried all pairwise comparisons of sample groups stratified by cell line/condition stratified (time-wise differences, e.g., WT-D3 vs. WT-D0) and between conditions stratified by stage (condition-wise differences, e.g., 17q-D9 vs. WT-D9) and recorded all significantly differentially accessible

regions ($P_{adj}$ ≤ 0.005, |$\log_2$FoldChange| ≥ $\log_2$(1.5); parameters: *pAdjustMethod* = "BH", *lfcThreshold*=log2(1.5), *independentFiltering* = TRUE; Supplementary Data 13). To define chromatin regulatory modules, we divided time-wise differences in WT hESCs ($n$ = 41,699 regions) into six chromatin modules (R1-R6) and condition-wise differences ($n$ = 3914 regions) into three chromatin modules (R7-R9) by hierarchical clustering using the Ward criterion (parameter: *method* = "ward.D2"). To associate ATAC-seq regions with putative target genes, we used the *GenomicRanges*[153] package (v1.46.1) to assign each region to all genes (using the *refdata-gex-GRCh38-2020-A* gene annotation provided by 10x Genomics) with overlapping promoters (transcription start side) or to distal genes whose promoter within a maximum distance of 250 kb whose expression was significantly correlated with the region's accessibility. To this end, we calculated the correlation coefficient between normalised read counts in our ATAC-seq data with the normalised read counts in our matching scRNA-seq data (mean of cells per sample; note, the ATAC-seq was collected from the same experiments as the first replicate experiments for scRNA-seq). We calculated an empirical false discovery rate (FDR) by shuffling RNA/ATAC assignments (10 repetitions) and retained associations with a value ≤0.05. Annotated regulatory regions from the analysis of ATAC-seq data are listed in Supplementary Data 12.

**Overlap enrichment analysis for chromatin modules.** To characterise the chromatin modules, we interrogated overlaps with genomic regions or associated genes using the hypergeometric test implemented in the *hypeR*[135] package (v1.10.0) via the *cb_hyper* function (*canceRbits* v0.1.6; parameters: collapse = FALSE, min_size = 5, max_size = (<75% of the size of the background dataset)). We looked at three types of overlaps: (a) Annotated reference regions from the DNase hypersensitivity index[78], from the Cis-element Atlas[79], from the Enhancer Atlas[80], and NB subgroup-specific super-enhancers[73], which all catalogue regulatory elements active in different cell or tissue types. (b) Matches to known TF motifs from the *HOCOMOCO* database[154] (v11). Here, we downloaded motifs from the *HOCOMOCO* website (*HOCOMOCOv11_full_annotation_HUMAN_mono.tsv*) and used *motifmatchr* (v1.16.0) to scan the DNA sequences underlying each genomic region for matches. Regions with at least one match to the motif were recorded as potential binding sites. (c) Marker genes from our scRNA-seq analysis of WT hESC differentiation (Fig. 1c; Supplementary Data 2). For this purpose, genomic regions were associated with genes as described above. In each case, we used the entire set of all analysed genomic regions as a background for the enrichment analysis, and we considered overlaps with an FDR-corrected $P$ value less than 0.005 as significant. For motifs, we find the reported $P$ values are inflated and therefore used stricter thresholds: $P_{adj}$ ≤ 0.0000001, |$\log_2$ odds| > $\log_2$(2). All enrichment results are reported in Supplementary Data 14.

**Integration with published ATAC/DNaseI-seq data.** To interrogate accessibility of the chromatin modules in existing data from NB cell lines we used fast gene set enrichment analysis *fgsea* (v1.20.0)[155]. We obtained ready-processed genomic coverage tracks (*wig* or *bigwig* files) from three studies profiling NB cell lines[73,156,157] (GSE138293, GSE224241, GSE136279). Additionally, we obtained data from three studies profiling breast[158] (GSE202511) and lung cancer[159] lines (GSE228832), or human tissue data[160] (https://epigenome.wustl.edu/epimap) as controls. For studies based on older genome assemblies (GSE138293, GSE224241, GSE136279, GSE228832, and EpiMap used hg19), we converted our peak coordinates to hg19 using the *liftOver* R package (v1.18.0). We then used the *GenomicRanges*[153] (v1.46.1) and *plyranges*[161] (v1.14.0) packages to identify genome segments overlapping our peaks and to aggregate the corresponding mean score reported in the coverage tracks, which were then used for gene set enrichment analysis via the *cb_fgsea* function (*canceRbits* v0.1.6; parameters: *max_size = Inf*).

**Identification of transcription factor targets.** To identify putative target genes of TFs, we used *GRNboost2*[81] (*arboreto* library v0.1.6, with Python v3.8.17 via reticulate [v1.24]) to identify genes whose expression could be predicted from the expression of each TF. We tested all TFs in the *HOCOMOCO* database[154] for which at least one motif could be identified in our dataset. We found that stronger association values were reported for stem-cell-related factors, likely because of a proportional overrepresentation of this developmental stage in our dataset. To alleviate this effect and create more balanced data to build our networks on, we downsampled our dataset to no more than 500 cells per cluster and took the average importance value of ten random samples forward for further analysis. Putative targets with high importance values but without a supporting nearby ATAC-seq peak with a motif matching the respective TF were considered indirect targets and discarded from the target gene sets. We found that the range of importance values varied between TFs. We therefore calculated a TF-specific threshold on the importance score to define target genes. To this end, we ranked importance values and used the *changepoint* package (v2.2.3; default parameters) to identify the first point at which the mean values of the curve of importance values changed (disregarding the top 1% highest importance values which often were outliers and disrupted this analysis). The resulting target gene sets were divided into putative activating and inhibiting interactions by the sign of the Pearson correlation coefficient *r* of the respective TF-target pairs (using the mean correlation value of the same eight random samples as used for *GRNboost2*). Interactions with $|r| < 0.1$ were discarded. To calculate the average expression of target genes in each cell we used only activated targets ($r > 0.1$) and the Seurat module score. To identify significant overlaps between target genes and gene sets D9_1 – D9_4 (Supplementary Data 15), we used the *hypeR*[135] package (v1.10.0) via the *cb_hyper* function (*canceRbits* v0.1.6; parameters: collapse = FALSE, min_size = 0, max_size = Inf), considering TFs with $P_{adj} \leq 0.05$, $|\log_2 \text{ odds}| \geq \log_2(4)$, and frequency ≥5% as significant. All target gene sets are reported in Supplementary Data 15 and all enrichment results in Supplementary Data 16.

**Gene-regulatory network visualisation.** For the visualisation of gene-regulatory networks, we used the *igraph* package (v1.3.1). A directed graph was constructed from edges between genes in the gene sets D9_1, D9_2, D9_3, or D9_4 (Supplementary Data 9) and TFs found enriched in the overlap with these genes (Fig. 10d). The same automated graph layout (function *layout_with_fr()*) was used to draw mutant-specific network diagrams. To generate mutant-specific networks (Fig. 10f), we selected cells derived at D9 and parameterised node colour to indicate the mean scaled expression of the genes in those cells and node size to indicate the mean scaled TF target score (Seurat module score) for TFs or the mean scaled expression for non-TFs. To simplify plots, we only labelled TFs with positive mean scaled expression values (>0.05) and manually aggregated many overlapping values, but all node labels are shown in Supplementary Fig. 13c.

**Reporting summary**
Further information on research design is available in the Nature Portfolio Reporting Summary linked to this article.

## Data availability
The single-cell RNA-seq and ATAC-seq data generated in this study have been deposited in the Gene Expression Omnibus (GEO) under accession code GSE219153. The public scRNA-seq data from NB tumours used in this study are available in GEO under the accession codes GSE147821, GSE216176, and GSE137804, and in the European Genome-Phenome Archive (EGA) under accession code EGAS00001004388. Public ATAC-seq data from NB cell lines and controls used in this study are available in GEO under accession codes GSE138293, GSE224241, GSE136279, GSE202511, and GSE228832, and from the EpiMap website (https://epigenome.wustl.edu/epimap). Bulk RNA-seq data from NB tumours are available in GEO under accession codes GSE49711, GSE94035, GSE147635, and GSE172184, and in dbGAP under accession code phs000467 [https://www.ncbi.nlm.nih.gov/projects/gap/cgi-bin/study.cgi?study_id=phs000218]. Source data are provided with this paper. Additionally, processed data from this paper can be accessed and browsed interactively via our GitHub page [https://github.com/cancerbits/saldana_montano2024_ncnb/] and via the R2 Genomics Analysis and Visualisation Platform [http://r2platform.com/halbritter24/]. Source data are provided with this paper.

## Code availability
Computer code used for the data analysis in this study is available via our GitHub page (https://github.com/cancerbits/saldana_montano2024_ncnb/) and a persistent copy of this repository is available via Zenodo (https://doi.org/10.5281/zenodo.10891507).

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

## Acknowledgements

We would like to thank the Biomedical Sequencing Facility at the CeMM Research Center for Molecular Medicine of the Austrian Academy of Sciences for assistance with next-generation sequencing, Bettina Brunner-Herglotz (CCRI) for her technical assistance, and Duncan Baker (Sheffield Diagnostic Genetic Services, Sheffield Children's Hospital) for carrying out karyotyping of hESC lines. We would like to acknowledge Yann Jamin and Barbara Martins da Costa (ICR, London) for help with the MRI and animal work, respectively. We are also grateful to Igor Adameyko and Polina Kameneva (Medical University of Vienna), Sofie Mohlin (Lund University), and Christoph Bock (CeMM Research Center for Molecular Medicine) for critical reading of the manuscript. We would also like to thank Eszter Söjtöry (CCRI) and Jan Koster (Amsterdam UMC) for their help with providing interfaces for exploratory data analysis for our data. For the purpose of open access, the authors have applied a Creative Commons Attribution (CC BY) licence to any Author Accepted Manuscript version arising. The following funding sources contributed directly or indirectly to the support of this study: Alex's Lemonade Stand Foundation for Childhood Cancer (ALSF) 20-17258 (M.D., M.F., F.H.); Austrian Academy of Sciences (OEAW) 25902 (M.C.B.); Austrian Research Promotion Agency (FFG) 7940628 533 (M.D.); Austrian Science Fund (FWF) 10.55776/P35072 (I.S.F.), 10.55776/P32001 (E.M.P.), 10.55776/P34832 (E.M.P.), 10.55776/P35841B (S.T.M.), 10.55776/TAI454 (F.H.), 10.55776/TAI732 (F.H.), 10.55776/PAT1300223 (F.H.); Biotechnology and Biological Sciences Research Council (BBSRC) BB/T007222/1 (K.B.), BB/P000444/1 (A.T.); Children's Cancer and Leukaemia Group/Little Princess Trust CCLGA 2018 06 (H.E.B.), CCLGA 2020 19 (H.E.B., A.T.); Neuroblastoma UK/Children's Cancer and Leukaemia Group/Little Princess Trust CCLGA 2019 28 (A.T.); CRUK Programme Award A28278 (E.P., L.C.); Donald E. and Delia B. Baxter Foundation Fellowship Award (M.H.); EC Horizon 2020 826494 (S.T.M.), 824070 (A.T.); Higher Education Funding Council for England (HEFCE) ICR (L.C.); Medical Research Council UK (MRC) MR/X000028/1 (I.B.), MR/X007979/1 (I.B.), MR/V002163/1 (A.T.); National Institutes of Health (NIH) R00CA197484 (M.H.); St. Anna Kinderkrebsforschung (E.M.P, R.L., G.C., M.D., S.T.M., F.H.); UK Regenerative Medicine Platform MR/R015724/1 (I.B.); Vienna Science and Technology Fund (WWTF) LS18-111 (S.T.M.).

## Author contributions

I.S.G. and L.M.G. championed the experimental and computational work in this study, respectively. A.T. and F.H. jointly supervised this work. Furthermore, the following authors contributed equally to important aspects of this study: K.B., C.B., E.P., L.S., D.S., as well as L.C., S.T.M., and M.F. Formal contributions in authorship order (CrediT taxonomy): Conceptualisation: A.T., F.H.; Data curation: L.M.G., I.S.G., L.S., K.B., E.P., D.S., S.W.Z., E.B., I.S.F., P.Z., U.P., G.C., S.T.M., M.F., A.T., F.H.; Formal Analysis: L.M.G., I.S.G., C.H., E.P., S.W.Z., P.Z., U.P., C. Sturtzel., M.S., G.C., H.E.B., A.T., F.H.; Funding acquisition: K.B., I.S.F., M.C.B., W.W., P.W.A., I.B., H.E.B., M.D., L.C., S.T.M., M.F., A.T., F.H.; Investigation: I.S.G., L.M.G., K.B., E.P., L.S., D.S., R.A.L., E.B., I.S.F., M.B., A.W.W., C. Sturtzel, C. Souilhol, S.T., P.B., M.R., M.G., M.C.B.; Methodology: I.S.G., L.M.G., L.S., E.P., I.S.F., M.B., P.B., M.G., G.C., H.E.B., M.D., L.C., S.T.M., M.F., A.T., F.H.; Project administration: A.T., F.H.; Resources: W.W., E.M,P., M.H., R.L., H.E.B., M.D., L.C., S.T.M, M.F., A.T., F.H.; Software: L.M.G., C.H.; Supervision: E.M.P., G.C., H.E.B., M.D., L.C., S.T.M., M.F., A.T., F.H.; Visualisation: I.S.G., L.M.G., K.B., E.P., D.S., C. Sturtzel, G.C., A.T., F.H.; Writing – original draft: I.S.G., L.M.G., A.T., F.H.; Writing – review & editing: I.S.G., L.M.G., K.B., C.H., E.P., L.S., I.S.F., C. Sturtzel, C. Souilhol, M.B., P.B., M.G., M.C.B., E.M.P., M.H., P.W.A., I.B., H.E.B., G.C., M.D., S.T.M., M.F., A.T., F.H.

## Competing interests

P.W.A. is a member of the Scientific Advisory Board of TreeFrog Therapeutics and receives royalties from the Wistar Institute for antibody sales. All other authors declare no competing interests.

## Additional information

[1]Centre for Stem Cell Biology, School of Biosciences, The University of Sheffield, Sheffield, UK. [2]Neuroscience Institute, The University of Sheffield, Sheffield, UK. [3]Sheffield Institute for Nucleic Acids (SInFoNiA), School of Medicine and Population Health, The University of Sheffield, Sheffield, UK. [4]St. Anna Children's Cancer Research Institute (CCRI), Vienna, Austria. [5]Division of Clinical Studies, The Institute of Cancer Research (ICR) & Royal Marsden NHS Trust, London, UK. [6]Department of Dermatology, Medical University of Vienna, Vienna, Austria. [7]Labdia Labordiagnostik GmbH, Vienna, Austria. [8]Biomolecular Sciences Research Centre, Department of Biosciences and Chemistry, Sheffield Hallam University, Sheffield, UK. [9]CeMM Research Center for Molecular Medicine of the Austrian Academy of Sciences, Vienna, Austria. [10]Children's Hospital Los Angeles, Cancer and Blood Disease Institutes, and The Saban Research Institute, Los Angeles, CA, USA. [11]Keck School of Medicine, University of Southern California, Los Angeles, CA, USA. [12]Present address: Medical University of Vienna, Department of Neurology, Division of Neuropathology and Neurochemistry, Vienna, Austria. [13]These authors contributed equally: Ingrid M. Saldana-Guerrero, Luis F. Montano-Gutierrez. [14]These authors jointly supervised this work: Anestis Tsakiridis, Florian Halbritter.
✉ e-mail: a.tsakiridis@sheffield.ac.uk; florian.halbritter@ccri.at

