## [Peer Review File · Nature Communications]

REVIEWER COMMENTS

Reviewer #1 (Remarks to the Author):

In their manuscript entitled “A human neural crest model reveals the developmental impact of neuroblastoma-associated chromosomal aberrations” Aldana-Guerrero I. and colleagues establish a hESC-based model to study the impact of large copy number alterations found in neuroblastomas on neural crest development. The model they use is nicely described using droplet based scRNAseq and recapitulates the neural crest development stages. To study the effect of copy number alteration, they use spontaneously altered isogenic cell pairs, created through culture adaptation. This represents a limitation of this study, as the alterations were not engineered but occurred spontaneously. It is unclear whether additional changes occurred in the cells during this process. MYCN, on the other hand, was expressed inducibly using doxycycline, making it a more controlled perturbation. Assessment of these cells with scRNAseq revealed that each perturbation reduced the number of cells in mature cell differentiation states, suggesting that the CNA and MYCN alter cell differentiation. The authors argue that the cells with CNA and MYCN expression were in a pretumorigenic cell state, as they survived for longer time when xenotransplanted into zebrafish. Cells also showed some similarity with patient tumor cells when scRNAseq of both were compared. Lastly, the authors performed ATAC sequencing and identified gene regulatory networks in their models that changed in the presence of CNA/MYCN. The methods used to analyze the data are well chosen and as far as I can evaluate rigorously performed.

Overall, this is a useful model to study neural crest differentiation that could facilitate research about neuroblastoma development. I would recommend this manuscript to be published after addressing my concerns listed below.

Major comments:

Major comment #1: The authors describe the effect of MYCN overexpression in a specific CNV background (17q1q gain). However, they do not discuss whether the effects observed in 17q1qMYCN regarding impaired differentiation and, particularly, acquisition of tumorigenic-like features might just be induced by MYCN overexpression alone in a WT genetic background. In the discussion, they speculate that CNVs facilitate MYCN overexpression by upregulating anti-apoptotic and DNA-damage-repair-related genes and discuss that MYCN overexpression alone has shown increased apoptosis in early sympathoadrenal cells in other studies, but they don't evaluate these effects in their own model. As the most drastic effects are observed in the 17q1qMYCN cell model, an additional MYCN overexpression in “WT” hESC cell model should be included in the study to test which effects are driven by MYCN alone. If this leads to massive apoptosis, as suggested by the authors, this datapoint should be presented and added.

Major comment #2: Which additional genomic changes do these models harbor? The authors performed WES but do not describe the mutations found there. The authors showed some karyotypes, but most

genomic alterations can't be assessed based on such karyotypes. At today's low prices for whole genome sequencing, I would have expected detailed genomic analyses to be presented to confirm that no additional alterations are harbored in these cells apart from 17q gain etc. . This is an essential piece of information without which it is impossible to evaluate the model system. At least the WES should be analyzed in detail and results presented in the manuscript.

Major comment #3: The data from the zebrafish xenograft are a weak point. Why did the cells not grow in vivo? Which additional changes would be necessary for the cells to grow? i.e. what are the differences between their model and a human neuroblastoma (eg. ALK/MAPK pathway alterations? TERT?) Without these answers I would consider omitting this part of the paper as it seems preliminary to me.

Minor comments:

Minor comment #1: Could the authors clarify that Doxycycline alone has no effect on cells or any other relevant phenotype to this study? Maybe by including a Dox+ WT / 17q / 17q1q control at D5 to discard Dox alone effects, at least in the assays in which effects in tumorigenesis are shown.

Minor comment #2: MYCN expression levels from the 17q1q MYCN model in scRNA-seq data are not shown. How does it compare with MYCN levels in MYCN amplified NB tumors?

Minor comment #3: Could the authors clarify the CNV genetic background (17q / 1q) of the 11 NB tumors used in this study? Could the authors group MYCN amplified tumors based on CNV background (17q / 17q1q / no CNV) evaluate differences in cell annotation?

Minor comment #4: In Extended Data fig 8, the InferCNV results are missing for tumor: Jansky_NB11.

Minor comment #5: In fig 5g, similarities across tumors from the same cohort in cell annotations are observed. Could the authors elaborate on why this might be the case?

Minor comment #6: consistent misuse of CNV to mean somatic aberrations throughout, should be adjusted to be CNAs to avoid confusion.

Reviewer #2 (Remarks to the Author):

The embryonic origins of developmental cancers such as neuroblastoma can be difficult to model due to a lack of suitable models, which reflect the embryonic state. Here, to model the effect of 17q/1q CNVs proposed to drive neuroblastoma, authors use human ES cell cultures that they then differentiate to trunk neural crest cells and sympathoadrenal derivatives. They find that the CNVs combined with MYCN overexpression are blocked in differentiation and that mutant NC cells resemble neuroblastoma.

There is a wealth of transcriptomic and epigenetic information in this paper pointing towards effects on the chromatin landscape specific to trunk neural crest, which is affected in the mutant conditions. This is an exciting paper with interesting data that will be useful for the field.

I have a few suggestions for improvement and clarification:

1) Effects of 17q and 17q1q on cell proliferation in Figure 4. There is a brief mention of decrease in cell proliferation/KI67 in the 17q and 17q1q lines, which is then reversed (and further exceeded) in the MYCN lines. There also seem to be very different cell platings from day 9 to day 14 (in Fig 4B). It would be helpful to see the relative morphology of the cells and cell colonies in these experiments, especially given the drastic differences in 17q1qMYCN. There are some images in Extended data Fig 4e, it may be helpful to have the full set of panels (all four genotypes) in the main text.

2) It will also be really helpful to look at apoptosis in these cultures (Fig 4b). Is there drastic die-off in the 17q and 17q1q lines?

3) 4C please show wt controls

4) 4D xenograft experiments – please show fish carrying the positive controls (e.g SK-N-BE2C-H2B-GFP cell lines from extended data Fig 6). Also include initial injected cells (day 0)?

5) Suppl Videos 1-2: there are only 2 movies (of 17q1q and 17q1qMYCN) – please include the other two genotypes

6) “Time lapse imaging showing higher degree of motility” please include actual quantitation to support this statement.

Reviewer #3 (Remarks to the Author):

Interesting paper that looks at NC differentiation and effects of chromosome 17/1q gains and MYCN expression in hESCs and the impact on differentiation to NC, on chromatin and developmental TFs.

1. It is likely that both MYCN amplification and CNVs in neuroblastoma represent late events, as normal cells likely activate checkpoints in response to these genetic changes. In this regard, can the authors detail whether the 17q and 17q1q ESC lines acquired mutations in genes that enabled them to tolerate these acquired CNVs? Did introduction of MYCN and differentiation to NCC change the frequency of any candidate mutations present in these lines, and absent in the parent lines? Damaging mutations or polymorphisms that cooperate with MYCN to drive proliferation might increase in frequency after introduction of MYCN.

2. Can the 17q and 1q gains in these ESCs be further detailed and compared to those found in human NB, both in regards to the region of 17q and 1q that were gained, and by sequencing comparison with human tumors that do or don't have both regions gained?

3. Similarly, MYCN amplification in human disease is likely acquired in established NC rather than during early NC specification. In this regard, MYCN might be expected to block differentiation even after NC establishment in this experimental system. Does turning on MYCN after differentiation to NC (on or after day 19) lead to similar changes as those seen in Fig 2, in which MYCN was turned on early in the differentiation protocol?

4. I was puzzled why the manuscript and Fig 4-7 in particular omitted analysis of MYCN in wild-type and 17q cells. Aren't these critical comparators to understand how MYCN, 1q gain and 17q gain cooperate contribute to the effects the authors observe in the 17q1q MYCN cells? Ideally, adding the analysis of MYCN in wt cells, and MYCN in 17q cells offers insights into distinct roles for MYCN, 17q, and 1q that can't be easily gleaned from comparing 17q1q with 17q1qMYCN in the absence of these other samples. These analyses would also help clarify roles for each genetic aberration in additional experiments detailed in subsequent experiments.

5. Why did authors implant cells into fish rather than into murine adrenal gland in Fig 4? In the 16% that developed increased xenotransplant size, were these tumors transplantable? Can histology be shown, and markers of proliferation, apoptosis, senescence, mes and adr differentiation?

Reviewer #4 (Remarks to the Author):

In this manuscript, Saldana-Guerrero et al utilized an hESC model of trunk neural crest (NC) differentiation to explore the connections between CNVs, MYCN amplification, and tumor initiation in NB. The authors differentiated hESC lines with CNV gains in chr. 17q and 1q and induced MYCN overexpression to examine how these factors affect NC differentiation and the acquisition of tumorigenic characteristics. The authors have conducted single-cell seq experiments at different time points during in vitro NC differentiation of different CNV hESC lines (17q, 17q1q, and 17q1qMYCN). They showed that the CNVs promote immature NC progenitor phenotypes and impair differentiation, and that combining CNVs with MYCN overexpression exacerbates these effects and increases cell proliferation and survival. Although the in vitro NB model study is of interest, there are several major limitations in their analyses and claims to be addressed.

Critiques:

1. The authors used in vitro differentiation of hESC cell line with human NB-associated CNVs to model tumour initiation of NB, however, those cells including 17q1qMYCN failed to initiate NB tumours. The cells (e.g., 17q1qMYCN) they generated, although transiently increase cell survival for a couple of days in xenotransplants, are not transformed or tumorigenic, and do not model NB tumor cells, which limits the impact of the study.
 2. An important reference of the iPSC cell line with overexpression of MYCN alone is lacking in the study. The authors should include the data for their analyses in Figures 2, 4, 6, and 7. The acquisition of “NB hallmarks” might be likely due to MYCN overexpression alone rather than 17q, 1q CNVs.
 3. In Figure 1B, the cell number on day 3 of WT cell differentiation is very low for the single-cell analysis. Since it is an important reference for comparison of other CNV mutant cells, the authors should increase the number of cells on day 3 in their analysis.
 4. In Fig 2g, 17q1qMYCN cells map to day 9 of WT cells, but the SOX10 level is lower than that of other cells such as WT and 17q1q if these cells are maintained in nc state.
 5. Along the same line, Plp1 is predominantly expressed in 17q1qMYCN cells, however, it has been shown as a Schwann cell lineage marker. The authors should use canonical nc markers to differentiate them from Schwann cell lineage cells.
 6. In Figure 3f, the authors should include the data with human NBs that have the same copy number variants (17q1q) and MYCN overexpression since they were identified in the majority of NB tumours.
- In Fig 3a, the authors should include human NB tumours for the pathway analysis.
7. Figure 4 cell survival and proliferation assays, the authors should include an iPSC cell line with overexpression of MYCN alone.

8. In Fig. 5d, the projection of tumor cells is hard to detect in the UMAP. The chromaffin-like and cortex-like cells are so rare, which makes it difficult to compare with SYM-like NB cells. The authors should include more chromaffin-like and cortex-like tumour samples.

9. In Fig. 5f heatmap, MYCN expression in different tumor cells (LATE SENSORY, HDMYCN, SCPSYM and SYM) appears inconsistent with Fig5e, which shows MYC amplifications in all these NB tumors.

Also, in Figure 5g, the authors should explain the significance of the tumour heterogeneity analysis. Does the predicted score correlate with malignancy or patient survival information?

10. In Fig 6, the authors should compare the data from their cell lines with the published human NB tumor data (bulk RNA-seq, single RNA-seq data, and ATAC-seq data) to show if the CNV cell lines exhibit similar tumorigenic hallmarks.

REVIEWER 1

“In their manuscript entitled “A human neural crest model reveals the developmental impact of neuroblastoma-associated chromosomal aberrations” Aldana-Guerrero I. and colleagues establish a hESC-based model to study the impact of large copy number alterations found in neuroblastomas on neural crest development. The model they use is nicely described using droplet based scRNAseq and recapitulates the neural crest development stages. To study the effect of copy number alteration, they use spontaneously altered isogenic cell pairs, created through culture adaptation. This represents a limitation of this study, as the alterations were not engineered but occurred spontaneously. It is unclear whether additional changes occurred in the cells during this process. MYCN, on the other hand, was expressed inducibly using doxycycline, making it a more controlled perturbation. Assessment of these cells with scRNAseq revealed that each perturbation reduced the number of cells in mature cell differentiation states, suggesting that the CNA and MYCN alter cell differentiation. The authors argue that the cells with CNA and MYCN expression were in a pretumorigenic cell state, as they survived for longer time when xenotransplanted into zebrafish. Cells also showed some similarity with patient tumors cells when scRNAseq of both were compared. Lastly, the authors performed ATAC sequencing and identified gene regulatory networks in their models that changed in the presence of CNA/MYCN. The methods used to analyze the data are well chosen and as far as I can evaluate rigorously performed. Overall, this is a useful model to study neural crest differentiation that could facilitate research about neuroblastoma development. I would recommend this manuscript to be published after addressing my concerns listed below.”

Response: We would like to thank the reviewer for their positive assessment of our manuscript and for supporting its publication in Nature Communications. The reviewer raised several valid concerns, which we believe we addressed in this revised submission.

“Major comment #1: The authors describe the effect of MYCN overexpression in a specific CNV background (17q1q gain). However, they do not discuss whether the effects observed in 17q1qMYCN regarding impaired differentiation and, particularly, acquisition of tumorigenic-like features might just be induced by MYCN overexpression alone in a WT genetic background.” (Reviewer comment continued below)

Response: We have included new analysis of trunk NC differentiation (based on scRNA-seq and immunofluorescence) and acquisition of tumorigenic features (resistance to apoptosis, cell cycle changes/proliferation, clonogenicity) in cells from all different backgrounds (wild type, 17q gain and 17q1q gains) in the presence and absence of DOX-inducible MYCN overexpression (revised **Fig. 4, Supplementary Figs. 8, 9**). Our data show that ectopic MYCN induces significant cell cycle changes (decrease in G1/increase in S phase) and increase in proliferation and clonogenicity only in the presence of chr17q and chr17q/1q gains but not in wild-type background (**Fig. 4b-d**, pasted below for convenience). Moreover, the presence of these CNAs was found to promote the differentiation block by MYCN overexpression which we previously described, while the differentiation defects were less pronounced in WT background (**Fig. 4a; Supplementary Figs. 8c,e**, pasted below for convenience). We have described these new data in the revised manuscript on pages 6-7.

Excerpt from Supplementary Fig. 8c. The bar plots show the closest matched cell in the wild-type differentiation landscape (day 3 to 28 of differentiation) for scRNA-seq data collected from different cell lines / conditions at D9. While wild-type D9 cells (note, this is independently collected data) map predominantly to D9 cells in the reference, cells from MYCN-overexpressing mutant map increasingly to earlier developmental stages. The effect increases in strength depending on CNA background (WT < 17q < 17q1q).

Supplementary Fig. 8e. IF images at D19 of differentiation show specification of sympathetic neurons is blocked by *MYCN* overexpression in 17q and 17q1q, whereas WT cells overexpressing *MYCN* can still form neurons (axons) in WT background.

“In the discussion, they speculate that CNVs facilitate MYCN overexpression by upregulating anti-apoptotic and DNA-damage-repair-related genes and discuss that MYCN overexpression alone has shown increased apoptosis in early sympathoadrenal cells in other studies, but they don’t evaluate these effects in their own model. As the most drastic effects are observed in the 17q1qMYCN cell model, an additional MYCN overexpression in “WT” hESC cell model should be included in the study to test which effects are driven by MYCN alone. If this leads to massive apoptosis, as suggested by the authors, this datapoint should be presented and added.”

Response: We have now included analysis of apoptosis (assayed by low cytometry-based quantification of cleaved caspase-3 levels) in cells derived from all distinct genetic backgrounds (in the presence and absence of *MYCN* overexpression) as these cells transit towards sympathoadrenal/autonomic progenitors (**Supplementary Fig. 9b,c**, pasted below). As we had hypothesised, our data indicate that *MYCN* overexpression induces significantly higher levels of apoptosis in cells that do not harbour the chr17q/1q chromosome arm gains. However, this effect was only seen at earlier differentiation time points (day 11), while at later time points (days 12-14), *MYCN*-driven apoptosis in WT and 17q1q cells were comparable and substantially higher than those in 17q cells. DNA damage levels (assessed by examining γ H2AX foci; **Supplementary Fig. 9d**) were lower in 17q and 17q1q cells than in WT at D9, and we found they increased significantly upon *MYCN* overexpression but only in 17q1q background. We have amended our conclusions accordingly and removed the speculative statement pointed by the reviewer from the discussion. Our new data are described on page 7 of the revised manuscript.

Supplementary Fig. 9b,c. Apoptosis is increased following *MYCN* overexpression in differentiating WT cells and not in 17q or 17q1q background. However, at later stages of differentiation, 17q1q cells expressing *MYCN* also became more apoptotic.

“Major comment #2: Which additional genomic changes do these models harbor? The authors performed WES but do not describe the mutations found there. The authors showed some karyotypes,

but most genomic alterations can't be assessed based on such karyotypes. At today's low prices for whole genome sequencing, I would have expected detailed genomic analyses to be presented to confirm that no additional alterations are harbored in these cells apart from 17q gain etc. This is an essential piece of information without which it is impossible to evaluate the model system. At least the WES should be analyzed in detail and results presented in the manuscript.”

Response: We have now included a detailed analysis of SNVs and CNAs in our cell lines in our revised manuscript (**Supplementary Figs. 4b,c**, pasted below; **Supplementary Tables 4,5**). We found one point mutation with a possible implication in NB biology (*BCOR*^{L1673F}) and a small loss on chr2q (harbouring no NB-linked genes). We have now added additional explanations to the text and discussion. Given the comparatively low number of affected genes (relative to the many genes in chr17q/chr1q) and the fact that none of these are currently considered “key drivers” in NB biology, we labelled the cell lines using their primary NB-associated CNA. Furthermore, we have now included additional scRNA-seq data for a second, unrelated hESC line of (H9/WA09) which had independently also acquired 17q1q gains. These gains are similar but not identical to the gains found in H7 hESCs. We also equipped the H9 cell line with an inducible *MYCN* construct. Reference mapping showed a high degree of consistency between H9 17q1qMYCN and with H7 17q1qMYCN in our data analysis (**Supplementary Fig. 8b**), supporting the notion that the differentiation block we observe is likely a direct result of the presence of the described CNAs rather than specific effects of other acquired mutations (complete [very time- and cost-intensive] characterization of H9, as we have done for H7, would be beyond the scope of this paper). Our new data are described on pages 4, 6, and 13 of the revised manuscript.

“Major comment #3: The data from the zebrafish xenograft are a weak point. Why did the cells not grow *in vivo*? Which additional changes would be necessary for the cells to grow? i.e. what are the differences between their model and a human neuroblastoma (eg. *ALK/MAPK* pathway alterations? *TERT*?) Without these answers I would consider omitting this part of the paper as it seems preliminary to me.”

Response: We employed xenografting in zebrafish larvae as a fast system for assessing the behaviour of our *in vitro*-derived cells in an *in vivo* environment. Zebrafish are a convenient tool for drug testing as they can be maintained inexpensively in large numbers, their larvae are transparent, and experiments do not require specific regulatory approval in early larval stages (e.g. see our recent papers Grissenberger *et al.* (2023), *Cancer Lett*; Sturtzel *et al.* (2023), *npj Precis Onc*). However, we believe that the lower temperatures required for maintaining zebrafish may have affected donor cell survival. We have therefore replaced this part of the manuscript with xenografting experiments performed in mice (and moved the zebrafish as supporting data to a supplementary figure). Our mouse xenografts

clearly demonstrated that DOX-treated *MYCN*-overexpressing 17q1q-derived trunk NC cells give rise to tumours *in vivo*, while this is not the case for their wild type and 17q1q counterparts in the absence of *MYCN* overexpression (see new Fig. 5, pasted below; Supplementary Fig. 10a). The new xenograft experiments are described on page 8 of the revised manuscript.

New Fig. 5. Xenografts in NSG mice demonstrated the tumorigenicity of 17q1qMYCN-derived trunk NC cells (but not WT) by subcutaneous injection (a) and adrenal injection (b).

“Minor comment #1: Could the authors clarify that Doxycycline alone has no effect on cells or any other relevant phenotype to this study? Maybe by including a Dox+ WT / 17q / 17q1q control at D5 to discard Dox alone effects, at least in the assays in which effects in tumorigenesis are shown.”

Response: We have now included data showing that Doxycycline treatment alone at day 5 of differentiation does not alter tumorigenic phenotype-associated features such as clonogenicity, resistance to apoptosis or DNA damage levels in unmodified, differentiating wild type/17q gain hESC-derived trunk NC cells (Supplementary Figs. 9a,c,e, pasted below).

Collage of Supplementary Fig. 9a,c,e. Doxycycline-treatment in WT and 17q cells does not affect clonogenicity (a), apoptosis (c), or DNA damage (e).

“Minor comment #2: *MYCN* expression levels from the 17q1q *MYCN* model in scRNA-seq data are not shown. How does it compare with *MYCN* levels in *MYCN* amplified NB tumors?”

Response: We have added plots indicating *MYCN* expression levels in our model and NB tumour data (bulk RNA-seq from SEQC, TARGET, and our own data from CCRI) to Supplementary Fig. 4f

(pasted below). We found that transcript levels in *MYCN*-amplified tumours are on average approximately 16-fold higher than in non-amplified tumours. Tumours may gain dozens or hundreds of copies of *MYCN* over time. In our *MYCN*-overexpressing cells, we recorded an average 4-fold increase over WT cells. The new data are referenced in 1.137f of the revised manuscript.

“Minor comment #3: Could the authors clarify the CNV genetic background (17q / 1q) of the 11 NB tumours used in this study? Could the authors group *MYCN* amplified tumours based on CNV background (17q / 17q1q / no CNV) evaluate differences in cell annotation?”

Response: We have tried to obtain DNA sequencing-based or SNP array-based annotations for CNAs from the original publications, but these were not available with the exception of Fetahu et al. (2023) data. Instead, the other two studies used a bioinformatics tool called inferCNV to infer CNV profiles based on their scRNA-seq data. We therefore also used inferCNV (**Supplementary Fig. 11a**) to make a call for each tumour dataset (which were consistent with SNP-array-based annotations for the Fetahu et al. data). However, we did not see any strong correlation between CNA background and the mappings in **Supplementary Fig. 11b** (pasted below), based on this small sample of datasets. The complete revised **Figure 6** and **Supplementary Figure 11** are described on page 9.

“Minor comment #4: In Extended Data fig 8, the InferCNV results are missing for tumor: Jansky_NB11.”

Response: We apologize for this mistake. Indeed, there were very few tumour cells in this sample according to our filter criteria ($n < 50$) and we have removed it from our analysis (now shown in **Supplementary Fig. 11a**).

“Minor comment #5: In fig 5g, similarities across tumors from the same cohort in cell annotations are observed. Could the authors elaborate on why this might be the case?”

Response: Given the lower number of samples, it was unclear whether the apparent differences in the “old” Figure 5 (now **Fig. 6**) were biological or technical in nature and we had therefore refrained from speculations. In the revised manuscript, we interrogated expression signatures associated to the mapped clusters in a large collection of NB datasets (bulk RNA-seq) from SEQC. For instance, we found our gene expression signature C5* and C13* to separate tumour data from *MYCN*-amplified and non-amplified tumours (**Fig. 6d**) and that C13* gene expression distinguishes between subgroups of patients with better or worse survival (**Figs. 6e**). The entire text relating to the **Figure 6** has been substantially revised and is now presented on pages 8-9 of the manuscript.

“Minor comment #6: consistent misuse of CNV to mean somatic aberrations throughout, should be adjusted to be CNAs to avoid confusion.”

Response: We have replaced the term “CNV(s)” with “CNA(s)” in line with the reviewer’s suggestion.

REVIEWER 2

“The embryonic origins of developmental cancers such as neuroblastoma can be difficult to model due to a lack of suitable models, which reflect the embryonic state. Here, to model the effect of 17q/1q CNVs proposed to drive neuroblastoma, authors use human ES cell cultures that they then differentiate to trunk neural crest cells and sympathoadrenal derivatives. They find that the CNVs combined with MYCN overexpression are blocked in differentiation and that mutant NC cells resemble neuroblastoma. There is a wealth of transcriptomic and epigenetic information in this paper pointing towards effects on the chromatin landscape specific to trunk neural crest, which is affected in the mutant conditions. This is an exciting paper with interesting data that will be useful for the field.”

Response: We would like to thank the reviewer for their enthusiastic support and helpful suggestions for improvement.

“I have a few suggestions for improvement and clarification. 1) Effects of 17q and 17q1q on cell proliferation in Figure 4. There is a brief mention of decrease in cell proliferation/Ki67 in the 17q and 17q1q lines, which is then reversed (and further exceeded) in the MYCN lines.” (comment 1 continued below)

Response: The proliferation decrease observed was mild and not statistically significant and we have now included more biological replicates of these experiments as well as additional analysis of MYCN-overexpressing cells in wild type and 17q-only backgrounds showing that ectopic MYCN induction increases significantly cell proliferation only in the presence of CNAs (**Fig. 4b-c**, pasted below for convenience; **Supplementary Fig. 9**). We have substantially revised and expanded the text describing the new data from these experiments on pages 7-8.

“There also seem to be very different cell platings from day 9 to day 14 (in Fig 4B). It would be helpful to see the relative morphology of the cells and cell colonies in these experiments, especially given the drastic differences in 17q1qMYCN. There are some images in Extended data Fig 4e, it may be helpful to have the full set of panels (all four genotypes) in the main text.”

Response: We would like to note that there is just one re-plating step in our differentiation protocol taking place at day 9 (=trunk NC stage), which employs the same cell number for all different genotypes

(150,000 cells/cm² for H7 hESCs; see Methods section, page 16). Thus, the day 9 images shown in former Figure 4b (now **Fig. 4d**) depict the confluent trunk NC cultures prior to replating, while the day 14 images show their counterparts after replating and culture in the presence of BMP/SHH signalling agonists for 5 days, conditions which promote a sympathoadrenal identity. We have also now included brightfield images of cells from all different genotypes in the presence and absence of DOX treatment to illustrate differences in morphology (**Fig. 4a**, pasted in the preceding response) and revised the text on pages 7-8.

“2) It will also be really helpful to look at apoptosis in these cultures (Fig 4b). Is there drastic die-off in the 17q and 17q1q lines?”

Response: We have now included analysis of apoptosis in cells derived from all distinct genetic backgrounds in our revised manuscript (**Supplementary Fig. 9b**). Interestingly, this confirmed that – as we had hypothesised – *MYCN* overexpression induced high levels of apoptosis in wild-type cells that do not harbour the chr17q/1q chromosome arm gains, while 17q and 17q1q cells had about the same level of apoptotic cells as untreated WT around day 11 of differentiation. However, at differentiation days 12-14, 17q1qMYCN cells had become as apoptotic as WTMYCN cells, at levels that were significantly increased compared to WT or uninduced (-DOX) controls. We thank the reviewer for the helpful suggestion, which shed light on this complex interplay of genetic backgrounds, differentiation timing, and *MYCN* expression on apoptosis. These new data are described in page 7 of the revised manuscript.

“3) 4C please show wt controls”

Response: We have now included data from unmodified DOX-treated wild type and 17q gain controls in **Supplementary Figs. 9a,c,e**. We detected no significant differences in clonogenicity, apoptosis, or DNA damage.

“4) 4D xenograft experiments – please show fish carrying the positive controls (e.g SK-N-BE2C-H2B-GFP cell lines from extended data Fig 6). Also include initial injected cells (day 0)?”

Response: Regrettably, images at day 0 were not recorded and we did not repeat these experiments. In the revised manuscript, we have now moved all data from the preliminary zebrafish xenotransplantation experiments from the main text to **Supplementary Fig. 10**, as we have added new mouse xenografting data (see also response to Reviewer 1 above).

“5) Suppl Videos 1-2: there are only 2 movies (of 17q1q and 17q1qMYCN) – please include the other two genotypes”

“6) Time lapse imagining showing higher degree of motility” please include actual quantitation to support this statement.”

Response: Due to the low resolution/frame rate of our videos, we were unable to carry out quantification of motility and therefore we have removed this sentence and the videos from the revised manuscript.

REVIEWER 3

“Interesting paper that looks at NC differentiation and effects of chromosome 17/1q gains and MYCN expression in hESCs and the impact on differentiation to NC, on chromatin and developmental TFs.”

Response: We would like to thank the reviewer for their positive comments and thought-provoking questions, which we discuss in detail below.

“1. It is likely that both MYCN amplification and CNVs in neuroblastoma represent late events, as normal cells likely activate checkpoints in response to these genetic changes. In this regard, can the authors detail whether the 17q and 17q1q ESC lines acquired mutations in genes that enabled them to tolerate these acquired CNVs? Did introduction of MYCN and differentiation to NCC change the frequency of any candidate mutations present in these lines, and absent in the parent lines? Damaging mutations or polymorphisms that cooperate with MYCN to drive proliferation might increase in frequency after introduction of MYCN.”

Response: Recent studies (Körber *et al.* (2023), *Nat Genet*; Gundem *et al.* (2023), *Nat Genet*) have indicated that CNAs appear upstream of MYCN amplification during early stages of sympathoadrenal development, in line with the scenario we are modelling in this manuscript. The CNAs (chr17q/1q chromosome arm gains) examined here are common chromosomal aberrations arising in human embryonic stem cell lines (hESC) as a result of culture adaptation during their maintenance under self-renewing-promoting conditions (Halliwell, Barbaric & Andrews (2020), *Nat Rev Mol Cell Biol*). These confer a selective growth advantage to embryonic stem cells, which tend to experience higher levels apoptosis due to increased replication stress/DNA damage (Halliwell *et al.* (2020), *Stem Cell Reports*) and a number of candidate chr17q/1q-located, “selection” driver genes have been described such as *MDM4* and *BIRC5* (Stavish *et al.* (2023), *bioRxiv*; Halliwell, Barbaric & Andrews (2020), *Nat Rev Mol Cell Biol*).

Addressing the specific points raised by the reviewer regarding the effect of MYCN overexpression on our cell lines:

- We performed a detailed analysis of the WES data (already included in the original submission) of 17q and 17q1q and found a point mutation in *BCOR* (BCL6 interacting corepressor; an epigenetic regulator sometimes considered a tumour suppressor gene) and a small loss on chr2 (not harbouring any genes with known implication in NB) present in 17q and 17q1q cells (**Supplementary Figs. 4b-d, Supplementary Tables 4,5**). Like CNAs, *BCOR* mutations are also quite commonly seen in induced pluripotent stem cell cultures (Rouhani *et al.* (2022), *Nat Genet*) and NB patients with *BCOR* mutations also have a high frequency of CNAs (see Extended Data Fig. 1 in Gundem *et al.* (2023), *Nat Genet*). *BCOR* mutations have also been reported together with CNAs in other cancers (e.g., retinoblastoma: Aschero *et al.* (2021), *Cancers*), but we are currently not aware of any data establishing a causal relationship. This will be an intriguing topic for follow-up studies.

Given the comparatively low number of other mutated genes compared to the many genes in chr17q/chr1q, we kept the cell line labels to indicate the primary NB-like CNA. However, this is of course a simplification, which we should have been clearer about from the start. We have now added additional explanations on pages 4 and 13 of the revised manuscript.

- We performed additional WES of WT, 17q, and 17q1q hESCs after 19 days of differentiation with and without MYCN induction (from D5 onward). We found few (n = 9-16) high-frequency (VAF > 0.2) mutations in cells at D19 compared to the respective cell line at D0, and no difference of MYCN overexpression on the number of mutations recorded (**Fig. 4e**, pasted below for convenience; **Supplementary Fig. 9g**). Complementarily, we also looked at DNA damage by counting γ H2AX foci and found a decrease in DNA damage in 17q and 17q1q cells at D9. Interestingly, upon MYCN overexpression, DNA damage increased significantly only in 17q1q background (**Supplementary Fig. 9d,e**). Our new data are described on pages 7-8 of the revised manuscript.

“2. Can the 17q and 1q gains in these ESCs be further detailed and compared to those found in human NB, both in regards to the region of 17q and 1q that were gained, and by sequencing comparison with human tumors that do or don’t have both regions gained?”

Response: We have now performed a detailed analysis of the CNAs in our cell lines using Sequenza (Favero *et al.* (2015), *Ann Oncol*) and included the detailed output in **Supplementary Table 5**. Additionally, we compared these coordinates with those of 17q and 1q gains in an Austrian cohort of 88 patients (from Abbasi *et al.* (2017), *Clin Cancer Res*; **Supplementary Fig. 4c**, pasted below). As expected, there is a lot of variation between patients but overall, the gains in our cell lines cover very similar regions. We have revised the text on page 4 of our manuscript to include these new data.

“3. Does turning on *MYCN* after differentiation to NC (on or after day 19) lead to similar changes as those seen in Fig 2, in which *MYCN* was turned on early in the differentiation protocol”.

Response: We compared the effect of early (day 5) vs late (day 19) DOX-induced *MYCN* overexpression on the differentiation capacity of chr17q1q gain hESC-derived trunk NC cells (**Supplementary Fig. 4h**). In line with the reviewer’s prediction, we found that both mediate a

differentiation block exemplified by the inability of the cells to generate PERIPHERIN-positive neurons at day 30. This new result is described in lines 162f in the revised manuscript.

“4. I was puzzled why the manuscript and Fig 4-7 in particular omitted analysis of *MYCN* in wild-type and 17q cells. Aren't these critical comparators to understand how *MYCN*, 1q gain and 17q gain cooperate contribute to the effects the authors observe in the 17q1q *MYCN* cells? Ideally, adding the analysis of *MYCN* in wt cells, and *MYCN* in 17q cells offers insights into distinct roles for *MYCN*, 17q, and 1q that can't be easily gleaned from comparing 17q1q with 17q1q*MYCN* in the absence of these other samples. These analyses would also help clarify roles for each genetic aberration in additional experiments detailed in subsequent experiments”.

Response: We have now included characterisation of *MYCN* overexpression in a wild type and chr 17q gain backgrounds in our study. Briefly, we found that: (i) *MYCN*-induced differentiation block is less severe in WT background (**Fig. 4a**, pasted below; **Supplementary Figs. 8c,d**); (ii) *MYCN* leads to increased proliferation and clonogenicity in 17q1q and 17q background but not in WT (**Figs. 4b-d**); (iii) *MYCN* leads to increased apoptosis in WT background. This effect is initially reduced in 17q and 17q1q, but comes back in 17q1q later in differentiation (**Supplementary Fig. 9b**); (iv) there is little difference in terms of the number of mutations or DNA damage induced by *MYCN* in all backgrounds (see previous answer; **Fig. 4e**; **Supplementary Figs. 9f,g**). A detailed analysis of cell mappings at D9 (using new scRNA-seq data), showed that chr17q gain led to more cells triaged to cluster 7 (C7) (transitional cell type including SCP-like cells expressing *ASCL1*) and additional gain of chr1q to cells triaged to C10 (another transitional cell type with an undefined identity, marked by *MSX2* and *CNTNAP2* expression). With *MYCN* overexpression, cell mappings to C4 (sensory neuronal development) were abrogated and instead many cells mapped to C2 (early cells transitioning between NMP and trunk NC stages; **Supplementary Fig. 8b**; pasted below). In 17q*MYCN* and 17q1q*MYCN*, we saw the compound effect of the shift to C2 added by *MYCN* overexpression, and the shifts towards C7 and C10 added by 17q and 1q, respectively. Our new results are described on pages 6-8 of the revised manuscript.

“5. Why did authors implant cells into fish rather than into murine adrenal gland in Fig 4? In the 16% that developed increased xenotransplant size, were these tumors transplantable? Can histology be shown, and markers of proliferation, apoptosis, senescence, mes and adr differentiation?”

Response: We have now included data from both subcutaneous and adrenal gland xenografting of 17q1q gain (+/-*MYCN*) hESC-derived trunk neural crest cells (new **Fig. 5; Supplementary Fig. 10a**). We observed striking differences between cell lines: 9 out of 9 mice injected with 17q1qMYCN cells (6 subcutaneously, 3 in the adrenal gland; in both cases receiving DOX to induce *MYCN*) developed visible growths within a month. In contrast, not a single mouse developed growths without DOX (i.e., 17q1q only, no *MYCN*; n = 6 subcutaneous, 3 adrenal) similar to control mice injected with trunk NC cells derived from wild type hESCs (n = 6 subcutaneous). Histologically, these growths were found to

resemble NB tumours and exhibited expression of the proliferation marker Ki67 (**Supplementary Fig. 10a**). The new mouse xenografting data is described on page 8 of the revised manuscript.

New Fig. 5. Xenografts into NSG mice yielded visible tumour growth within a month for 17q1qMYCN cells only (but not 17q1q or WT) following subcutaneous (a) and adrenal gland (c) injection.

REVIEWER 4

“In this manuscript, Saldana-Guerrero et al utilized an hESC model of trunk neural crest (NC) differentiation to explore the connections between CNVs, MYCN amplification, and tumor initiation in NB. The authors differentiated hESC lines with CNV gains in chr. 17q and 1q and induced MYCN overexpression to examine how these factors affect NC differentiation and the acquisition of tumorigenic characteristics. The authors have conducted single-cell seq experiments at different time points during in vitro NC differentiation of different CNV hESC lines (17q, 17q1q, and 17q1qMYCN). They showed that the CNVs promote immature NC progenitor phenotypes and impair differentiation, and that combining CNVs with MYCN overexpression exacerbates these effects and increases cell proliferation and survival. Although the in vitro NB model study is of interest, there are several major limitations in their analyses and claims to be addressed.”

Response: We would like to thank the reviewer for recognizing that our study is of interest to the community. We believe that we have now addressed the reviewer’s concerns in the revised manuscript, as detailed below.

“1. The authors used in vitro differentiation of hESC cell line with human NB-associated CNVs to model tumour initiation of NB, however, those cells including 17q1qMYCN failed to initiate NB tumours. The cells (e.g., 17q1qMYCN) they generated, although transiently increase cell survival for a couple of days in xenotransplants, are not transformed or tumorigenic, and do not model NB tumor cells, which limits the impact of the study.”

Response: We have now performed xenografting experiments in mice. Our new data (new **Fig. 5; Supplementary Fig. 10a**; see excerpts added to the previous page of this response letter) indicate that acquisition of tumorigenic hallmarks *in vitro* (e.g., cell cycle changes, increased proliferation and clonogenicity, impaired differentiation) by CNA-carrying, MYCN-overexpressing hESC-derived trunk NC cells correlates with the capacity to form tumours *in vivo*: 9 out of 9 mice injected with 17q1qMYCN cells either subcutaneously or into the adrenal gland developed tumours, while 0 out of 15 control injections with 17q1q or WT cells grew into visible tumours. This finding strongly suggests that our model recapitulates early events in neural crest-derived tumour initiation.

“2. An important reference of the iPSC cell line with overexpression of MYCN alone is lacking in the study. The authors should include the data for their analyses in Figures 2, 4, 6, and 7. The acquisition of “NB hallmarks” might be likely due to MYCN overexpression alone rather than 17q, 1q CNVs”.

Response: We have now included data to characterise the effects of MYCN overexpression in wild-type H7 hESCs and additionally also in the 17q-only background. Quoting from our response to Reviewer 3: “Briefly, we find that: (i) MYCN-induced differentiation block is less severe in 17q and WT background (**Fig. 4a; Supplementary Figs. 8c,d**); (ii) MYCN leads to increased proliferation and clonogenicity in 17q1q and 17q background but not in WT (**Figs. 4b-d**); (iii) MYCN leads to increased apoptosis in WT background. This effect is initially reduced in 17q/17q1q but comes back later in differentiation (**Supplementary Fig. 9b**); (iv) We found little difference in terms of the number of mutations or DNA damage induced by MYCN in all backgrounds (see previous answer; **Fig. 4e; Supplementary Figs. 9f,g**)”. Thus, our data indicate a complex interaction between MYCN overexpression and chr17q/1q gains, which is most evident in our new scRNA-seq data generated from trunk NC cells (focused on D9 of differentiation) from hESCs in different combinations of CNAs and MYCN: With MYCN alone, there is a retardation of development seen by a loss of sensory neuronal derivatives (C4) and instead an accumulation of cells in C2 (reminder: MYCN overexpression is induced at D5 in our experimental design, which is “in between C2 and C3”; these induced cells are quite different from normal trunk NC in C3 and thus triaged differently). With 17q alone there is an increase of C7-triaged cells and with 1q an increase of C10-triaged cells. The effect of the 17q/17q1q (C7+/+/C10+/+) states is rather independent of MYCN effect (C4-/C2+/+). C7 contains cells at a transition of identities, including SCPs, which we believe is consistent with the potentiation of an SCP signature in 17q1q seen in **Fig. 2b** (C7 along with C8 also expresses some characteristics of so-called

Bridge cells like *ASCL1*, that have attracted some attention in the NB community; **Fig. 1c**). If cells are allowed to continue through differentiation, WTMYCN cells eventually form neuronal structures with axons at D19 (but less so than without MYCN overexpression), while we did not observe these in 17qMYCN or 17q1qMYCN (**Supplementary Fig. 8e**).

The text corresponding to the quoted figures has substantially changed in the revised manuscript and can now be found on pages 6-8.

Supplementary Fig. 8b. Top: Projection of scRNA-seq data from WT, 17q, and 17q1q cells with/without *MYCN* overexpression at D9 of differentiation onto the reference differentiation landscape from Fig. 1 of our manuscript. Bottom: The same projections represented as a heatmap. The heatmap contains separate columns for all replicates of the same condition and also additional cell lines not shown in the UMAPs for simplicity (e.g., cells derived from the H9 parental cell lines that had also acquired 17q and 1q gains).

Supplementary Fig. 8e. At D19 of differentiation WT, 17q, and 17q1q cells from distinctive axonal structures indicative of sympathetic neurons. WT cells overexpressing *MYCN* also generate such cells, but 17qMYCN and 17q1qMYCN do not.

“3. In Figure 1B, the cell number on day 3 of WT cell differentiation is very low for the single-cell analysis. Since it is an important reference for comparison of other CNV mutant cells, the authors should increase the number of cells on day 3 in their analysis.”

Response: Unfortunately, we had lost multiple samples in the preparation of the single-cell dataset for the original submission (cells at the transient neuromesodermal progenitor stage at day 3 seemed particularly fragile). In the revised manuscript, we have repeated these experiments and added new scRNA-seq data for the sparsely represented stage pointed out by the reviewer. Additionally, we now provide better coverage of days 14 and 19, an additional late timepoint (D28) and several intermediate timepoints (D6, D10, D12). The revised dataset doubles the number of cells represented and timepoints to fill in detail on previously unseen stages of differentiation. The extended dataset afforded a clearer delineation of an early branching event toward sensory neurons (clusters C4, C5), an emergence of cells with partial chromaffin-like identity after a month of differentiation (in C14), and a continuum of MES-SYM cell states (all **Fig. 1; Supplementary Figs. 2, 3**), relating mutation effects and developmental timing (**Fig. 2; Supplementary Fig. 8c**), genes related to the spectrum of mutations (**Fig. 3; Supplementary Figs. 6, 7**), mappings between tumour cells and *in vitro* differentiation (**Fig. 6; Supplementary Fig. 11**), and refinement of transcription factor-target correlations (**Fig. 8; Supplementary Fig. 13**). As a result of augmenting the dataset, all mentioned figures and text details (e.g., cell and cluster numbers) in the revised manuscript have been updated. We believe that these additions further increase the value of our study as an interesting resource for developmental and cancer biologists.

“4. In Fig 2g, 17q1qMYCN cells map to day 9 of WT cells, but the SOX10 level is lower than that of other cells such as WT and 17q1q if these cells are maintained in nc state”.

Answer: This is another good observation. Indeed, the mapping to the WT reference (now **Fig. 2b,c**) resolves 17q1qMYCN cells to largely to cells from the early NC stage of development in WT (day 3-9). This is because their overall transcriptomic profile is most similar to those WT cells. However, the mutant cells do no longer express some key features of proper trunk NC development, including decreased *SOX10* (**Fig. 2g, h**). This is further evidenced by the mapping to the *in vivo* reference, where 17q1qMYCN have only “weak” matches (foetal adrenal glands; **Fig. 2e,f**). We interpreted this as an unsuccessful specification of the trunk NC identity (“suggesting a failure to fully specify the expected cell types”, l. 157) and it is likely that the loss of *SOX10* expression impairs the ability of the cells to further differentiate toward the trunk NC lineage (see also the loss of a *SOX10*-centered GRN in **Fig. 8e,f**), in line with the known critical role of *SOX10* (Southard-Smith *et al.* (2023), *Nat Genet*; Britsch *et al.* (2001), *Genes & Dev*; Carney *et al.* (2006), *Development*; Kim *et al.* (2023), *Neuron*). Notably, when integrating the mutant data with WT (**Fig. 3d**), 17q1qMYCN cells form largely separate clusters (M4, 5, 7, 12; **Supplementary Fig. 6b**), indicating a cell state that is distinct from those assumed during normal development. This is driven by the genes shown in **Fig. 3f**.

“5. Along the same line, *Plp1* is predominantly expressed in 17q1qMYCN cells, however, it has been shown as a Schwann cell lineage marker. The authors should use canonical nc markers to differentiate them from Schwann cell lineage cells.”

Response: There was a typo in the Reviewer’s comment here: the expression of *PLP1*, together with a set of other markers indicative of a Schwann cell precursor (SCP) transcriptional signature, is enriched in 17q1q cells rather than 17q1qMYCN cells. This signature decreases upon *MYCN* overexpression in the same genetic background (**Fig. 2g**). Our three main labels (Schwann cell precursors, mesenchymal cells, sympathoblasts) and cell type markers are based primarily on the classification in the scRNA-seq analysis of human foetal material from Kameneva *et al.* (2021), which is highly relevant to neural crest development/NB biology and hence our work (cp. **Fig. 1d**). However, the data does not capture some of the very earliest lineage branching events, such as the branching off of sensory neuronal development from early NC lineage, which is why, for example, we observed a weak match of those sensory neuronal-like cells to sympathoblasts, i.e., the “next-best match” in the foetal reference data (**Fig. 1f**). In the same way, we observed a weak match of the earlier trunk NC (~D9 of our differentiation) to

SCPs (**Fig. 1e**). We have chosen the Kameneva reference data as one of the popular reference datasets known to most of the NB research community and provide alternative mapping to another popular resource (Jansky *et al.* (2021), *Nat Genet*). We aimed to stick with a categorization to the three main cell types (SYM/MES/SCP) for simplicity but trust expert readers will be able to identify more fine-grained distinctions from the cell type marker profiles (**Supplementary Tables 2**), and we will use the time post submission to implement interactive visualization features (cellxgene and Loupe files) to allow future readers to quickly explore the data.

“6. In Figure 3f, the authors should include the data with human NBs that have the same copy number variants (17q1q) and MYCN overexpression since they were identified in the majority of NB tumours. In Fig 3a, the authors should include human NB tumours for the pathway analysis”.

Response: In line with the reviewer’s suggestion, we compared data from RNA-seq of NB tumours from SEQC, TARGET, and RNA-seq data collected at our institution to the genes shown in **Fig. 3f**. This analysis demonstrated high expression of many genes upregulated in 17q1qMYCN cells (gene set D9_1) in MYCN-amplified tumours, while genes that are downregulated in 17q1qMYCN cells or upregulated with 17q/1q (gene sets D9_2-D9_4) were rather lowly expressed in these tumours. Regrettably, annotations for chr17q and 1q gains for these datasets are incomplete but it does not appear that there are any further groupings among the tumours based on the other three gene sets.

“7. Figure 4 cell survival and proliferation assays, the authors should include an iPSC cell line with overexpression of MYCN alone.”

Response: We have now included these data in the revised manuscript -- please also refer to our response to comment 2. Briefly, MYCN leads to increased proliferation and clonogenicity in 17q1q and 17q background but not in WT (**Figs. 4b-d**).

“8. In Fig. 5d, the projection of tumor cells is hard to detect in the UMAP. The chromaffin-like and cortex-like cells are so rare, which makes it difficult to compare with SYM-like NB cells. The authors should include more chromaffin-like and cortex-like tumour samples.”

Response: We apologize for the confusing display in in “old” Figure 5 of our initial submission (which has now become **Fig. 6** in the revised manuscript; pasted below in this response letter). As our *in vitro* differentiation protocol employs culture conditions that promote predominantly the generation of sympathoblasts/sympathetic neurons, the spontaneous emergence of chromaffin-like cells in the same cultures is a rare event (we begin to see some cells with partial chromaffin-like identity at D28 of differentiation (see revised **Fig. 1**). In the old figure, the “chromaffin-like” and “cortex-like” labels used to refer to the mapping of cells from one particular NB tumour (from Dong *et al.* 2020) to the foetal adrenal gland reference (from Kameneva *et al.* 2021) and neither to different tumour samples nor to a mapping to our dataset. As this was just auxiliary information and not really part of our study, we have removed this mapping from our revised manuscript. Overall, we attempted to streamline this part of the manuscript, in line with the reviewers’ suggestions. Briefly, the new revised figures show that cells from the same, karyotypically homogeneous tumour, map to different cell types in our *in vitro* reference (**Fig. 6a,b; Supplementary Fig. 11**). Intersection of the markers of the *in vitro* cell types and DEGs between tumour cells mapped to those clusters, provides a refined list of gene signatures presenting different NB tumour cell subsets (**Fig. 6c**). Applied to a large collection of bulk RNA-seq data with associated clinical data from NB tumours, these gene signatures also correlate with clinical staging and prognosis (**Figs. 6d,e**), see next point.

“9. In Fig. 5f heatmap, MYCN expression in different tumor cells (LATE SENSORY, HDMYCN, SCPSYM and SYM) appears inconsistent with Fig5e, which shows MYC amplifications in all these NB tumors. Also, in Figure 5g, the authors should explain the significance of the tumour heterogeneity analysis. Does the predicted score correlate with malignancy or patient survival information?”

Response: The heatmap in old Fig. 5f was based on the same example tumour as old Fig. 5d (“Dong_230”). In fact, all cells in selected tumours are thought to harbour a MYCN amplification (this was a selection criterion for the tumour datasets we included), but they express MYCN at variable levels (that is what is shown in the side panel of the heatmap). In the old Fig. 5f we chose to scale gene expression counts by row to emphasise expression differences between the different groups of cells. This resulted in the MYCN-high cells appearing in orange colour, while other cells (which still expressed MYCN, but at lower levels) appeared in blue colour. Therefore, in fact both panels were consistent. However, we acknowledge that the display might not have been optimally chosen causing confusion. In the revised figure (Fig. 6; pasted above), we have picked cells from tumour “Jansky_NB14” as an example to display in the new heatmap in Fig. 6b (see above). We show a mapping to our wild-type *in vitro* differentiation atlas to illustrate that genetically similar cells (seen via the inferCNV heatmaps) map to both C13/C14 and other clusters, representing different developmental stages. We had previously refrained from making speculative statements about the relevance of different mappings due to the low number of samples analysed, which is additionally confounded by technical differences (“batch effects” → different studies). In the revised manuscript we took the analysis a step further by examining gene expression signatures identified in our single-cell data in comparison to NB tumours from a large collection of well-annotated RNA-seq data from SEQC. We found that our gene expression

signature C13* correlated with clinical staging (low signature strength ~ stage 4; **Fig. 6d**) and, as the reviewer had predicted, the C13* signature also distinguished between subgroups with differences in survival (low C13* ~ worse survival both across the whole cohort and even within stage-4 cases only; **Figs. 6e,f**). We have substantially revised this entire section of our manuscript, see pages 8-9 of the revised manuscript.

“10. In Fig 6, the authors should compare the data from their cell lines with the published human NB tumor data (bulk RNA-seq, single RNA-seq data, and ATAC-seq data) to show if the CNV cell lines exhibit similar tumorigenic hallmarks.”

Response: We have surveyed publicly available epigenomics data for NB tumours and did not identify any ATAC-seq data from primary NB tumours, but we found several publications providing ATAC-seq data for NB cell lines. As suggested by the reviewer, we used these data to interrogate the accessibility of the chromatin modules R1-R9 in NB cell lines. To overcome problems with batch effects, we performed a gene set enrichment analysis (GSEA) to analyse each dataset separately and additionally included ATAC-seq data from two adult cancer cell lines and ATAC-seq data from EpiMap (Boix *et al.* (2021), Nature; a re-analysis of data from ENCODE and others) as controls. Our results (**Supplementary Fig. 12g**) indicate that module R8 was accessible in all samples (consistent with the enrichment of “tissue invariant” and “adult stromal” terms in **Supplementary Figs. 12d,e**) and also modules R6 and R7 were accessible in many (but not all) NB and other cancer cell lines and tissues. Chromatin regions in R3 (corresponding to genomic regions accessible at the trunk NC stage of differentiation in our dataset) were accessible in NB cell lines, brain, and reproductive tissue. Interestingly, we also found this accessibility to vary slightly between NB cell lines with an ascribed adrenergic (more R3 accessibility) and mesenchymal (less R3 accessibility) character.

Additionally, we also interrogated the expression of TFs identified in our gene-regulatory network analysis (“old” Fig. 7, now **Fig. 8**) in the NB bulk RNA-seq datasets mentioned in the preceding responses. We found that many of the developmental regulators are expectedly weakly expressed in developed tumours (**Supplementary Fig. 14a**), but interestingly the expression of targets of the TFs with dynamic activity across the “mutational spectrum” in our *in vitro* models also clustered with distinct groups of NB tumours (**Supplementary Fig. 14b**). The most well interpretable clustering was related to the presence of *MYCN*-amplification in NB tumours, which clearly correlates with a strong expression of *MYCN* target genes and the targets of other TFs identified in the respective network module together with *MYCN* (e.g., *NR1D1*, *TFAP4*). There are also clusters of NB tumour data based on the activity of target genes of the other TF groups, which one might speculate could relate to presence/absence of CNAs like chr17q, but these aberrations are not sufficiently annotated to draw any conclusions.

The new analyses are described on pages 10-12 of the revised manuscript and both figures are pasted below / on the next page for convenience.

Supplementary Fig. 12g. Quantification of chromatin modules R1-R9 in public ATAC-seq data from NB cell lines (GSE138293, GSE224241, GSE136279) compared to breast (GSE202511) and lung (GSE228832) as well as data from normal tissues (EpiMap).

Supplementary Fig. 14. Expression of TFs that are shown in our gene-regulatory network analysis (Fig. 8) across human NB tumour datasets (panel a), and of the targets of these TFs in the same datasets (panel b).

REVIEWERS' COMMENTS

Reviewer #1 (Remarks to the Author):

The authors have addressed all of my comments in sufficient detail. I congratulate the authors for this important work and look forward to seeing this in print.

Reviewer #1 (Remarks on code availability):

I am not a computational biologist and can not run these codes.

Reviewer #2 (Remarks to the Author):

Saldana-Guerrero et al have done an excellent job addressing reviewers comments, including adding substantial new data. Most significantly, new Figure 5 uses mouse xenograft models to demonstrate pathogenicity of the 17q1qmycn cells.

I commend the authors for their excellent job on this paper, and for being clear regarding limitations and challenging experiments. I am satisfied that the authors have addressed all my concerns.

Reviewer #3 (Remarks to the Author):

Revised manuscript addresses points raised in earlier review.

Reviewer #4 (Remarks to the Author):

In the revised manuscript, the authors have included in vivo xenograft data demonstrating the ability of 17q1qMYCN cells to form neuroblastoma tumors, along with further analyses to address the concerns previously raised. Their responses are satisfactory. I recommend it for publication.